# Glycogen phase-separation drives macromolecular rearrangement and asymmetric division in *E. coli*

Yashna Thappeta[1,2,12], Silvia J Cañas-Duarte [1,3,12], Haozhen Wang[1,3], Till Kallem[4], Alessio Fragasso[1,2], Yingjie Xiang[5], William Gray[6], Cheyenne Lee [7,10], Georgeos Hardo [8,11], Lynette Cegelski[4] & Christine Jacobs-Wagner [1,2,3,9✉]

## Abstract

**Bacteria often experience nutrient limitation. While the exponential and stationary growth phases have been characterized in the model bacterium *Escherichia coli*, little is known about what happens inside individual cells during the transition between these two phases. Through quantitative cell imaging, we found that the positions of nucleoids and cell division sites become increasingly asymmetric during the transition phase. These asymmetries were accompanied by an asymmetric reorganization of protein, ribosome, and RNA probes in the cytoplasm. Results from live-cell imaging experiments, complemented with genetic and [13]C whole-cell nuclear magnetic resonance spectroscopy studies, show that preferential accumulation of the storage polymer glycogen at the old cell pole leads to the observed rearrangements and asymmetric divisions. Live-cell atomic force microscopy analysis, combined with in vitro biochemical experiments, suggests that these phenotypes are due to the propensity of glycogen to phase-separate into soft condensates in the crowded cytoplasm. Glycogen-associated differences in cell sizes between strains and future daughter cells suggest that glycogen phase-separation allows cells to store large glucose reserves that are not perceived by the cell as cytoplasmic space.**

**Keywords** Asymmetric Division; Bacteria; Glycogen; Nutrient Limitation; Phase Separation

**Subject Categories** Cell Adhesion, Polarity & Cytoskeleton; Microbiology, Virology & Host Pathogen Interaction; Organelles

## Introduction

Spatial order is an inherent feature of living cells across the tree of life. In eukaryotic cells, this is exemplified by the presence of membrane-bound organelles. These cells also display extensive membrane-less organization, including that mediated by phase separation between various cellular components (Alberti and Hyman, 2021; Banani et al, 2017; Boeynaems et al, 2018; Hyman et al, 2014; Rostam et al, 2023). Although bacteria typically lack membrane-bound organelles in their cytoplasm, they exhibit spatial organization at multiple levels (Surovtsev and Jacobs-Wagner, 2018), in the form of protein localization, chromosome structure, and phase-separated condensates involving proteins and/or nucleic acids, to name a few (Azaldegui et al, 2021; Bakshi et al, 2012; Landgraf et al, 2012; Cohan and Pappu, 2020). While subcellular organization is particularly evident in bacteria that undergo developmental programs and/or have dimorphic life cycles (e.g., sporulating bacteria, asymmetrically dividing α-proteobacteria), even bacteria with comparatively simpler life cycles, such as *Escherichia coli*, display spatial organization. For example, *E. coli* divides precisely in the middle of the cell through two separate mechanisms. The time-average pole-to-pole oscillation of Min proteins positions the FtsZ cytokinetic ring at mid-cell (Lutkenhaus, 2008; Shih and Zheng, 2013), while nucleoid occlusion prevents cell constriction over nucleoid regions (Woldringh et al, 1990). Furthermore, *E. coli* spatially organizes two of its most important cytoplasmic components: the chromosome and polysomes (mRNAs in complex with ribosomes engaged in translation). The chromosome, which compacts into a meshwork structure known as the nucleoid, is spatially positioned inside cells, either near the cell center prior to DNA replication or near the ¼ and ¾ cell positions after DNA replication and segregation (Sherratt, 2003; Badrinarayanan et al, 2012; Bates and Kleckner, 2005). Polysomes adopt a near-opposite localization profile. They are depleted in the nucleoid and enriched in nucleoid-free regions (Azam et al, 2000; Bakshi et al, 2012; Chai et al, 2014). This anticorrelated pattern is thought to be mediated at least in part by the steric exclusion of polysomes from the nucleoid mesh (Mondal et al, 2011; Papagiannakis et al, 2025; Castellana et al, 2016). In contrast, smaller components such as free ribosomal subunits and cytoplasmic proteins can diffuse throughout the nucleoid

[1]Sarafan Chemistry, Engineering, and Medicine for Human Health Institute, Stanford University, Stanford, CA, USA. [2]Department of Biology, Stanford University, Stanford, CA, USA. [3]Howard Hughes Medical Institute, Stanford University, Stanford, CA, USA. [4]Department of Chemistry, Stanford University, Stanford, CA, USA. [5]Mechanical Engineering and Materials Science, Yale University, New Haven, CT, USA. [6]Department of Pharmacology, Yale University, New Haven, CT, USA. [7]Microbial Sciences Institute, Yale University, New Haven, CT, USA. [8]Department of Engineering, University of Cambridge, Cambridge, Cambridgeshire, UK. [9]Department of Microbiology and Immunology, Stanford University, Stanford, CA, USA. [10]Present address: Department of Molecular Genetics and Microbiology, Duke University, Durham, NC, USA. [11]Present address: Department of Biology, United Arab Emirates University, Al Ain, Abu Dhabi, UAE. [12]These authors contributed equally: Yashna Thappeta, Silvia J Cañas-Duarte. ✉E-mail: jacobs-wagner@stanford.edu

unimpeded (Amselem et al, 2023; Bakshi et al, 2012; Sanamrad et al, 2014), resulting in homogeneous distribution throughout the cytoplasm. This archetypical macromolecular organization is reproduced every generation.

The bulk of our knowledge of *E. coli* (and other well-studied bacteria) has largely been derived from exponentially growing cultures. In their natural environments, *E. coli* and other bacteria often experience nutrient deprivation to which they must adapt (Dworkin and Harwood, 2022). In the laboratory, nutrient deprivation occurs in batch cultures when cells exit exponential growth and transition to the stationary phase due to the limitation of an essential nutrient. Growth arrest, however, is not immediate. As nutrient availability decreases, cell growth slows progressively until the cell density reaches saturation, marking the onset of the stationary phase (Dworkin and Harwood, 2022). During this transition, cells are thought to block new rounds of DNA replication before they stop dividing, ultimately entering the stationary phase with a smaller size and lower chromosome content on average relative to their exponential phase counterparts (Akerlund et al, 1995). Yet, the transition between exponential growth and the stationary phase remains under-characterized, especially at the single-cell and subcellular organizational levels. This study sets out to fill this gap in knowledge.

## Results

### Population characteristics change along the culture growth curve

We first used microscopy and DNA staining with 4',6-diamidino-2-phenylindole (DAPI) to quantify various cellular characteristics (cell area, nucleoid area, number of nucleoids per cell, fraction of dividing cells, intracellular positioning of the division site and of the nucleoid, etc.) along the growth curve of liquid cultures of *E. coli* at 30 °C in M9 medium containing 0.2% glucose, casamino acids and thiamine (M9gluCAAT). Under our experimental conditions, cultures exhibited exponential growth up to an optical density (OD) at 600 nm of about 1.0, after which population growth slowed down gradually until reaching saturating ODs around 3.0 (Fig. 1A). While the culture might experience a diauxic shift in this medium when transitioning between exponential and stationary phases, we will hereafter refer to this period of sub-exponential growth (between ODs ~1 and ~3) as "transition phase" for simplicity (Fig. 1A). This phase is more readily discernible in a logarithmic scale than a linear one (Fig. 1A, inset), as it better highlights the deviation from exponential growth.

Imaging live cells from the same liquid culture at a high sampling resolution along the growth curve was technically impractical with our setup. Therefore, we imaged DAPI-stained cells (Fig. 1B) from independent cultures at multiple ODs on different days and examined how various cellular characteristics may change as a function of the OD. As expected (Buchanan, 1918; Morita, 1990; Bakshi et al, 2021), we found that the average cell area decreased with increasing OD (Fig. 1C). The mean nucleoid area also decreased as cells entered transition phase (Fig. 1D), as did the mean number of nucleoids per cell (Fig. 1E). As a result, the fraction of cells with a single nucleoid increased in transition phase, reaching a plateau at OD ~ 1.5, before the cell density saturated

(Fig. 1F). Meanwhile, the fraction of cells undergoing constriction at the division site gradually decreased around the onset of transition phase (OD ~ 1) and continued to decline until the culture reached stationary phase (Fig. 1G). These results are consistent with cells undergoing one or more so-called "reductive divisions" after DNA replication initiation has stopped (Nyström, 2004; Nyström and Kjelleberg, 1989). This uncoupling between DNA replication and cell division explains the reduction in chromosome copy number per cell by the time the culture enters the stationary phase (Akerlund et al, 1995).

To our surprise, while the position of the cell constriction site was symmetric across cell populations sampled during the exponential phase (OD < 1), it became increasingly asymmetric during sub-exponential growth in the transition phase (OD > 1) (Fig. 1H). Transition phase cells with asymmetric constriction sites also displayed asymmetrically positioned nucleoids, creating larger DNA-free space at one pole than the other (Fig. 1B, inset where the magenta arrowheads indicate the cell constriction sites). To quantify this asymmetric feature, we calculated the absolute distance between the nucleoid mid-point and the cell center, normalized by cell length. For cells with two nucleoids, we averaged their relative mid-points such that when the center of each nucleoid was positioned at the quarter cell positions, the mean mid-point was located at the cell center (see schematic in Fig. 1I). In exponentially growing populations, the mean nucleoid mid-point was maintained close to the cell center (Fig. 1I, gray curves). In the stationary phase, the nucleoid position was more variable (Fig. 1I, blue curves), as qualitatively noted before (Chai et al, 2014). The most striking phenotype was observed in the transition phase when populations often displayed a bimodal distribution of nucleoid mid-points (Fig. 1I, red curves). This result implies that the nucleoid mid-point tends to be closer to a pole than the cell center, confirming visual inspection (Fig. 1B, inset).

### Cells exhibit asymmetric intracellular distributions of ribosomes, RNAs, and cytosolic protein probes in the transition phase

Given the common offset in division site and nucleoid positioning in transition-phase cells, we wondered whether the intracellular organization of other macromolecules also changes when cells exit the exponential phase. First, we examined the localization of mCherry-tagged ribosomal protein RplA and freely diffusing msfGFP in the cytoplasm. As expected, in exponential phase, msfGFP was distributed homogeneously throughout the cytoplasm while the RplA-mCherry signal was enriched outside the DAPI-stained nucleoid regions (Fig. 2A). The latter is consistent with most ribosomes forming polysomes (Phillips et al, 1969; Mohapatra and Weisshaar, 2018) and polysomes being partially excluded from the nucleoid (Azam et al, 2000; Bakshi et al, 2012; Chai et al, 2014). In contrast, in transition phase, both msfGFP and RplA-mCherry signals were depleted from the cell pole farthest away from the asymmetrically localized nucleoid labeled with DAPI (Fig. 2A). This depletion was confirmed quantitatively across the population ($n > 448$ cells) by calculating the normalized pole signal difference, which is the fluorescence signal at one (randomly selected) pole subtracted from the corresponding signal at the other pole, and divided by the average fluorescence in the cell. The distributions of normalized pole signal differences were narrow and close to 0 in the

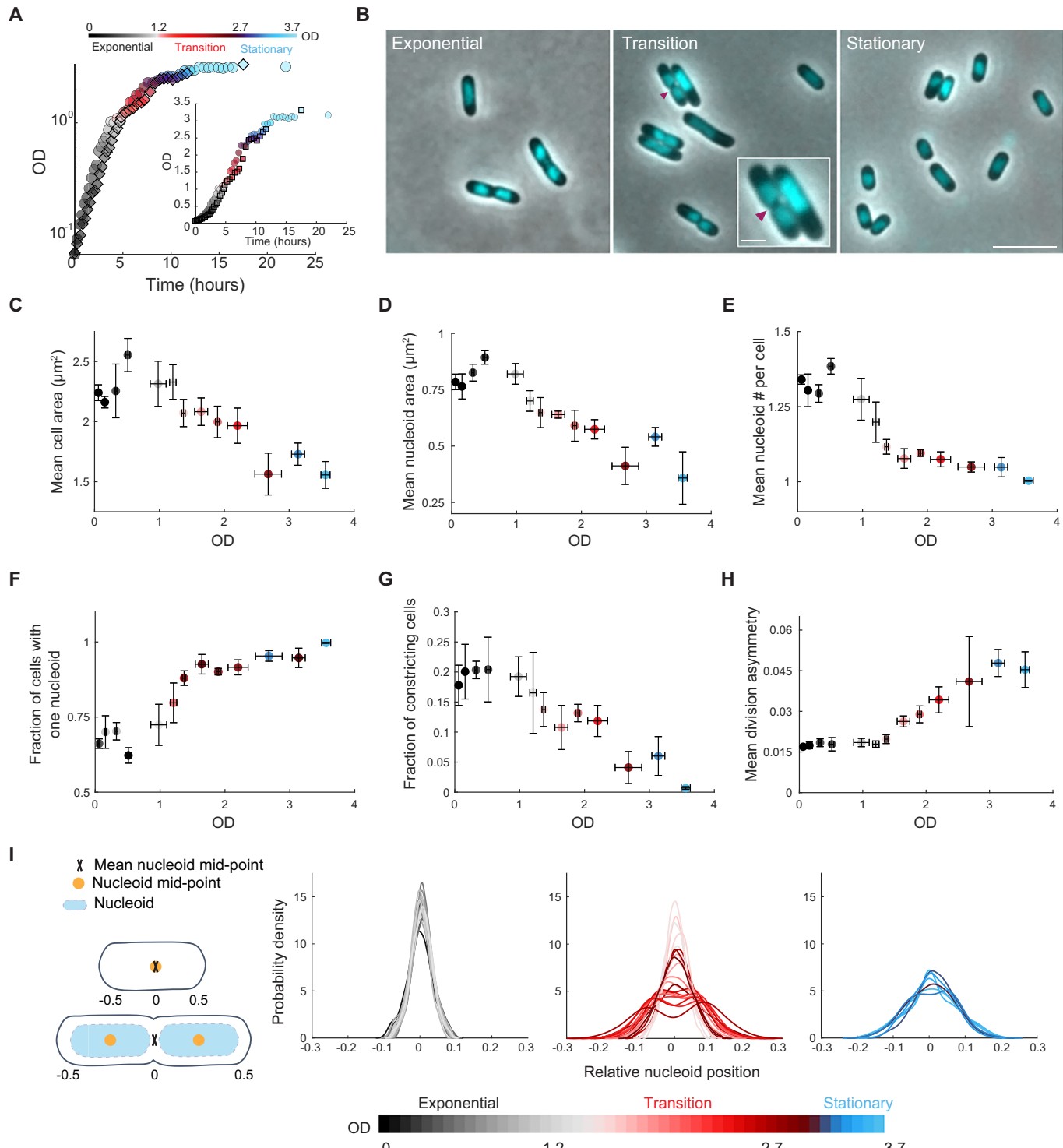

exponential phase for both msfGFP and RplA-mCherry (Fig. 2B), consistent with near-equal concentrations of fluorescent markers between cell poles. In contrast, the distributions of the normalized pole differences were broader and bimodal in transition-phase cells (Fig. 2B). In these cells, signal pole differences that were negative (positive) for msfGFP were also negative (positive) for RplA-mCherry (Fig. 2B). This correlation (Spearman's correlation

$\rho = 0.67$) indicates that the ribosomes and protein probes are preferentially depleted from the same pole in transition-phase cells. We observed the same intracellular reorganization in the transition phase using other ribosomal protein fusions (RplA-msfGFP, RplA-mCherry, or RpsB-msfGFP) and another cytoplasmic protein probe (mScarlet-I) (Fig. EV1A,B). Furthermore, GFP derivatives with surface net charges varying between −30 and +25, which reflect

**Figure 1. Cellular features change as the cell density of the culture increases.**

(A) Growth curves of two independent cultures of strain CJW4677. The datapoints for the same culture have the same shape (spherical vs square). OD measurements were manually collected by spectrophotometry over time. (B) Representative phase-contrast images of DAPI-labeled cells in exponential (OD = 0.5), transition (OD = 2.0), and stationary phase (OD = 3.25). The magenta arrowhead indicates the site of cell constriction. The inset shows a zoomed view of two cells in the transition phase displaying cellular asymmetries in nucleoid and/or constriction positioning. The scale bars in the inset and larger field of view are 1 and 5 μm, respectively. (C) Scatter plot of OD vs. mean cell area. Samples of different cultures of strain CJW4677 were collected at 35 different ODs (*n* > 500 cells for each OD) for cell imaging. The binned data for the indicated OD ranges are shown, with the mean describing the averaged mean value for cell samples from three OD values and error bars indicating the standard deviation. Gray, red, and blue shades indicate exponential, transition, and stationary phases, respectively. (D) Same as (C) but for OD vs. mean nucleoid area. (E) Same as (C) but for OD vs. mean number of nucleoids per cell. (F) Same as (C) but for OD vs. fraction of cells with one nucleoid. (G) Same as (C) but for OD vs. the fraction of constricting cells in the population. (H) Same as (C) but for OD vs. mean division asymmetry, calculated by dividing the absolute distance of the division position from the cell center by cell length and averaging across cells. This was done for cultures at ODs at which the fraction of dividing cells in the population was greater than 0.05. (I) Kernel density plots of the relative nucleoid positioning for cells in exponential, transition, or stationary phases, determined by calculating the absolute distance between the nucleoid mid-point and cell center, normalized for cell length. Note that for cells with two nucleoids, their relative mid-points were averaged such that when the center of each nucleoid was positioned at the relative quarter cell positions, the mean mid-point was at the cell center (see schematic). Source data are available online for this figure.

most of the range of net surface charges in the *E. coli* proteome (Schavemaker et al, 2017), were similarly enriched in the nucleoid region during the transition phase (Fig. EV1C). The comparable localization phenotype across cytoplasmic protein probes of different charges and genetic origins (Figs. 2A and EV1B,C) suggests that this subcellular reorganization is likely common among cytoplasmic proteins during the transition phase.

To determine whether this change in localization pattern extends beyond ribosomes and cytoplasmic protein probes, we also assessed RNA localization in two ways. First, we used a cytosolic mutant variant of RNase E, RNase E ΔMTS, fused to mCherry (Strahl et al, 2015), based on the assumption that its localization would, at least in part, reflect that of its RNA substrates. RNase E, which is involved in the degradation of mRNAs and the processing of rRNAs (Strahl et al, 2015), is normally membrane-associated via a short membrane-targeting sequence (MTS) (Khemici et al, 2008). Deletion of the MTS sequence releases the protein into the cytoplasm (Khemici et al, 2008). We found that in exponentially growing cells, a mCherry fusion to ΔMTS RNase E mutant exhibited various degrees of accumulation in nucleoid-free regions (Fig. 2C). This localization pattern changed during transition phase, with RNase E ΔMTS-mCherry primarily colocalizing with the asymmetrically localized nucleoid, resulting in a marked depletion of the RNase E ΔMTS-mCherry marker at the pole most distal to the nucleoid (Fig. 2C). This switch was confirmed at the population level by the drastic change in the distribution of the normalized pole signal differences between phases (*n* ≥ 3110 cells, Fig. 2D). The values in signal pole difference were centered around 0 in exponential phase, indicating a near-equal distribution of probe signal between the cell poles. In contrast, the distribution became bimodal with peak values deviating from 0 (Fig. 2D), indicative of pole depletion, as observed for the ribosomes and protein markers (Fig. 2B). Second, we imaged cells incubated with SYTO RNASelect, a membrane-permeable fluorogenic RNA dye that has been validated in *E. coli* (Bakshi et al, 2014). We found that the RNASelect signal, which was relatively homogeneously distributed in exponential-phase cells, exhibited depletion at a cell pole during the transition phase (Fig. EV1D).

So far, for our analyses of signal depletion at the cell poles, we used snapshot images where the pole identity was unknown and thus randomly assigned. To examine whether the pole selection is stochastic or deterministic, we carried out time-lapse microscopy

on transition-phase cells carrying RplA-GFP and a DNA-binding protein HupA fused to mCherry to visualize the nucleoid and ribosomes, respectively. These experiments revealed that the HupA-mCherry-labeled nucleoids of cells in the transition phase remained closer to the new pole generated at division (Fig. 2E; Movie EV1). The ribosome signal largely followed the nucleoid, resulting in ribosome depletion preferentially at the old pole (Fig. 2E; Movie EV1). This pattern was observed in 88.6% of all observed division events (695 out of 784), consistent with a deterministic pole selection.

## The cytoplasmic reorganization in the transition phase cannot be explained by membrane retraction or nucleoid association

It was recently reported that, in the stationary phase or upon acute starvation, the inner membrane of *E. coli* cells often retracts due to the sudden loss of cytoplasmic water and atrophy of the cytoplasmic volume (Shi et al, 2021). Therefore, we wondered whether a membrane collapse may explain both the observed nucleoid position asymmetry and the enrichment of cytoplasmic components in the nucleoid space in the transition phase. Based on phase-contrast images and cell staining with MitoTracker Green, a fluorescent dye that labels the inner membrane (Shi et al, 2021), we found no evidence of membrane retraction driving fluorescent signal depletion at the poles during the transition phase (Fig. EV2A). In fact, membrane retraction was rare during this phase and was observed in only 2 cells out of 3035.

Interestingly, when we generated filamentous polyploid cells in transition phase using the cell division inhibitor cephalexin (Hedge and Spratt, 1985; Pogliano et al, 1997; Rolinson, 1980), we observed that the ribosome (RplA-mCherry) and protein (msfGFP) probes colocalize with the nucleoids and are depleted in nucleoid-free spaces, including between nucleoids (Fig. 2F). We confirmed this localization pattern for msfGFP using an FtsZ depletion strain as an orthogonal method to block cell division (Fig. EV2B). Given the colocalization with the nucleoid, we questioned whether the observed rearrangement of cytoplasmic components may be driven by direct or indirect association with the nucleoid. Two lines of evidence suggest otherwise. First, in filamentous polynucleoid cells, we occasionally observed a sharp accumulation of fluorescently labeled ribosomes and msfGFP in nucleoid-free space (Fig. 2G, purple arrowheads within the bracket area). Second, we noted that

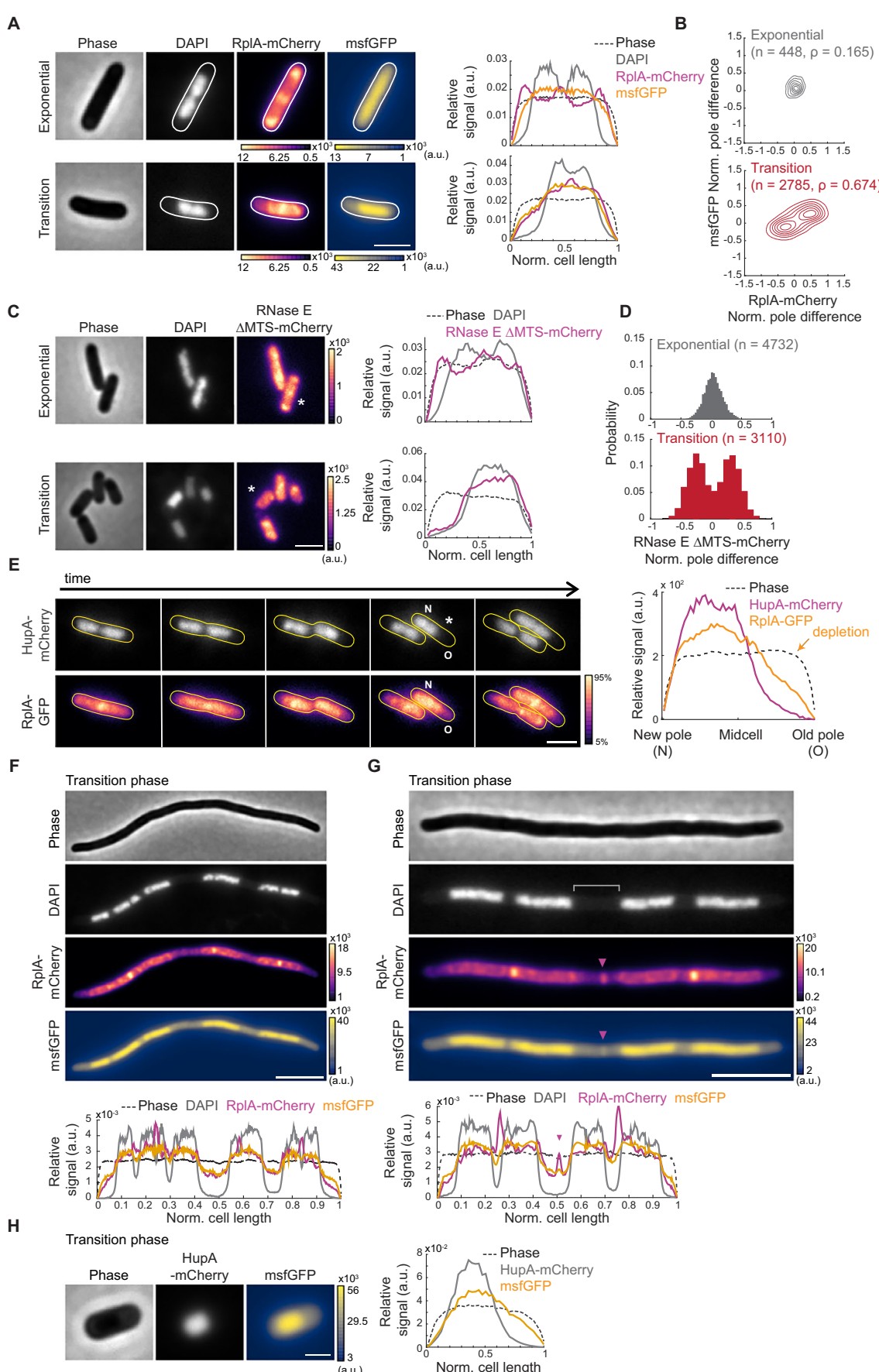

**Figure 2.   The localization pattern of cytoplasmic components changes in the transition phase.**

(A) Representative phase-contrast and fluorescence images of DAPI-labeled CJW7325 cells expressing RplA-mCherry and msfGFP, along with corresponding cell signal intensity profiles. Cultures were in exponential phase (OD = 0.14) or transition phase (OD = 1.83), as indicated. Fluorescence intensity is indicated in arbitrary units (a.u.). Scale bar: 2 μm. (B) Contour plots showing the normalized pole difference of msfGFP and RplA-mCherry signals in cells in exponential (top, $n = 448$) or transition (bottom, $n = 2785$) phase. The normalized pole difference of a fluorescent signal corresponds to the difference in this signal between pole regions divided by the average signal intensity across the cell. The contour lines represent the 0.167, 0.33, 0.5, 0.66, 0.83, and 1 kernel density envelopes of the data. (C) Representative images of DAPI-stained CJW5685 cells expressing RNase E ΔMTS tagged with mCherry in exponential (OD = 0.1) or transition (OD = 1.3) phase, along with signal intensity profiles for the indicated (*) cell. Scale bar: 2 μm. (D) Histograms of normalized pole difference of RNase E ΔMTS-mCherry in CJW5685 cells in exponential (OD = 0.1, $n = 4732$) or transition (OD = 1.7, $n = 3110$) phase. (E) Time-lapse sequence of a cell (CJW5159) expressing RplA-GFP and HupA-mCherry, with "N" and "O" indicating the new and old poles, respectively. Cells in exponential phase at OD ~ 0.5 were washed in spent medium from a transition-phase culture (OD = 2.63) and spotted on an agarose pad containing the same spent medium. The RplA-GFP images were scaled to reflect the 5-95% range of signal intensity for each image. The signal intensity profile is provided by the cell marked by an asterisk. Scale bar: 2 μm. (F) Representative phase-contrast and fluorescence images of a cephalexin-treated and DAPI-labeled CJW7325 cell carrying RplA-mCherry and msfGFP in the transition phase, with the corresponding signal intensity profiles shown below. Fluorescence intensities are indicated in arbitrary units (a.u.). Scale bar: 5 μm. (G) Same as (F) but highlighting a band of enriched RplA-mCherry and msfGFP signals (purple arrowheads) in the region lacking DAPI staining (gray bracket). Scale bar: 5 μm. (H) Phase-contrast and fluorescence images of a CJW7326 cell carrying HupA-mCherry and msfGFP, along with the corresponding signal intensity profile. Scale bar: 1 μm. Source data are available online for this figure.

the intensity profiles of the msfGFP signal extend beyond that of the DAPI signal (Fig. 2A). This was not due to the lower excitation wavelength of DAPI relative to that of the fluorescent protein probes, as the msfGFP signal also extended beyond the nucleoid signal labeled with HupA-mCherry (Fig. 2H), which emits at a higher wavelength than msfGFP. Thus, the intracellular rearrangement of ribosomes and proteins in the transition phase is not driven by an association with the nucleoid. Instead, it suggests that these cytoplasmic components are excluded by a cytosolic element that accumulates in nucleoid-free space during the transition phase, particularly at the old pole when cells are allowed to divide.

## Glycogen accumulation contributes to the transition-phase phenotypes

In our search for the cytosolic element driving the observed intracellular reorganization in the transition phase, we used whole-cell $^{13}C$ cross-polarization magic-angle spinning (CPMAS) solid-state NMR to compare the carbon composition of cells between exponential and transition phases. We found that in comparison to exponential-phase samples, transition-phase samples showed a large increase in carbon intensities in the $^{13}C$ CPMAS NMR spectra between 110 and 55 ppm (Fig. 3A, top spectra), characteristic of polysaccharide contributions. We speculated that these carbon intensity increases might correspond to an accumulation of glycogen, a glucose polymer that *E. coli* cells can accumulate under limitation of a nutrient such as nitrogen, sulfur, or phosphate (Preiss, 1984; Wilson et al, 2010; Preiss and Romeo, 1994; Mulder et al, 1962; Sigal et al, 1964; Madsen, 1963; Zevenhuizen, 1966). Furthermore, glycogen has been reported to accumulate at cell poles (Shively, 1974; Preiss, 1984; Alonso-Casajús et al, 2006; Preiss and Romeo, 1994; Liu et al, 2021b). Consistent with our speculation, the $^{13}C$ CPMAS NMR spectra of bovine and mussel-derived glycogen were consistent with the spectral changes observed in the transition-phase samples (Fig. 3A, middle spectra). We then generated a glycogen-deficient strain (ΔglgBXCAP) by deleting the glycogen metabolic operon from the chromosome. The $^{13}C$ CPMAS NMR spectrum of transition-phase samples from this glycogen-deficient strain lacked the large carbon intensity increase observed in the wild-type (WT) strain (Fig. 3A, bottom spectra), confirming that this peak corresponds to glycogen.

Next, we examined whether the accumulation of glycogen in the transition phase drives the phenotypes we observed in that phase, starting with the asymmetries in cell constriction and nucleoid positioning. Since the extent of these asymmetries changes with the OD during the transition phase (Fig. 1H,I), we co-cultured WT and ΔglgBXCAP strains to compare their phenotypes at precisely the same OD. Cells from each strain carried a different fluorescent protein, either mVenus or mSCFP3, to allow their differentiation by fluorescence microscopy (Fig. 3B). These strains also expressed HupA-mCherry for nucleoid visualization (Fig. 3B). We found that the offset of cell constriction and nucleoid positions from the cell center was minimal for both glycogen-free (ΔglgBXCAP) and glycogen-producing (WT) cells in exponential phase (Fig. 3C,D, Mix 1). While the offsets considerably increased in the transition phase for the WT cells, they did not for the mutant cells (Fig. 3C,D, Mix 1). We obtained similar results across biological replicates (Figs. EV3 and EV4) or when the strain-identifying proteins mVenus and mSCFP3 were swapped between the WT and ΔglgBXCAP backgrounds (Fig. 3C,D, Mix 2). The findings are consistent with glycogen production in the transition phase driving the observed nucleoid and cell constriction asymmetries.

Deletion of the glycogen biosynthesis operon also abrogated the intracellular rearrangement of ribosomes (RplA-mCherry) and protein probes (msfGFP) in the transition phase (Fig. 3E to compare with WT data in Fig. 2B). This was also true for the RNase E ΔMTS-mCherry marker (Fig. 3F to compare with WT data in Fig. 2D).

We reasoned that if the physical presence of glycogen at cell poles drives the transition-phase phenotypes, the severity of the phenotypes should correlate with the extent of glycogen accumulation across cells. To visualize glycogen in live cells, we built a glycogen biosensor and placed its synthesis under the control of the isopropyl β-D-1-thiogalactopyranoside (IPTG)-inducible promoter $P_{tac}$ inserted on the chromosome. Inspired by a previous eukaryotic study (Skurat et al, 2017), we generated a translational mGFPmut3 fusion to the N-terminus of the starch/glycogen binding domain (CBM20) of the human protein Stbd1. We found that the basal expression from the $P_{tac}$ promoter (i.e., without IPTG addition) was sufficient to visualize the glycogen sensor when cells reach the transition phase, likely due to metabolic regulation (Bren et al, 2013; Grossman et al, 1998). In the absence of glycogen production, the distribution of the glycogen sensor remained diffuse throughout

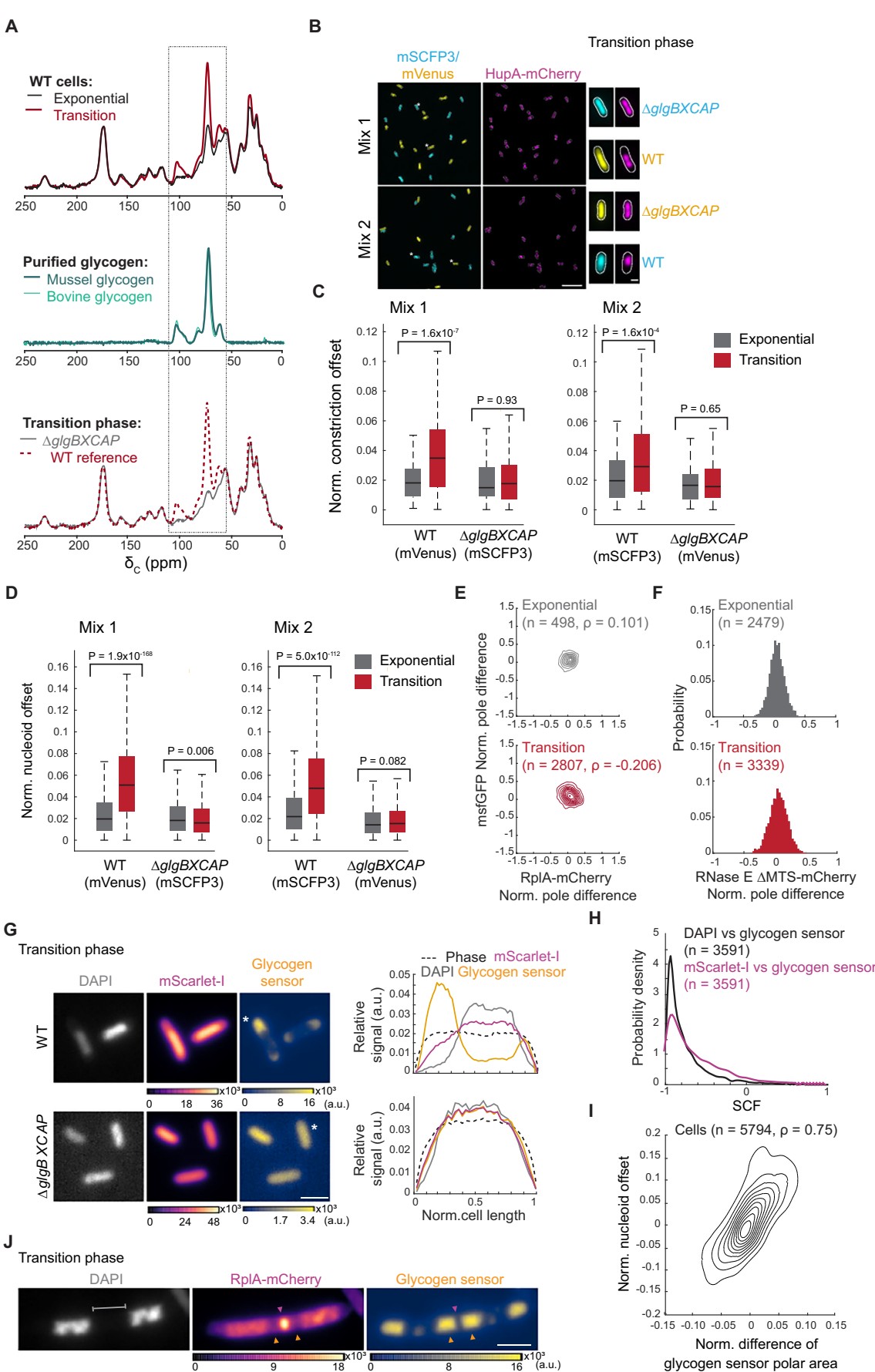

◀ **Figure 3.  Glycogen accumulation in the transition phase drives intracellular rearrangement.**

(A) Representative relative whole-cell $^{13}$C CPMAS spectral overlays of WT cell extracts (strain CJW2168) from cultures in exponential vs. transition phase (top), purified bovine and mussel glycogen (middle), and WT vs. Δ*glgBXCAP* (CJW7537) cell extracts from transition-phase cultures (bottom). The dashed rectangle marks the sugar carbon region of the spectrum. (B) Representative fields of view of fluorescence images showing cells from co-cultures in the transition phase. The top two fields of view show glycogen-producing cells (CJW7666) expressing mVenus and HupA-mCherry mixed with glycogen-devoid cells (CJW7667) expressing mSCFP3 and HupA-mCherry (Mix 1). The bottom fields of view illustrate glycogen-producing cells (CJW7665) expressing mSCFP3 and HupA-mCherry mixed with glycogen-devoid cells (CJW7668) expressing mVenus and HupA-mCherry (Mix 2). Scale bar: 10 µm. On the right side of the fields of view, the images of single cells (indicated by the single or double asterisks) show examples of the localization of mVenus (yellow), mSCFP3 (cyan), and HupA-mCherry (purple) for the relevant strain. Scale bar: 1 µm. (C) Boxplots showing the normalized (norm.) offset of the cell constriction site of the co-cultured WT and Δ*glgBXCAP* cells carrying either mVenus or mSCFP3 in exponential and transition phases. The horizontal lines in the boxes correspond to the medians, with the bottom and top of the boxes showing the 25th and 75th percentiles, respectively. The endpoints of the whiskers mark the minimum and maximum values within a range that excludes the outliers. Outlier values are defined as those more than 1.5 times the interquartile range away from the bottom or top of the boxes. The results shown correspond to the first replicate for Mix 1 and Mix 2. The results for two other biological replicates of each mix are shown in Fig. EV3. The indicated *P* values were obtained using a two-sided Wilcoxon rank sum test. (D) Same as (C) but for the normalized nucleoid position offset. The second and third biological replicates of Mix 1 and Mix 2 are shown in Fig. EV4. (E) Contour plots showing the normalized (Norm.) pole differences of msfGFP vs. the normalized pole differences of RplA-mCherry for exponential (top, OD = 0.065, *n* = 498) and transition-phase (bottom, OD = 1.2, *n* = 2807) cells of the glycogen-deficient strain Δ*glgBXCAP* (CJW7878). The normalized pole difference of a fluorescent signal corresponds to the difference in this signal between pole regions divided by the average signal intensity across the cell. The contour lines represent the 0.167, 0.33, 0.5, 0.66, 0.83, and 1 kernel density envelopes of the data. (F) Histograms of the normalized pole difference of RNase E ΔMTS-mCherry for cells (strain CJW7877) in exponential (OD = 0.07, *n* = 2479) or transition (OD = 0.82, *n* = 3339) phase. (G) Representative images of WT (CJW7606) and Δ*glgBXCAP* (CJW7607) cells in transition phase (OD 2.7) labeled with DAPI and expressing mScarlet-I and the glycogen sensor, along with the signal intensity profiles for the indicated (*) cells. Fluorescence intensities are indicated in arbitrary units (a.u.). Scale bar: 2 µm. (H) Distributions of SCF values of the glycogen sensor vs. DAPI or mScarlet-I for glycogen-producing cells (*n* = 3591, CJW7606 in transition phase. (I) Density contour plot showing the area difference of the glycogen sensor signal between the cell poles (normalized by the cell area) vs. the nucleoid position asymmetry in CJW7606 cells (*n* = 5794). The latter was calculated by determining the offset of the nucleoid mid-point from the cell center, normalized to the cell length. The contour lines represent the 0.10, 0.2, 0.3, 0.4, 0.50, 0.6, 0.7, 0.8, and 0.9 probability envelopes of the data. Spearman correlation coefficient (ρ) is 0.748 (*P* value = 0). (J) Example phase-contrast and fluorescence images of a DAPI-labeled CJW7606 cell carrying RplA-mCherry and the glycogen sensor. The cell was from a culture in transition phase following cephalexin treatment. Gray bracket indicates DNA-free region where glycogen accumulation (orange arrowheads) sandwiches a band of RplA-mCherry enrichment (purple arrowhead). Scale bar: 2 µm. Source data are available online for this figure.

the cytoplasm of Δ*glgBXCAP* cells, similar to a free fluorescent protein such as mScarlet-I (Fig. 3G). In contrast, glycogen accumulation in WT cells during the transition phase resulted in the accumulation of the glycogen sensor in nucleoid-free space, with its highest concentrations observed in regions from where mScarlet-I was depleted (Fig. 3G). The anti-correlation of the fluorescent glycogen biosensor with both the nucleoid marker (DAPI) and the cytoplasmic protein probe (mScarlet-I) across cells (*n* = 3591) was evident by their markedly negative values for the signal correlation factor (SCF) (Fig. 3H), a metric that assesses the correlation between two signals at pixel resolution within individual cells (Gray et al, 2019). Furthermore, the glycogen sensor tended to accumulate more at one pole than the other, and this asymmetric enrichment correlated strongly with the nucleoid offset from midcell (Fig. 3I, *n* = 5794 cells, Spearman's correlation ρ = 0.75), consistent with glycogen accumulation effectively "pushing" the nucleoid closer to the opposite pole.

Another line of support for the physical presence of glycogen driving the transition-phase phenotypes was provided by observations of cephalexin-treated cells expressing the glycogen sensor and fluorescently labeled ribosomes. Whenever we observed a band of ribosome enrichment in nucleoid-free space (as first illustrated in Fig. 2G), this region was flanked by glycogen accumulations (Fig. 3J). This observation was independently corroborated using 2-NBDG (2-(N-(7-nitrobenz-2-oxa-1,3-dia-zol-4-yl)amino)-2-deoxyglucose) (Yoshioka et al, 1996b), a glucose analog that is taken up by *E. coli* (Yoshioka et al, 1996b, 1996a) where it can be incorporated into the glycogen polymer (Zhu et al, 2020). Upon uptake, the analog is modified to a non-fluorescent form, resulting in a loss of fluorescence signal (Yoshioka et al, 1996a) and leading to a low signal-to-noise ratio. Nevertheless, the signal intensity was sufficient to reveal 2-NBDG accumulations flanking ribosome enrichment in nucleoid-free

space of cephalexin-treated cells producing glycogen during the transition phase (Fig. EV5).

## Preferential glycogen accumulation at the old cell pole through inheritance contributes to cellular asymmetries

We hypothesize that the preferential accumulation of glycogen at the old pole through inheritance over divisions (Boehm et al, 2016) causes the cellular asymmetries observed during the transition phase. To test this hypothesis, we performed microfluidic experiments on a mother machine-like device in which we visualized the localization of both the glycogen sensor and the nucleoids (HupA-mCherry) in single cells. For the analysis, we focused only on the "mother" cells (i.e., those located at the closed end of the trenches, see Fig. 4A), as they can be tracked over many generations (Wang et al, 2010). By connecting the microfluidic device to a liquid batch culture, the cells inside the trenches experienced the same changes in nutrient levels during the exponential-to-stationary phase transition as those in the batch culture (Fig. 4A) (Bakshi et al, 2021). At the start of imaging, exponentially growing mother cells already had a small but detectable glycogen sensor signal at the old pole, presumably because these cells inherited the old pole and its content at each division over many generations, resulting in a gradual build-up. This is illustrated in Fig. 4B where a mother cell was computationally extracted (i.e., the progenies were removed) from a trench to help visualize the relevant information (see Movie EV2 for the corresponding data from all the cells—mothers and progenies—in the selected trench). *E. coli* is known to synthesize glycogen at a low (basal) rate under exponential growing conditions (Wang et al, 2020). Under our experimental conditions, mother cells transitioned between exponential and stationary phases after 3 to 4 generations based on cell growth measurements (Fig. EV6). We found that mother cells increased the area of

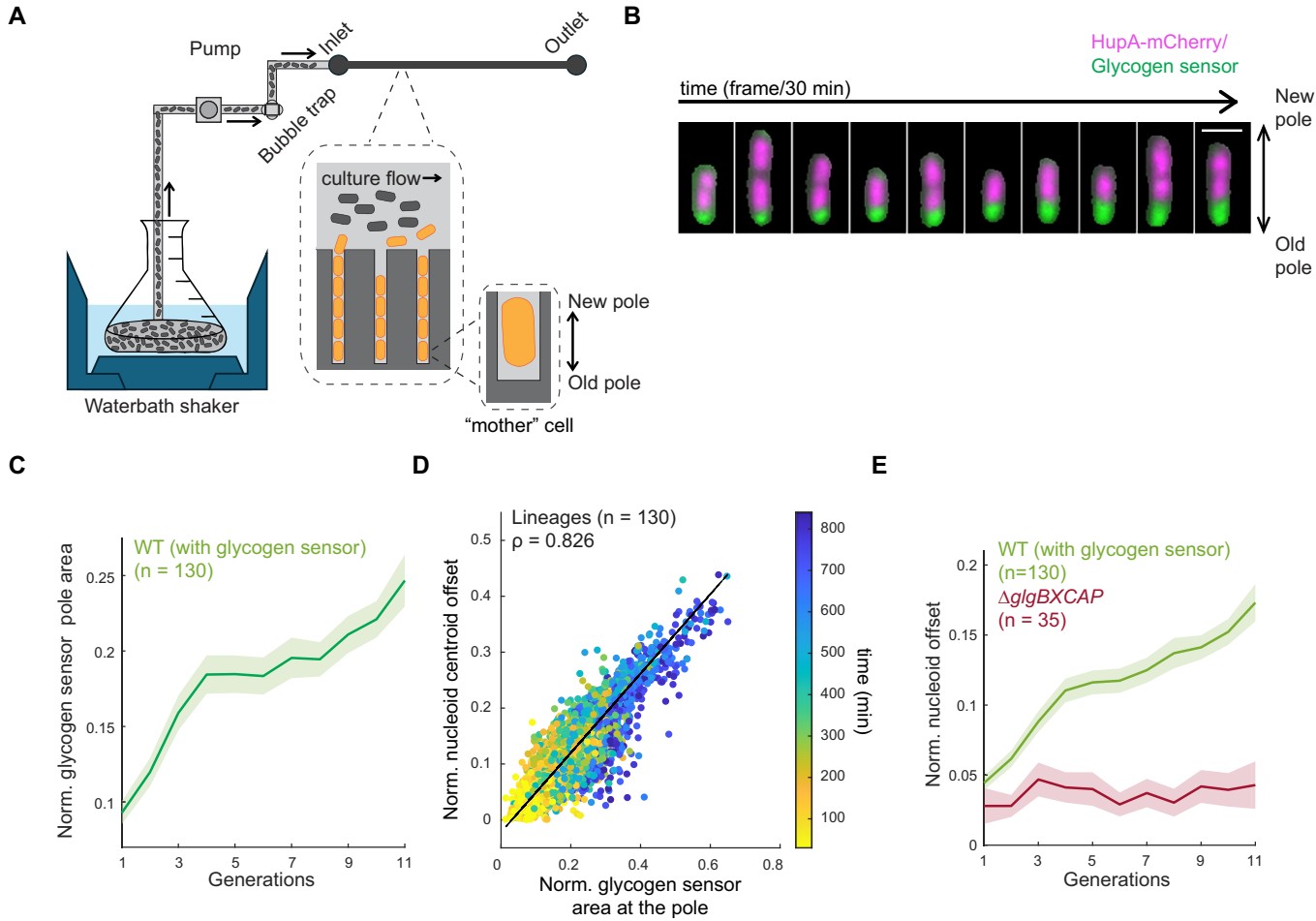

**Figure 4. Preferential glycogen accumulation at the old cell pole through inheritance contributes to the asymmetry in nucleoid positioning.**

(A) Schematic of the setup used in the microfluidic experiment. A waterbath shaker is used to grow a batch culture with controlled temperature (30 °C) and shaking (200 rpm). Culture is first passed through a custom-built bubble trap and then through the microfluidics chip (mother machine) using a peristaltic pump. The large inset (dotted lines) shows a zoomed-in depiction of the microfluidic chip, with the culture flowing through the feeding channels and cells growing in the narrow trenches. The smaller inset shows an enlargement of the mother cell. (B) Montage of a representative mother cell lineage of CJW7605 showing the overlay between the HupA-mCherry and glycogen sensor signals. For clarity purposes, the mother cell was computationally extracted using its segmentation mask. Scale bar 2 μm. (C) Plot showing the normalized glycogen sensor pole area over all observed generations. The solid line represents the mean across all tracked cell lineages ($n = 130$), while the shaded area indicates the 95% confidence interval. (D) Scatter plot of the correlation between the normalized nucleoid offset and the normalized glycogen sensor pole area for all tracked lineages ($n = 130$) across all analyzed time points (color bar). Spearman correlation coefficient (ρ) is 0.826 ($P$ value $= 0$). (E) Plot showing the normalized nucleoid offset as a function of cell generations for the glycogen WT strain (CJW7605, $n = 130$ lineages) and the glycogen-deficient (ΔglgBXCAP) strain (CJW7668, $n = 35$ lineages). The solid lines represent the means across all tracked cell lineages, while the shaded areas indicate the 95% confidence interval. Source data are available online for this figure.

glycogen sensor signal at the old pole at each generation (Fig. 4B,C; Movie EV2), consistent with a progressive accumulation of glycogen due to inheritance at division combined with de novo synthesis.

Furthermore, the nucleoid offset from midcell correlated with the area of glycogen sensor signal at the old pole, with both variables increasing over time (Fig. 4D). This is in agreement with the preferential accumulation of glycogen at the old pole, effectively "pushing" the nucleoid toward the new pole by mutual exclusion. As glycogen accumulation at the old pole increased at each generation (Fig. 4C), so did the nucleoid position offset (Fig. 4E). In contrast, the nucleoid position offset did not increase in the glycogen-devoid ΔglgBXCAP mother cells experiencing the same

conditions in separate channels (Fig. 4E). These observations are consistent with a causal relationship between glycogen accumulation and nucleoid position asymmetry. Consumption of glycogen during the stationary phase (Fung et al, 2013; Wang et al, 2021, 2020) explains the gradual decline in nucleoid position asymmetry observed with prolonged time in this phase (Figs. 1I and EV4).

## Nucleoid position asymmetry correlates with division asymmetry across cells

What promotes asymmetric division in cells in the transition phase? We considered two possibilities. First, large accumulations

of glycogen at the old pole may interfere with the pole-to-pole oscillatory behavior of MinD, which, through its binding partner MinC, is known to affect the position of the FtsZ cytokinetic ring on time average (Shih and Zheng, 2013; Lutkenhaus, 2008). Second, by effectively pushing the nucleoid toward the new pole, the accumulation of glycogen at the old pole may shift the division site toward the new pole through nucleoid occlusion, a fail-safe mechanism that prevents cell constriction over nucleoid regions, thereby avoiding chromosome scissoring events (Woldringh et al, 1990). We found the first possibility unlikely as MinD-GFP continued to oscillate from the edge of one pole to the other in transition-phase cells even when nucleoid asymmetry was evident (Movie EV3). The apparent lack of interference is presumably due to glycogen accumulations not adhering to the cytoplasmic membrane, allowing MinD-GFP to diffuse around them and interact with the membrane at the poles. To test the nucleoid occlusion hypothesis, we measured the positions of nucleoids (labeled with HupA-mCherry) and cell constriction sites relative to the cell center from snapshot images of transition-phase cells (where the pole identity is unknown and thus randomly assigned). We found these two variables to be highly correlated across the cell populations, as illustrated in Fig. 5A. The Spearman's correlation of $\rho$ was over 0.58 across five other replicates ($n > 278$ cells for each replicate). This is consistent with a mechanism in which the preferential enrichment of glycogen at the old pole promotes asymmetric division primarily by offsetting nucleoid positioning.

## Glycogen accumulation leads to a corresponding cell size increase

The preferential accumulation of glycogen at the old pole also affects the cytoplasmic organization of the cell by partially excluding ribosomes, proteins, and RNAs (Fig. 3). If division were symmetric and produced two daughter cells of equal size, the daughter cell that inherits the glycogen-rich pole would therefore receive less cytoplasmic content than its sibling. However, analysis of constricting cells in the transition phase revealed that the future daughter cells with more glycogen tended to be bigger than their future siblings with less glycogen (Fig. 5B), with a Spearman's correlation $\rho = 0.54$ ($P$ value $= 0$, $n = 365$ cells). While there was considerable noise in our measurements, a linear fit across the data suggested a near direct proportionality (i.e., slope ~1) between the difference in glycogen amount and the difference in cell size between future daughter cells (Fig. 5B). This cell size difference is unlikely to be due solely to a growth benefit associated with glycogen metabolism since it is generated in the mother cell before cell constriction is complete, i.e., when the cytoplasm (and thus, the metabolism) of the two future daughter cells is still shared (Fig. 5B). Thus, cells do not appear to "count" glycogen accumulations as cytoplasmic space and effectively create extra (i.e., bonus) space to accommodate the large amount of glycogen produced in transition phase.

This bonus-space hypothesis predicts that glycogen-deficient cells should, on average, be smaller than WT cells producing glycogen. To test this prediction, we compared the size of glycogen-producing (WT) and glycogen-deficient ($\Delta glgBXCAP$) cells grown in co-cultures (Fig. 3B). Consistent with our hypothesis, co-cultures of mVenus-expressing WT and mSCFP3-expressing $\Delta glgBXCAP$ cells revealed that glycogen-deficient cells were smaller than

glycogen-producing cells in the transition phase (Figs. 5C and EV7A, Mix 1). We obtained similar results when the fluorophores were swapped between the WT and $\Delta glgBXCAP$ backgrounds (Figs. 5C and EV7A, Mix 2), indicating that the cell size difference was independent of the type of fluorescent protein used to identify cells. Smaller but statistically significant differences in cell area were also found between WT and $\Delta glgBXCAP$ cells in the exponential phase (Figs. 5C and EV7A), which is consistent with basal glycogen accumulation during this growth phase (Wang et al, 2020).

We also found that the nucleoid signal occupies a considerably smaller fraction of the cytoplasm in glycogen-producing (WT) cells compared to the glycogen-deficient ($\Delta glgBXCAP$) mutant, particularly in transition-phase cells. This notable difference is evident in single-cell images (Fig. 3B, single-cell images) and is further demonstrated at the population level by the lower nucleocytoplasmic (NC) ratios (defined by the nucleoid area divided by the cell area) in WT cells relative to $\Delta glgBXCAP$ cells (Figs. 5D and EV7B). This disparity was primarily attributable to the differences in cell size between the two strains during the transition phase (Figs. 5C and EV7A), as the nucleoid areas themselves were comparatively more similar (Figs. 5E and EV7C).

## Glycogen condensates exclude fluorescent proteins in vitro

In the bacteriology literature, glycogen accumulations are often described as "granules" due to their round or oval shapes when visualized by electron microscopy (Preiss, 1984; Liu et al, 2021b; Alonso-Casajús et al, 2006; Preiss and Romeo, 1994). This term can give the impression of a solid. However, recent work in mammalian liver cells has reported that glycogen can undergo liquid–liquid phase separation in the cytosol (Liu et al, 2021a). Glycogen extracted from mouse livers has also been shown to undergo concentration-dependent phase separation in vitro when exposed to a crowding agent (Liu et al, 2021a). Given these results, we asked the following questions: Does glycogen form liquid condensates under in vitro conditions that mimic the *E. coli* cytoplasm? If so, can these glycogen condensates exclude proteins as they appear to do inside *E. coli* cells?

For our in vitro conditions, we focused on three aspects of the *E. coli* cytoplasm: glycogen concentration, ionic strength, and crowder concentration. Given that glycogen represents 0.75-1.2% of the cell's wet weight in stationary phase after growth in lysogeny broth (Neidhardt and Curtiss, 1996; Wang et al, 2019a), we assumed the glycogen concentration range to be between 8 and 13 g/L based on a cell density of 1.1 g/mL (Baldwin et al, 1995) (see Table EV1). We elected to use glycogen from mussels because of its commercial availability at a high (>99%) purity. We used a potassium phosphate buffer (pH = 7.0) that includes 85 mM NaCl, 250 mM KCl, 2.5 mM $MgCl_2$, and 0.1 mM $CaCl_2$ to reflect the ionic strength of the *E. coli* cytoplasm (Alatossava et al, 1985; Gangola and Rosen, 1987; Szatmári et al, 2020; Schultz et al, 1962). Hereafter, we refer to this solution as the "intracellular salt (IS) buffer". As a stand-in for cytoplasmic macromolecules, we used the synthetic biocompatible nonionic polymer polyethylene glycol (abbreviated as PEG or PEO depending on its molecular weight). PEG/PEO polymers are available in a broad range of molecular weights. Since size, rather than molecular weight, is the relevant variable for a crowding agent

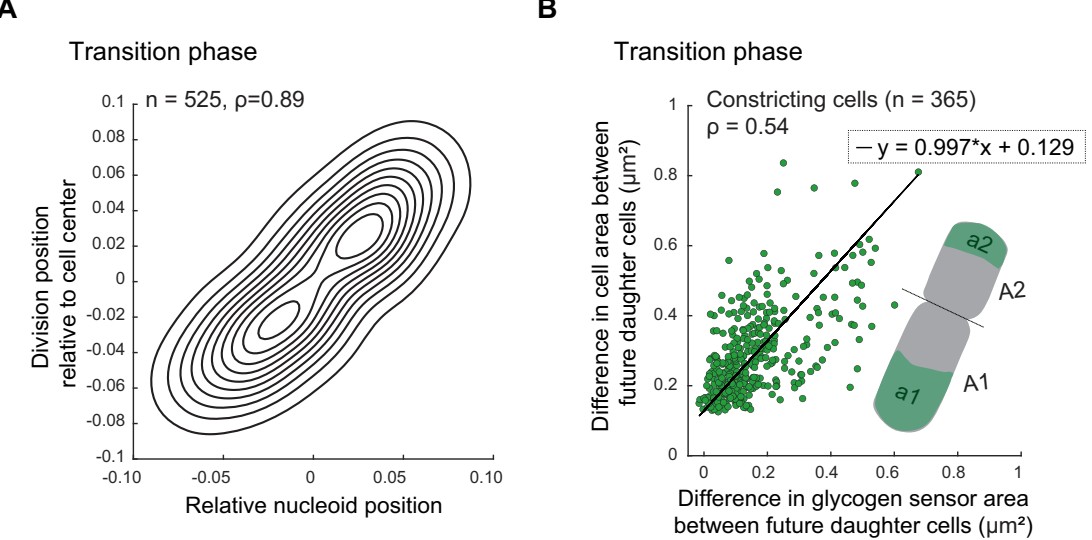

**A** Transition phase

n = 525, ρ=0.89

Division position relative to cell center (y-axis)
Relative nucleoid position (x-axis)

**B** Transition phase

Constricting cells (n = 365)
ρ = 0.54

— y = 0.997*x + 0.129

Difference in cell area between future daughter cells (µm²) (y-axis)
Difference in glycogen sensor area between future daughter cells (µm²) (x-axis)

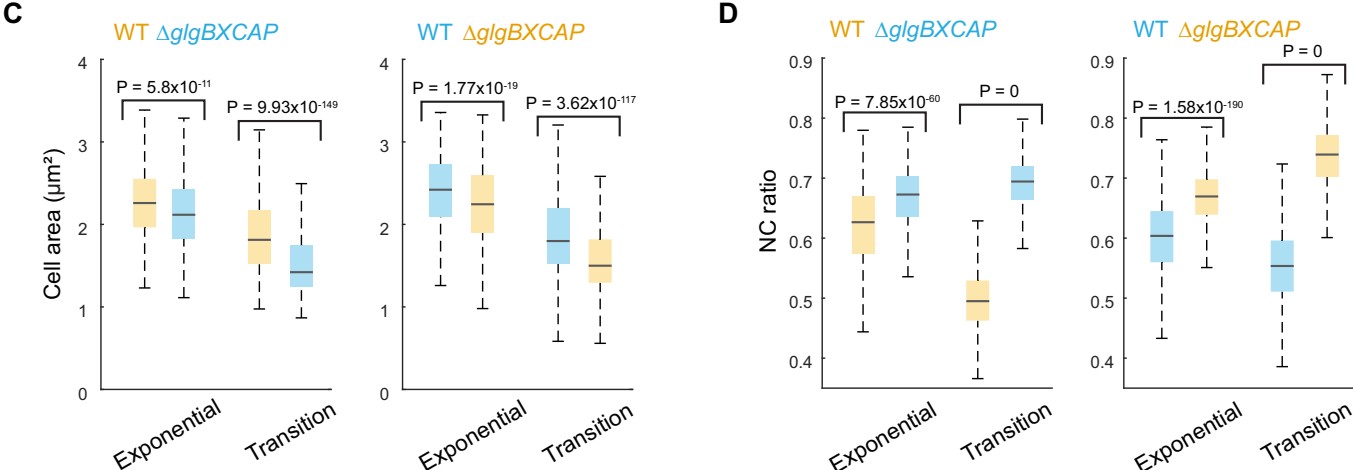

**C** WT Δ*glgBXCAP*    WT Δ*glgBXCAP*

Cell area (µm²)

P = 5.8x10⁻¹¹   P = 9.93x10⁻¹⁴⁹   P = 1.77x10⁻¹⁹   P = 3.62x10⁻¹¹⁷

Exponential   Transition

**D** WT Δ*glgBXCAP*    WT Δ*glgBXCAP*

NC ratio

P = 7.85x10⁻⁶⁰   P = 0   P = 1.58x10⁻¹⁹⁰   P = 0

Exponential   Transition

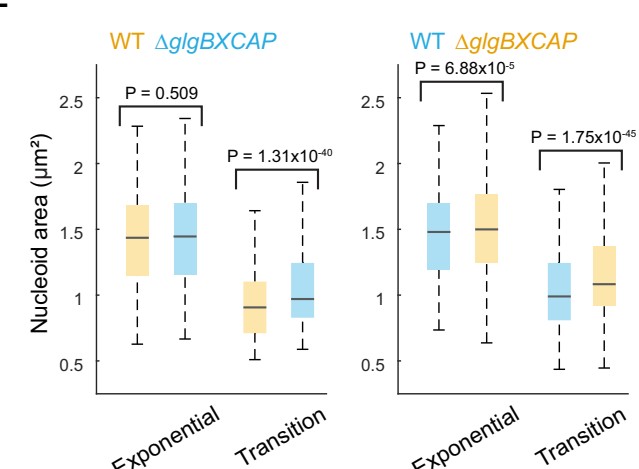

**E** WT Δ*glgBXCAP*    WT Δ*glgBXCAP*

Nucleoid area (µm²)

P = 0.509   P = 1.31x10⁻⁴⁰   P = 6.88x10⁻⁵   P = 1.75x10⁻⁴⁵

Exponential   Transition

◄ **Figure 5. Asymmetry in glycogen accumulation creates cell size differences and asymmetric cell divisions.**

(A) Density contour plot of the relative division position as a function of the nucleoid centroid position in dividing cells for transition-phase cells (strain CJW7666, $n = 525$). The values $+0.5$ and $-0.5$ represent the cell poles, while 0 corresponds to the cell center. Spearman correlation coefficient ($\rho$) for WT cells is 0.89 ($P$ value $= 0$). (B) Scatter plot showing the difference in cell area and the difference in the area occupied by the glycogen sensor between future daughter cells (strain CJW7606). Constricting cells ($n = 365$) were computationally divided into two future daughter cells based on the cell constriction position (illustrated in the inset). Spearman correlation coefficient ($\rho$) is 0.54 ($P$ value $= 1.28 \times 10^{-27}$). (C) Boxplots showing the area of exponential and transition-phase cells from WT (CJW7665 and CJW7666) and $\Delta glgBXCAP$ (CJW7667 and CJW7668) cultures carrying either mVenus (yellow) or mSCFP3 (cyan). The horizontal lines in the boxes correspond to the medians, with the bottom and top of the boxes showing the 25th and 75th percentiles, respectively. The endpoints of the whiskers mark the minimum and maximum values within a range that excludes the outliers. Outlier values are defined as those more than 1.5 times the interquartile range away from the bottom or top of the boxes. The indicated $P$ values were obtained using a two-sided Wilcoxon rank-sum test. The results shown correspond to the first replicate for Mix 1 and Mix 2. The results for two other biological replicates of each mix are shown in Fig. EV7. (D) Same as (C) but for the NC ratio. (E) Same as (C) but for the total nucleoid area. Source data are available online for this figure.

(Sharp, 2015; Asakura and Oosawa, 1958, 1954), we used dynamic light scattering to measure the hydrodynamic (Stokes) diameter of PEG and PEO polymers of different molecular weights in the IS buffer (Fig. 6A). Our results agree well with an empirical relationship determined in deionized water (pH = 5.5) (Devanand and Selser, 1991) (Fig. 6A, line), despite the differences in pH and ion concentrations of the solvents used.

We first chose 3 kDa PEG (~3 nm of diameter) to mimic proteins, which are the most abundant macromolecular crowders in the *E. coli* cytoplasm (Neidhardt and Curtiss, 1996) with diameters ranging between 2 and 6 nm (Erickson, 2009; Lukatsky and Shakhnovich, 2008; Hink et al, 2000). We found that glycogen (10 g/L), visualized by fluorescence microscopy using ConA-FITC labeling (Lvov et al, 1996; Becker et al, 1976), transitioned from a homogeneously mixed state in the solution to forming droplets as the concentration of 3 kDa PEG increased from ~9 to ~24 mM (Fig. 6B). Consistent with the liver study (Liu et al, 2021a), these droplets exhibited liquid-like behaviors, demonstrating fusion events on the minute time scale (Fig. 6C; Movie EV4). These fusion events resulted in the coalescence of small condensates into larger ones at longer time scales (e.g., 30 min, Fig. 6D), which is consistent with a liquid–liquid phase separation. Note that these images were captured near the glass surface to keep droplets in focus and facilitate their visualization. In this context, some fusion events resulted in the formation of oval-shaped droplets that did not immediately become spherical. This phenomenon is likely due to the interaction between the droplets and the glass surface, as the droplets within the liquid column remained spherical after fusion and only became less spherical after they settled on the glass surface (Movie EV5). We also found that at higher 3 kDa PEG concentrations (~30 mM), glycogen underwent a second phase transition, in which the droplets collapsed into amorphous aggregates (Fig. 6B). Time-lapse imaging of these aggregates showed no evidence of fusion upon collision (e.g., Movie EV6), consistent with a more solid-like form.

In addition, we confirmed that the phase separation of glycogen is dependent on its concentration. As shown in a phase diagram (Fig. 6E), higher concentrations of glycogen required a lower concentration of crowder to form liquid droplets. By using PEG/PEO of higher molecular weights, we also demonstrated that the formation of glycogen droplets is sensitive to the size of the crowder (Fig. 6F). For instance, at least 19 mM of 3 kDa PEG was required for liquid droplets to form, while the same could be achieved with only 150 µM of 100 kDa PEO or 10 µM of 1 MDa PEO. The two larger crowders (100 kDa and 1 MDa) were chosen

based on their estimated diameters, which are ~30 and ~70 nm, respectively (Fig. 6A). These sizes closely resemble those of ribosomes in their free and polysome forms (Nilsson et al, 1997; Brandt et al, 2009).

As both proteins and ribosomes (mostly assembled into polysomes) are excluded from glycogen in vivo (Fig. 3G–J), we hypothesized that the combined effect of their cellular concentrations could lead to the phase separation of glycogen into droplets. Consistent with this hypothesis, combining the stand-in crowders (10 mM of 3 kDa PEG, 25 µM of 100 kDa PEO, and 5 µM of 1 MDa PEO) at biologically relevant concentrations for proteins, ribosomes, and polysomes (see Tables EV1 and EV2) induced the formation of glycogen droplets in vitro (Fig. 6G). In contrast, none of these crowders alone triggered droplet formation at the same concentrations (Fig. 6G). Note that, for the 1 MDa PEO crowder, the concentrations required to drive droplet formation are above its "overlap" concentration (self-crowding) (see "Methods") and thus our estimated diameter for this crowder (Fig. 6A) is likely overestimated. Regardless of the crowder size used, glycogen condensation into droplets was found to be reversible: diluting the crowders after the droplets formed led to their rapid (< 1 min) disappearance (Fig. 6H).

To test whether glycogen condensation into droplets could account for the exclusion of proteins observed in vivo, we added GFP to the solutions and analyzed the spatial distribution of fluorescence intensity relative to the phase-contrast signal of the glycogen droplets. We found that glycogen condensates at the glass surface partially excluded the fluorescent proteins (Fig. 6I). This was also observed by performing z-stack imaging of the glycogen condensates suspended in the liquid column, in which protein exclusion remains observable despite the added background from the contribution of GFP molecules in surrounding focal planes (Movie EV7). This protein exclusion was observed in all tested combinations of PEG/PEO crowder sizes (2.7–74 nm) and concentrations that led to glycogen condensate formation (Table EV3).

## Glycogen condensates inside cells are as soft as the rest of the cytoplasm, unlike protein aggregates

Our in vitro data, together with estimates of intracellular macromolecule concentration (Tables EV1 and EV2) and the shape of so-called glycogen "granules" in electron micrographs (Preiss, 1984; Alonso-Casajús et al, 2006), are more consistent with glycogen forming droplets (liquid-like condensates) over solid

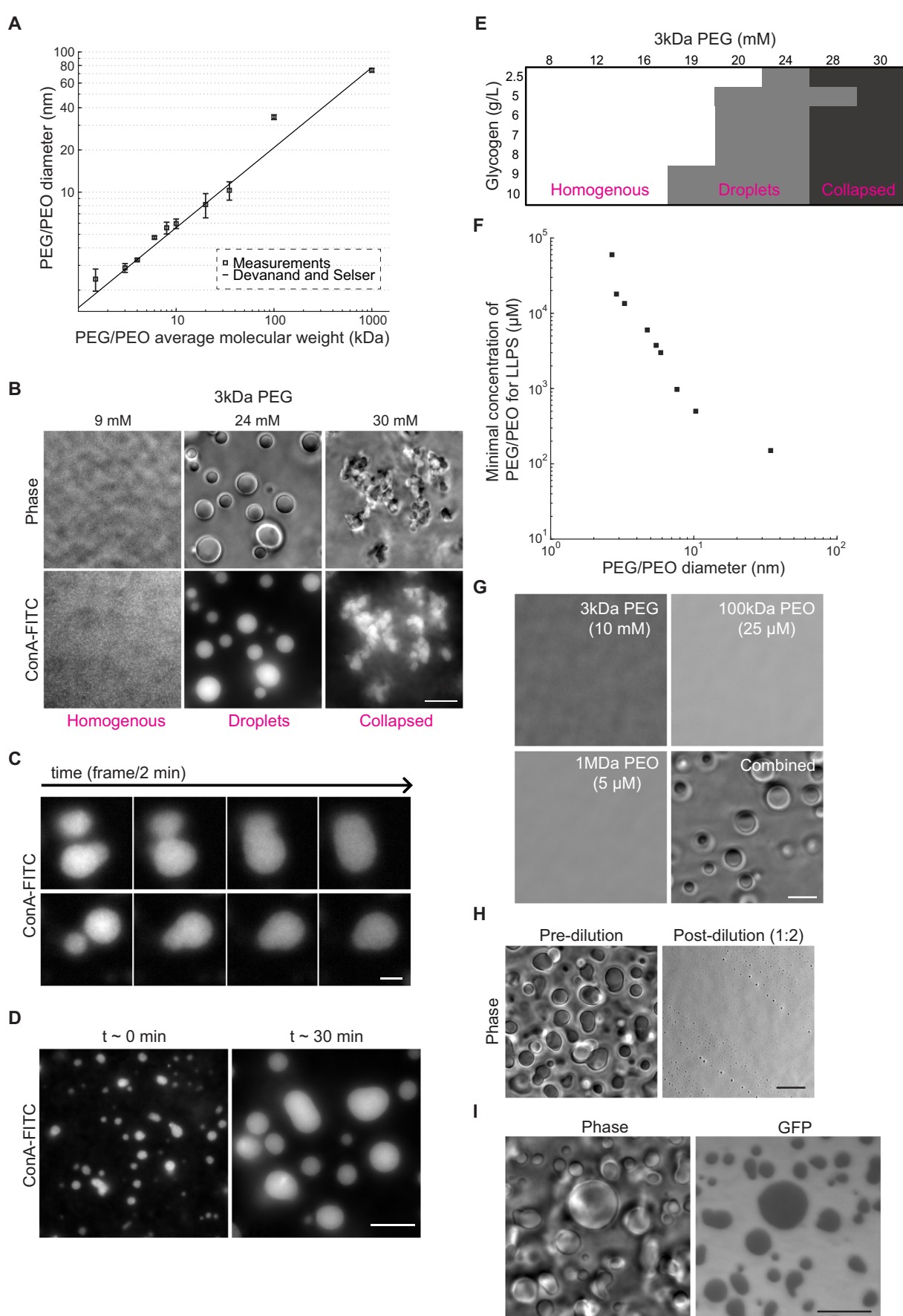

**Figure 6. Mimicking intracellular conditions of crowding and ionic strength induces phase transition of glycogen into liquid condensates that partially exclude GFP.**

(A) Plot showing dynamic light scattering (DLS) measurements of the particle diameter size for PEG and PEO polymers of different molecular weights in IS buffer. Samples were measured in triplicate (technical replicates) for 300 s at 25 °C with a 90-degree detection angle. The line indicates the empirical relation (hydrodynamic diameter $(\sigma_{PEG/PEO}) = 0.029*MW_{PEG/PEO}^{(0.571\pm0.009)})$ determined by Devanand and Selser (1991). Error bars represent the standard error of the mean. (B) Phase-contrast and ConA-FITC fluorescence images of representative fields of view showing a homogeneous distribution of glycogen (left), glycogen droplets (center), and collapsed glycogen aggregates (right) obtained with the indicated concentrations of 3 kDa PEG. All samples were made using the IS buffer and contained 9 g/L of glycogen. Scale bar: 10 μm. (C) Example montages of glycogen condensates labeled with ConA-FITC undergoing fusion events over time. The samples contained 9 g/L of glycogen and 8 kDa PEG (3 mM) in IS buffer. After mixing with the crowder, the samples were immediately spotted in the glass-bottom dish and imaged for a total of 7 min. Scale bar: 3 μm. (D) Fluorescence images of ConA-FITC-labeled glycogen condensates imaged at $t \sim 3$ min and $t \sim 30$ min after the addition of 3 kDa PEG (24 mM). The samples contained 9 g/L of glycogen in IS buffer. The Samples were imaged immediately after spotting in the glass-bottom dish. Scale bar: 10 μm. (E) Phase diagram of glycogen phases as a function of glycogen and 3 kDa PEG concentrations. Samples were imaged immediately after mixing with the appropriate amount of crowder. Each data point was evaluated two to three times in independent experiments to determine consensus classification based on visual inspection. (F) Plot showing the minimal concentration required to drive droplet formation as a function of the average Stokes diameters for all tested PEG/PEO crowders. Phase separation was determined by assessing the formation of droplets using phase-contrast imaging. The average diameter for each PEG/PEO crowder corresponds to the measurements shown in (A). (G) Phase-contrast images of representative fields of view of glycogen (10 g/L) in IS buffer mixed with the indicated concentration and molecular weight of PEG or PEO (separately or combined). Scale bar: 10 μm. (H) Phase-contrast images of glycogen (10 g/L) and 3 kDa PEG (20 mM) mixtures in IS buffer before and after a two-fold dilution. For both conditions, imaging was performed within 1 min after mixing. Scale bar: 10 μm. (I) Phase-contrast and GFP images of a representative field of view of glycogen (10 g/L) in IS buffer mixed with GFP (15 μM) and 3 kDa PEG (20 mM). Scale bar: 10 μm. Source data are available online for this figure.

aggregates inside cells. Testing this hypothesis is, however, not trivial. The small size of bacterial cells makes it impractical to examine liquid-like behaviors such as droplet fusions in vivo, as is often done in eukaryotic cells (Hoang et al, 2024; Liu et al, 2021a; Alberti et al, 2019, 2018). Assays based on fluorescence recovery after photobleaching (FRAP) and single-molecule tracking have been implemented to probe the liquid-like properties of condensate-forming proteins inside live bacterial cells (Alberti et al, 2019; Wang et al, 2019b). However, while proteins can be easily covalently tagged to a fluorophore through genetic engineering, this is not the case for a polysaccharide such as glycogen. FRAP experiments using the glycogen sensor showed that the fluorescence of the labeled glycogen region recovered from photobleaching (Fig. EV8A,B), with half-maximum recovery times of $11.6 \pm 2.3$ s (mean ± standard deviation, $n = 70$ cells, Fig. EV8C). These values are consistent with FRAP analyses of liquid glycogen condensates in mammalian liver cells (Liu et al, 2021a). However, our FRAP measurements are inconclusive due to the non-covalent nature of the interaction between the fluorescent biosensor and glycogen. The observed FRAP dynamics may reflect not only the motion of labeled glycogen molecules but also the (unknown) binding/unbinding kinetics of the fluorescent biosensor to/from glycogen. The use of the fluorescent glucose analog 2-NBDG, which covalently incorporates into glycogen (Fig. EV5), is unfortunately not suitable for FRAP experiments. This analog is modified to a non-fluorescent form upon cell uptake (Yoshioka et al, 1996b, 1996a), leading to a low intracellular signal-to-noise ratio (Fig. EV5).

Given these caveats, we turned to atomic force microscopy (AFM) to determine the material properties of glycogen condensates inside *E. coli* cells at the nanometric scale. Our first approach leveraged AFM-based subsurface imaging, which has been used to visualize nuclei and cytoskeletal structures in eukaryotic cells (Roduit et al, 2009; Guerrero et al, 2019). In this technique, the AFM tip indents the cell without puncturing the membranes, and the indentation is deep enough to detect subsurface structures or organelles through variations in their mechanical resistance to the deformation. We adapted this method to bacterial cells to generate two-dimensional (2D) depth-resolved stiffness maps of transition-phase cells (CJW7605) with glycogen

condensates labeled with our green fluorescent glycogen sensor (Fig. 7A). For comparison, we imaged two other strains: a glycogen-free ($\Delta glgBXCAP$) strain (CJW7668) expressing the yellow fluorescent protein mVenus and a strain (CJW7798) overproducing a blue fluorescence protein, which results in the formation of protein aggregates. The location of these protein aggregates could be easily identified in fluorescence images by the region depleted in mTagBFP2 signal (white arrow, Fig. 7B), as previously shown (Papagiannakis et al, 2025). All three strains were grown in co-culture to ensure that they experienced the same environment during the transition phase, as cell density increases when cultures exit exponential growth (Shi et al, 2021). We used correlated fluorescence microscopy to distinguish between cell types (Fig. 7B) and to identify the location of glycogen condensates and protein aggregates within cells (arrows, Fig. 7B). For all analyzed cells, the preservation of membrane integrity (i.e., no abrupt change in AFM cantilever deformations) was confirmed by examining the force-distance curves (see examples in Fig. EV9), using a previously validated method (Janel et al, 2019). The 2D stiffness maps revealed that protein aggregates are stiffer than the rest of the cytoplasm (black arrow, Fig. 7B). In contrast, the stiffness values were uniformly low across the 2D maps of cells with glycogen condensates, similar to glycogen-free cells (Fig. 7B).

In addition, we performed an analysis known as "stiffness tomography", which spatially maps changes in stiffness along the indentation depth (Roduit et al, 2009). Comparative stiffness tomography revealed that intracellular regions with protein aggregates became increasingly stiffer with indentation depth, unlike surrounding areas. This indicates that protein aggregates are relatively stiff, consistent with a solid-like state. In contrast, the same type of measurements for regions with glycogen condensates showed a relatively uniform stiffness distribution along the indentation depth, comparable to glycogen-free regions (Fig. 7C).

Under these conditions, the average cell height across strains ranged from 600 to 700 nm (Fig. EV10), as determined by measuring the distance between the glass surface and the AFM tip's contact point with the cell surface. Using subsurface imaging, we measured to a depth of 400 nm beneath the cell surface (Fig. EV9B–D). To examine the material properties of glycogen condensates and protein aggregates in a more direct way, we used a

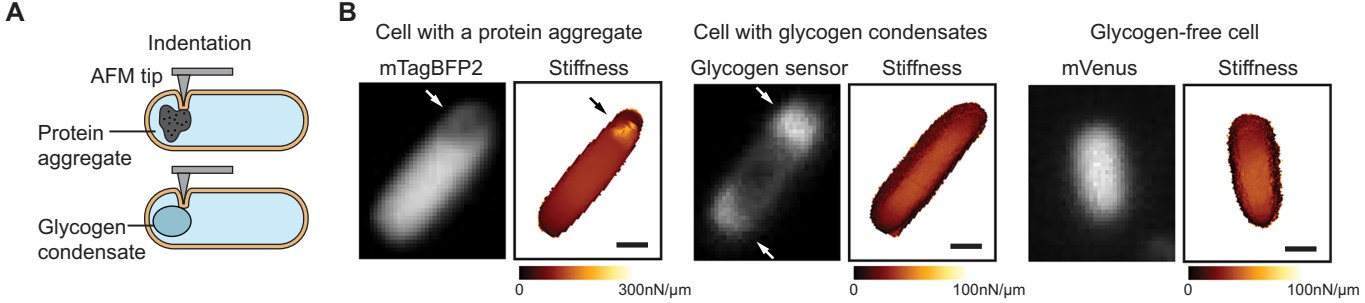

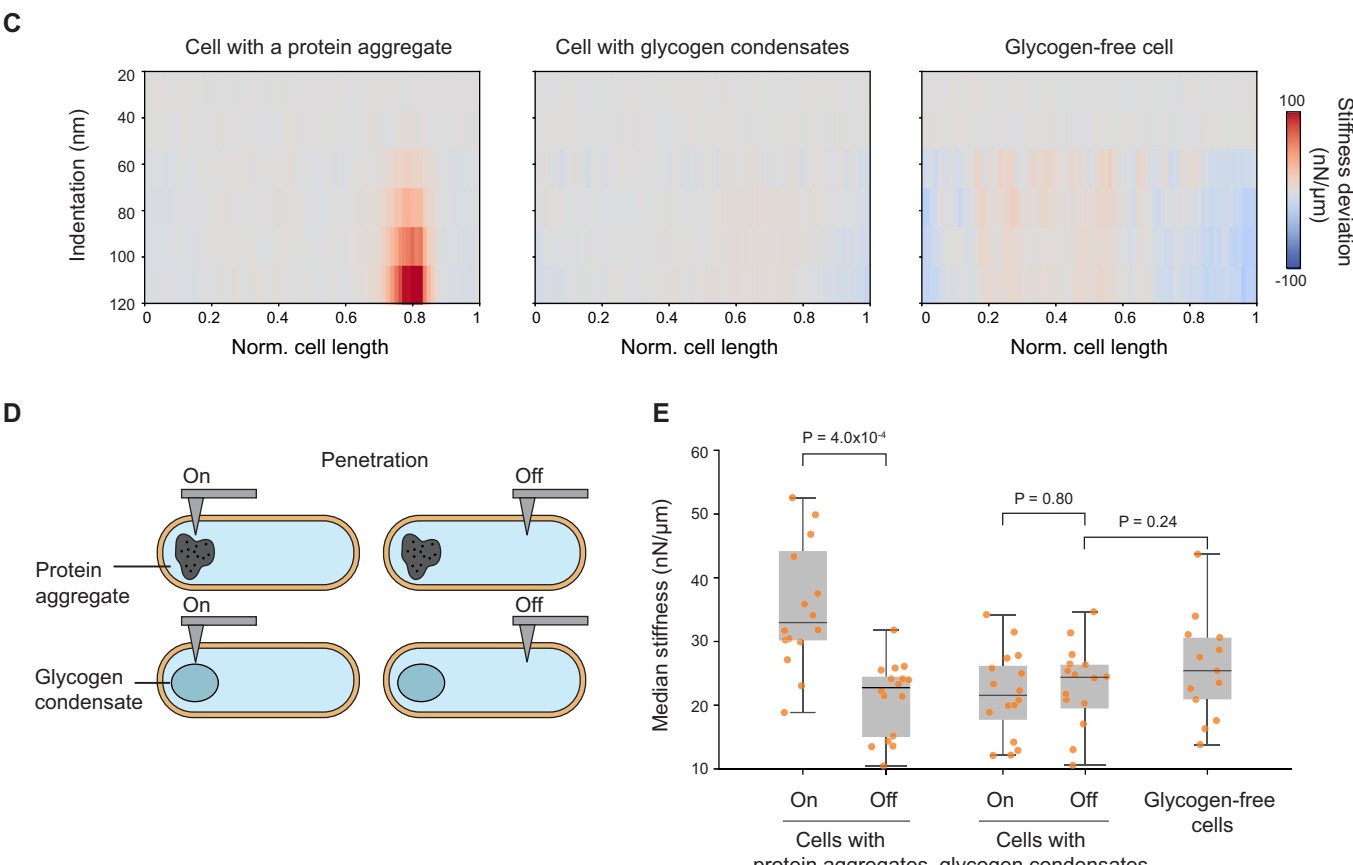

**Figure 7. Intracellular regions containing glycogen condensates are softer than those with protein aggregates and are comparable in stiffness to other cytoplasmic regions.**

AFM experiments were performed on cells from three strains co-cultured to the transition phase (ODs ranging between 1.5 and 1.7): CJW7798 overexpressing mTagBFP2 and accumulating protein aggregates at a cell pole, CJW7605 expressing the glycogen sensor to visualize glycogen condensates, and CJW7668 (ΔglgBXCAP) carrying free mVenus. (**A**) Schematic of indentation-based subsurface AFM imaging with a cell containing a protein aggregate or a glycogen condensate. (**B**) Two-dimensional depth-resolved stiffness maps (right) with the correlated fluorescence images (left) showing stiffness distributions for a representative cell with a protein aggregate, a cell with accumulations of the fluorescent glycogen sensor (white arrows), or a cell without glycogen. Scale bar: 500 nm. (**C**) Stiffness tomography showing the distribution of stiffness deviations along the indentation depth of cells with either a protein aggregate, glycogen condensates, or without glycogen. The stiffness deviation is the difference in stiffness between the stiffness value at each location and the median stiffness value across the cell length at the same indentation depth. Gray means no deviation. (**D**) Schematic illustrating penetration-based stiffness measurements at targeted regions containing protein aggregates or glycogen condensates (labeled "On") and at regions away from them (labeled "Off"). (**E**) Plot showing the median stiffness of "On" and "Off" intracellular regions of either cells ($n = 15$) containing protein aggregates or cells ($n = 16$) with glycogen condensates, compared to cytoplasmic regions of glycogen-free cells ($n = 13$). In the boxplots, the central line in the box indicates the median (50th percentile), the bounds of the box represent the 25th and 75th percentiles (interquartile range), and the whiskers extend to the minimum and maximum values of a range that excludes the outliers. Outlier values are defined as those more than 1.5 times the interquartile range away from the bottom or top of the boxes. Each dot represents the median stiffness of an individual cell obtained from 64 puncturing events (technical replicates). Data were collected over three independent experiments (biological replicates) with an average of about five cells per replicate. Source data are available online for this figure.

sharper AFM tip to penetrate through the cell membranes and directly interact with the intracellular content (Fig. 7D). Such an AFM-based approach has been used to measure the stiffness of nuclei inside eukaryotic cells (Liu et al, 2014; Oak et al, 2025). Here again, we used correlated fluorescence microscopy to position the AFM tip on and off the target sites (Fig. 7D). First, we confirmed successful membrane penetrations indicated by an abrupt change in AFM cantilever deformations (see curve examples in Fig. EV11A–C), as prior studies have demonstrated (Penedo et al, 2021; Del Valle et al, 2020). For these cell puncturing experiments, we used a force of 10 nN as a setpoint, which resulted in an average penetration depth of about 500 nm (i.e., ~¾ of the cell depth) across intracellular regions with or without glycogen condensates (Fig. EV11D). In areas containing protein aggregates, the penetration depth was reduced to ~400 nm due to the stiffness of these intracellular structures, which resist deformation more than the surrounding cytoplasm (Fig. EV11D). To calculate the stiffness of each intracellular region, we determined the slope of the force-distance curves after the membrane puncture point and plotted the median stiffness values from multiple puncture events ($n = 64$) in the target regions ("On") or control regions ("Off") (Fig. EV11A–C, see also "Methods"). This analysis demonstrated that intracellular regions with protein aggregates ("On") are considerably stiffer than other areas of the cytoplasm ("Off") (Fig. 7E). In contrast, regions with ("On") or without ("Off") glycogen condensates within the same cells exhibited similarly low stiffness values (Fig. 7E). Collectively, our results suggest that the stiffness of glycogen condensates is considerably lower than that of protein aggregates and is comparable to that of other cytoplasmic regions. These findings argue that glycogen forms condensates that are more liquid-like than solid-like inside cells, though we cannot exclude the possibility of a gel-like state.

## Discussion

Our quantitative analysis of the transition phase at the subcellular and single-cell level reveals a picture of *E. coli* cell biology that departs from the textbook view in several ways.

The first difference relates to division. Asymmetric divisions are typically thought of as exceptions in the bacterial world, restricted to a small subset of bacteria that undergo developmental programs that produce either two functionally divergent daughter cells at each cell cycle (e.g., *Caulobacter crescentus*) or an endospore in response to starvation (e.g., *Bacillus subtilis*). In fact, *E. coli* is often used as the quintessential example of a symmetrically dividing bacterium. However, its division has primarily been examined during exponential growth. Our study shows that asymmetric division and production of daughter cells of unequal sizes become common in *E. coli* when the population exists in the exponential phase and enters the transition phase under our experimental conditions (Figs. 1H, 3C, and 5A,B), likely contributing to the documented increase in cell size variability in the stationary phase (Bakshi et al, 2021).

We propose that these asymmetric divisions occur because of the propensity of glycogen to spontaneously phase separate into condensates in crowded environments (Fig. 6) (Liu et al, 2021a). In vitro, glycogen condensates can grow to very large sizes through

fusion (Fig. 6). Inside cells, their sizes are likely restricted by spatial constraints (cellular and nucleoid boundaries) and out-of-equilibrium thermodynamics (glycogen synthesis/degradation, cell growth/division). In principle, the accumulation of large glycogen condensates could occur in any DNA-free regions, as observed in polyploid filamentous (cell division-arrested) cells (Fig. 3J). However, our data suggest that in untreated cells, cell division results in the inheritance of glycogen condensates, which, together with de novo synthesis, results in a gradual accumulation of glycogen at the old cell pole, effectively "pushing" the nucleoids closer to the new pole (Fig. 4). In fact, such nucleoid pushing effect can be artificially exacerbated in exponential phase through mutation of the carbon storage regulator gene *csrA*, which results in massive glycogen overproduction (Boehm et al, 2016). We propose that the nucleoid position offset generated by glycogen condensates promotes off-center division through nucleoid occlusion (Fig. 5A). In stationary phase, this cellular asymmetry decreases and nucleoid positioning becomes more variable (Figs. 1I and EV4) due to glycogen consumption (Wang et al, 2021; Fung et al, 2013; Wang et al, 2020), allowing the nucleoid to diffuse more freely through the cytoplasm. Although our proposed mechanism is backed by compelling evidence, definitive proof for the role of phase separation will necessitate perturbations that specifically disrupt the mechanism of glycogen condensation without abolishing the synthesis of this polymer. Phase separation can be disrupted through mutations when proteins or nucleic acids are involved (Qian et al, 2022; Alberti et al, 2025; Sprunger et al, 2025). Achieving this feat for carbohydrates remains a challenge.

Cells in the transition phase also diverge from the traditional picture with respect to their spatial organization of ribosomes/polysomes, proteins, and RNAs. In transition phase, these cytoplasmic components adopt a more asymmetric distribution (Figs. 2 and EV1), which is largely caused by glycogen production (Figs. 3–5). Our in vitro experiments suggest that glycogen does not have to assemble into a solid (i.e., a granule) to exclude protein probes, as liquid condensates achieve similar results in vitro (Fig. 6I). In fact, based on our calculations (Tables EV1 and EV2) and in vitro experiments (Fig. 6E–G), the physiological range of intracellular concentrations of proteins, ribosomes, and polysomes is compatible with the combined PEG/PEO concentrations needed for glycogen to phase separate into liquid condensates, but not to collapse into a solid-state. Collapsing glycogen into aggregates requires crowder concentrations that well exceed the macromolecular concentration expected inside cells (Fig. 6E; Tables EV1 and EV2). Furthermore, unlike the amorphous glycogen aggregates, the liquid condensates formed in vitro are morphologically consistent with the round and oval shapes of glycogen bodies observed in electron micrographs of starved cells (Alonso-Casajús et al, 2006). Consistent with our results, a preprinted study has shown that concentrating macromolecules in frog egg extracts by ≥1.4-fold demixes the cytoplasm into two liquid phases: a glycogen-enriched phase and a ribosome-enriched phase (Pelletier et al, 2021). In *E. coli*, and presumably other glycogen-producing bacteria, this cytoplasmic demixing may be even more likely, as the bacterial cytoplasm is thought to be more crowded than the eukaryotic cytosol based on fluorescent protein diffusion measurements (Swaminathan et al, 1997; Elowitz et al, 1999; Konopka et al, 2009; Potma et al, 2001). Furthermore, our AFM measurements

suggest that the mechanical properties of glycogen accumulations within cells align more closely with (liquid-like) cytoplasmic material than with solid aggregates.

Glycogen is well-known to provide a way for cells to store a large amount of glucose without dramatically increasing the osmolarity of their cytoplasm, as a branched glycogen polymer contains thousands of glucose residues (Bezborodkina et al, 2018). Whether its phase separation into condensates is associated with beneficial or detrimental implications for the cell is an interesting question that warrants further exploration. For instance, since glycogen has viscogenic properties (Persson et al, 2020), compartmentalization into condensates might prevent high levels of glycogen from increasing the cytoplasmic viscosity, which could decrease macromolecular diffusion to deleterious levels. Furthermore, if glycogen did not phase separate (i.e., remained soluble and homogeneously distributed in the cytoplasm), the distribution of glucose reserves between daughter cells would be equal. In contrast, our data suggest that phase separation leads to an asymmetric distribution that increases with each division through additive glycogen accumulation at the old pole (Fig. 4). This unequal distribution endows a fraction of the cell population with larger energy reserves, which may increase the survival probability of the species under prolonged periods of duress. Glycogen has indeed been reported to facilitate the adaptation of environmental, phototrophic, and pathogenic bacteria to starvation and other stresses (Bourassa and Camilli, 2009; Gründel et al, 2012; Klotz et al, 2016; Klotz and Forchhammer, 2017; Sekar et al, 2020; Wang et al, 2020). As for the partitioning of other macromolecules, their exclusion from glycogen condensates does not appear to affect their distribution between daughter cells. This is because division becomes asymmetric in the presence of glycogen condensates; the larger the glycogen accumulation, the more asymmetric the division (Fig. 5B), minimizing any potential detrimental disparity in cytoplasm partitioning between daughter cells. In effect, the cells appear not to "count" the space occupied by the glycogen condensates (Fig. 5B).

The relevance of our work may extend beyond *E. coli*, as glycogen metabolism is widespread across bacteria (Almagro et al, 2015; Henrissat et al, 2002; Wang and Wise, 2011; Preiss and Romeo, 1990; Wang et al, 2019a). Induction of glycogen synthesis is a common response when a nutritional element such as nitrogen, sulfur, or phosphate becomes limiting (Mulder et al, 1962; Sigal et al, 1964; Madsen, 1963; Zevenhuizen, 1966; Holme et al, 1956, 1957; Strange et al, 1961). Given that bacteria often live in nutrient-fluctuating environments, our findings suggest that the cellular asymmetries in division and macromolecule distribution may be more prevalent in the bacterial world than currently appreciated. In addition, phase separation may be relevant to other bacterial storage polymers besides glycogen. Polyphosphate, an energy-rich anionic polymer, is a good candidate. It accumulates into intracellular bodies, often referred to as "granules", in various bacteria (Albi and Serrano, 2016). Time-lapse experiments on starved *Pseudomonas aeruginosa* cells have shown that these so-called granules decrease in number while increasing in size during de novo biogenesis, consistent with fusion events and liquid behavior (Racki et al, 2017). Future exploration may shed light on the prevalence, role, and implications of phase separation across storage polymers in different bacteria.

## Methods

### Reagents and tools table

| Reagent/resource | Reference/source | Identifier or catalog number |
|---|---|---|
| **Experimental models** | | |
| F-lambda- ilvG- rfb-50 rph-1 | Jensen, 1993 | MG1655 |
| MG1655 *rplA::rplA-gfp* | Gray et al, 2019 | CJW4677 |
| MG1655 *rne::rneΔMTS-mcherry* | This work | CJW5685 |
| MG1655/pEB2-mScarlet-I | AddGene; deposited by Dr. Philippe Cluzel | 104007 |
| MG1655 *rplA::rplA-msfGFP* | Gray et al, 2019 | CJW7020 |
| MG1655 *rpsB::rpsB-msfGFP* | Gray et al, 2019 | CJW7021 |
| MG1655 *attB::PproC-msfGFP-FRT-kan-FRT* | This work | CJW7083 |
| MG1655 *attB::PproC-msfGFP* | This work | CJW7275 |
| MG1655 *rplA::rplA-mCherry* | Gray et al, 2019 | CJW7324 |
| MG1655 *rplA::rplA-mCherry attB::PproC-msfGFP* | This work | CJW7325 |
| MG1655 *attB::PproC-msfGFP hupA::hupA-mCherry* | This work | CJW7326 |
| MG1655 *rplA::rplA-mcherry/pBAD-GFP(-30)* | This work | CJW7485 |
| MG1655 *rplA::rplA-mcherry/pBAD-GFP(-7)* | This work | CJW7486 |
| MG1655 *rplA::rplA-mcherry/pBAD-GFP(0)* | This work | CJW7487 |
| MG1655 *rplA::rplA-mcherry/pBAD-GFP(+7)* | This work | CJW7488 |
| MG1655 *rplA::rplA-mcherry/pBAD-GFP(+11a)* | This work | CJW7489 |
| MG1655 *rplA::rplA-mcherry/pBAD-GFP(+11b)* | This work | CJW7490 |
| MG1655 *rplA::rplA-mcherry/pBAD-GFP(+15)* | This work | CJW7491 |
| MG1655 *rplA::rplA-mcherry/pBAD-GFP(+25)* | This work | CJW7492 |
| MG1655 *ΔglgBXCAP-FRT-kan-FRT* | This work | CJW7537 |
| MG1655 *ΔglgBXCAP* | This work | CJW7587 |
| MG1655 *lacY(A177C) araFGH::spec ΔlacI ΔaraE araBAD::dCas9 galM < PBBa-J23119-sgRNA(ftsZ)-(S. pyogenes terminator)-(rrnB terminator)> gmpA attB::PproC-msfGFP* | This work | CJW7588 |
| MG1655 *Tn7::Ptac-GFPmut3-CBM20 FRT-cat-FRT* | This work | CJW7601 |
| MG1655 *ΔglgBXCAP Tn7::Ptac-GFPmut3-CBM20-FRT-cat-FRT* | This work | CJW7604 |
| MG1655 *Tn7::P$_{tac}$-GFPmut3-CBM20 FRT-cat-FRT hupA::hupA-mCherry FRT-kan-FRT* | This work | CJW7605 |

| Reagent/resource | Reference/source | Identifier or catalog number |
|---|---|---|
| MG1655 Tn7:: Ptac -GFPmut3-CBM20-FRT-cat-FRT | This work | CJW7606 |
| MG1655 ΔglgBXCAP Tn7::Ptac-GFPmut3-CBM20 FRT-cat -FRT | This work | CJW7607 |
| MG1655 hupA::hupA-mcherry | This work | CJW7660 |
| MG1655 ΔglgBXCAP hupA::hupA-mCherry | This work | CJW7661 |
| MG1655 hupA::hupA-mCherry Tn7:: PRpsL-mSCFP3-FRT-kan-FRT | This work | CJW7665 |
| MG1655 hupA::hupA-mCherry Tn7:: PRpsL-mVenus-FRT-kan-FRT | This work | CJW7666 |
| MG1655 ΔglgBXCAP hupA::hupA-mCherry Tn7::PRps-mSCFP3-FRT-kan-FRT | This work | CJW7667 |
| MG1655 ΔglgBXCAP hupA::hupA-mCherry Tn7:: PRps-mVenus-FRT-kan-FRT | This work | CJW7668 |
| MG1655 ΔglgBXCAP /pEB2-mscarlet-I | This work | CJW7718 |
| MG1655 ΔminD minE::sfgfp-minD minE::frt kanR frt | This work | CJW7872 |
| MG1655 ΔglgBXCAP rne::rneΔMTS-mCherry | This work | CJW7877 |
| MG1655 ΔglgBXCAP rplA::rplA-mCherry attB::P_{proC}-msfGFP | This work | CJW7878 |
| MG1655 (DE3) hupA::hupA-mcherry rplA::rplA-msfgfp-frt-kanR-frt/ pET28:mTagBFP2-CmR | This work | CJW7798 |
| MG1655 Tn7:: PRpsL-mSCFP3-FRT-kan-FRT | Gift from Johan Paulsson | JP1456 |
| MG1655 Tn7::PRpsL-mvenus-FRT-kan-FRT | Gift from Johan Paulsson | JP1457 |
| MG1655 lacY(A177C) araFGH::spec ΔlacI ΔaraE araBAD::dCas9 galM < PBBa-J23119-sgRNA(ftsZ)-(S. pyogenes terminator)-(rrnB terminator)> gmpA | Li et al, 2016 | SJ_XTL229 |
| **Recombinant DNA** | | |
| pBAD-GFP (−30) | Schavemaker et al, 2017 | |
| pBAD-GFP (−7) | Schavemaker et al, 2017 | |
| pBAD-GFP (0) | Schavemaker et al, 2017 | |
| pBAD-GFP (+7) | Schavemaker et al, 2017 | |
| pBAD-GFP (+11a) | Schavemaker et al, 2017 | |
| pBAD-GFP (+11b) | Schavemaker et al, 2017 | |
| pBAD-GFP (+15) | Schavemaker et al, 2017 | |

| Reagent/resource | Reference/source | Identifier or catalog number |
|---|---|---|
| pBAD-GFP (+25) | Schavemaker et al, 2017 | |
| pEB2-mScarlet-I | AddGene | 104007 |
| pkD13-msfGFP | Gray et al, 2019 | |
| pET28:mTagBFP2-CmR | Papagiannakis et al, 2025 | |
| **Oligonucleotides** | | |
| Gene blocks used for strain construction | Integrated DNA Technologies | See Table EV5 |
| TACTAGAGGACGAACAAT AAGGCCTCCCTAACGGGG GGCCTTTTTTATTGATAAC AAAAGTGTAGGCTG GAGCTGCTTCG | Integrated DNA Technologies | YP32 |
| AATCGCTCAAGACGT GTAATGCTGCAATC | Integrated DNA Technologies | YP33 |
| TCAAAAGGCGAAGA ACTTTTTACCG | Integrated DNA Technologies | YP34 |
| TAGGGAGGCCTTATTGT TCGTCCTCTAGTATTATTT ATACAATTCATCCATT CCATGAGTGA | Integrated DNA Technologies | YP35 |
| CCAGTGCCAAGCTTGC ATGCAGATTGCAGCAT TACACGTCTTGAGCGATT CACAGCTAACACCACGTCG | Integrated DNA Technologies | YP36 |
| TACACCGGTAAAAAGTTC TTCGCCTTTTGACATCTA GTATTCTCCTCTTTCTCT AGTAAAAGTTAAACAA AATTATTTGTAGAGGGAAAC | Integrated DNA Technologies | YP37 |
| **Chemicals, enzymes, and other reagents** | | |
| P1 phage | ATCC | 25404-B1 |
| Ampicillin | Fisher Scientific | BP1760-25 |
| Kanamycin | Sigma-Aldrich | K1377-25G |
| Chloramphenicol | Sigma-Aldrich | C0378-25G |
| 4′,6-diamidino-2-phenylindole (DAPI) | ThermoFisher | D1306 |
| 2-NBDG | Cayman Chemical Company | 11046 |
| MitoTracker Green | ThermoFisher | M7514 |
| SYTO RNASelect | ThermoFisher | S32703 |
| Cephalexin | Sigma-Aldrich | C4895 |
| PEG1500 | Sigma-Aldrich | 81210-500 G |
| PEG3000 | Sigma-Aldrich | 8190151000 |
| PEG4000 | Hampton Research | HR2-605 |
| PEG6000 | Sigma-Aldrich | 81253-250 G |
| PEG8000 | Sigma-Aldrich | 89510-250G-F |
| PEG10000 | Sigma-Aldrich | 81280-1KG |
| PEG20000 | ThermoFisher | A17925.0B |
| PEG35000 | Sigma-Aldrich | 81310-1KG |

| Reagent/resource | Reference/ source | Identifier or catalog number |
|---|---|---|
| PEO100000 | Sigma-Aldrich | 181986-250 G |
| PEG1000000 | Sigma-Aldrich | 372781-250 G |
| Glycogen | Millipore Sigma | 361507-1 ML |
| FITC-ConA | Sigma-Aldrich | C7642-2MG |
| Pluronic® F-108 | Sigma-Aldrich | 542342-250 G |
| GFP | ThermoFisher | A42613 |
| mCherry | Abcam | AB199750 |
| **Software** | | |
| Oufti | Paintdakhi et al, 2016 | https://oufti.org/ |
| MATLAB R2024b | MathWorks | https://www.mathworks.com/products/matlab.html |
| ImageJ | National Institute of Health | https://fiji.sc/ |
| Python 3 | Phyton Software Foundation | https://www.python.org/ |
| SyMBac | Hardo et al, 2022 | https://symbac.readthedocs.io/en/latest/intro.html |
| **Other** | | |
| Custom-made functions and scripts | This study | https://github.com/JacobsWagnerLab/published/tree/master/Thappeta_Canas-Duarte_et_al_2025 |

## Methods and protocols

### Bacterial strains and growth conditions

Bacterial strains and descriptions of their constructions can be found in the Reagent Table and Table EV4. P1 transductions were performed as previously described (Thomason et al, 2007) with the exception that donor strains were grown without glucose supplementation. A lysate of P1 phages was obtained from ATCC (25404-B1). Oligomers used for polymerase chain reactions (PCR) are listed in the Reagent table. Lambda red recombination was performed as previously described using the pKD46 (Datsenko and Wanner, 2000) or pSIM6 plasmids (Diner et al, 2011). Unless otherwise indicated, cells were grown at 30 °C in M9 medium (26.11 mM Na$_2$HPO$_4$, 22 mM KH$_2$PO$_4$, 8.55 mM NaCl, 18.7 mM NH$_4$Cl, 2 mM MgSO$_4$, and 0.1 mM CaCl$_2$) supplemented with 0.2% glucose, 0.1% casamino acids, and 1 µg/ml thiamine (M9gluCAAT). Cells were first inoculated in the appropriate growth medium and grown to the stationary phase in culture tubes. They were then re-inoculated into fresh media by diluting at least 10,000-fold and grown until they reached the indicated optical density (OD) at 600 nm. When appropriate, ampicillin (100 µg/mL), kanamycin (50 µg/mL), or chloramphenicol (30 µg/mL) was added to the media for selection.

Strains expressing green fluorescent proteins with different net charges (Schavemaker et al, 2017) were grown in M9gluCAAT, and fluorescent protein expression was induced with 0.4% arabinose

upon re-inoculation of stationary phase cultures. Given the richer medium condition from the addition of arabinose, the transition phase OD ranged from 2.9 to 3.5 instead of 1.2 to 2.7.

FtsZ depletion experiments were conducted using strain CJW7588 in the presence of 0.2% arabinose. For exponentially growing cultures, arabinose was added for the equivalent of 2–3 doublings. For transition-phase cultures, arabinose was added in the late exponential phase (~ OD 0.4–0.5) and imaged ~5 h later when cultures reached transition-phase ODs (the equivalent of 2–3 doublings) on an agarose pad made of spent medium.

### Dyes and labels

To label the nucleoid, cells were incubated with 1 µg/ml 4′,6-diamidino-2-phenylindole (DAPI) for 10 min prior to imaging. To visualize intracellular 2-NBDG localization, cells were grown in M9CAAT supplemented with 0.2% L-arabinose to the desired OD and incubated with 10 µM 2-NBDG for 10 min, then washed twice in prewarmed spent medium filtered with a 0.22-µm filter. For the following dyes, concentrations were increased when used on transition-phase cells, as cell permeability seemed to decrease with increasing ODs, consistent with a previous report (Bakshi et al, 2014). To visualize the inner membrane of *E. coli* cells, MitoTracker Green (Shi et al, 2021) was added to the cell culture at a final concentration of 100 nM (exponential phase) or 1 µM (transition phase) and incubated for 30 min prior to imaging. To label RNA, SYTO RNASelect was used at a final concentration of 500 nM (exponential phase) or 2.5 µM (transition phase) for 10 min and washed twice in filtered spent medium (Bakshi et al, 2014) before imaging.

### Cephalexin treatment

For cephalexin treatment of exponentially growing cultures, the antibiotic (50 µg/mL) was added for the equivalent of 2–3 doublings. For transition phase samples in M9gluCAAT, cells were incubated with cephalexin (50 µg/mL) in late exponential phase (OD ~ 0.4–0.5) and imaged ~5 h later when cultures reached transition phase (the equivalent of 2–3 doublings) on agarose pads made of spent medium containing cephalexin.

### Co-cultures

For co-cultures experiments related to Figs. 3B–D, 5A,C–E, EV3, EV4, and EV7, cells of four strains (CJW7665-7668), each expressing a different fluorescent protein in either the WT or ΔglgBXCAP background, were inoculated in 2 mL of M9gluCAAT and grown overnight at 30 °C with shaking (220 rpm). Each culture was then diluted 1:10,000 into fresh 2-mL cultures and returned to the shaker until they reached an OD of ~0.1. Samples of WT and glycogen-devoid strains were mixed proportionally to their ODs to create a ~50:50 mixture. For these experiments, two types of mixtures were used: one included WT cells expressing mSCFP3 and mutant cells expressing mVenus, while the other included WT cells expressing mVenus and mutant cells expressing mSCFP3. Mixtures were made independently in three biological replicates. Each mixture was then diluted 1:10,000 into 50 mL of medium in a 250-mL flask and returned to the shaker. Samples were taken at OD of ~2.6 for the transition phase. Samples were prepared for imaging on agarose pads as described above.

For the AFM experiments related to Figs. 7, EV9, EV10 and EV11, WT cells expressing the glycogen sensor (CJW7605),

ΔglgBXCAP cells expressing the glycogen sensor (CJW7604) or mVenus (CJW7668), and cells overexpressing mTagBFP2 (CJW7798) were inoculated separately in 2 mL of M9gluCAAT medium and grown overnight at 30 °C with shaking (220 rpm). Mixtures of the three cell types were made and diluted 1:100,000 into 50 mL of medium in a 250-mL flask and returned to the shaker. When the mixtures reached an OD of ~0.5, IPTG was added to a final concentration of 100 mM to induce the over-expression of mTagBFP2. Cultures were returned to the shaker and allowed to reach a final OD of ~1.6. Samples were imaged in agarose pads containing filtered spent medium ($OD_{600}$~ 1.7) to verify the presence of protein aggregates in cells with mTagBFP2 fluorescence.

### Microscopy

Unless otherwise indicated, cells were imaged on 1.5% agarose pads made with the appropriate spent growth medium. For samples with high cell density in late transition or stationary phase, cells were diluted in warm filtered spent medium before spotting on an agarose pad for ease of cell segmentation.

Microscopy was performed using Nikon inverted microscope set-ups controlled by the NIS-Elements AR software, with the following specifications. One of the set-ups consisted of a Nikon Ti2-E inverted microscope equipped with a Perfect Focus System (PFS), a motorized stage, a 100x Plan Apo 1.45NA Ph3 oil objective, a Photometrics Prime BSI back-illuminated sCMOS camera, a Lumencor Spectra III LED light engine excitation source, a polychroic mirror (FF-409/493/596-Di02 for by Shemrock) combined with a triple-pass emitter (FF-1-432/523/702-25 by Shemrock) for GFP/DAPI/mCherry, and a polychroic mirror (FF-459/526/596-Di01 by Shemrock) combined with a triple-pass emitter (FF01-475/543/702-25 by Shemrock) for CFP/YFP/mCherry. The temperature was set and maintained at the indicated value using a temperature chamber (Okolabs). The second microscope setup consisted of a Nikon Eclipse Ti microscope equipped with a Hamamatsu ORCA-Flash 4.0 camera, a 100X objective (Nikon, OFN Ph3 DM, N.A. 1.45), and a Spectra X light engine (Lumencor). The following Chroma filter sets were used to acquire fluorescence images: DAPI (excitation ET350/50x, dichroic T400lp, emission ET460/50 m), GFP (excitation ET470/40x, dichroic T425lpxr, emission ET525/50 m), and mCherry/TexasRed (excitation ET560/40x, dichroic T585lpxr, emission ET630/75 m). The temperature was maintained at 30 °C using a customized enclosure and temperature controller (Air-Therm SWT, World Precision Instrument) for time-lapse imaging. The temperature was also maintained for snapshots, but it did not affect transition-phase phenotypes in the time window required for imaging (under 12 min).

### Image processing and analysis

For sample images with cell contours on agarose pads, cells were detected from phase-contrast images using the open-source software package Oufti (Paintdakhi et al, 2016). For the images shown in the figures, fluorescent image background subtraction was done using Fiji (Schindelin et al, 2012) and a sliding paraboloid with a rolling ball radius of 50 pixels. Cell signal intensity profiles were generated using Oufti's signal output divided by the total signal intensity to calculate the relative signal, which was plotted for individual cells.

For all experiments on agarose pads, cells were segmented from phase-contrast images using the SuperSegger-Omnipose software (https://github.com/tlo-bot/supersegger-omnipose) (Stylianidou et al, 2016; Cutler et al, 2022). The software's pre-trained bact_phase_omni model (Cutler et al, 2022; Stylianidou et al, 2016) was retrained in-house prior to usage due to its tendency to split cells prematurely before cell division was completed, as determined based on visual inspection. Model retraining was carried out in two steps. First, images of CJW7606 cells (OD = 2.83) grown in M9gluCAAT at 30 °C were collected on agarose pads containing M9gluCAAT. Cells from 11 fields of view (each of $2048 \times 2048$ pixels, ~100–250 cells per field of view) were segmented using the pre-trained model bact_phase_omni. The obtained masks were visually inspected, and when the pre-trained model was splitting cells too early, the cell masks were merged back into one single mask using a custom MATLAB code (LabelsMerge_ConstrictingCells.m). The corrected $2044 \times 2048$-pixel images (both phase-contrast images and cell masks) were then divided into smaller $510 \times 512$-pixel images. The dataset was further augmented by performing image rotations (90°, 180°, and 270°), resulting in 704 images. This initial training dataset was then used to retrain Omnipose from scratch (with parameters: n_epochs = 4000; tyx = 224,224; batch_size = 16; learning_rate = 0.1, see details in https://omnipose.readthedocs.io) to generate a preliminary model. In the second step, the retrained model was used to re-segment the publicly available bact_phase training dataset (https://osf.io/xmury/) that was originally used to train bact_phase_omni, including additional images of cells with extreme phenotypes (e.g., with bright intracellular regions in phase-contrast images).

This generated a larger dataset with a broad spectrum of cell morphologies. The obtained segmented images were then manually curated to eliminate poorly segmented cells. Finally, both training datasets (the first one created in-house and the one re-segmented from bact_phase) were used to retrain Omnipose from scratch ($n = 1249$ images, with parameters: n_epochs = 4000; tyx = 224,224; batch_size = 16; learning_rate = 0.1). The resultant model (merge_model_omni) and the Python code used for curating (screening_good_bad.py) and generating the training datasets (generate_training_dataset.py) are available on GitHub (https://github.com/JacobsWagnerLab/published/tree/master/Thappeta_Canas-Duarte_et_al_2025).

Unless specified otherwise, merge_model_omni was used to segment cells across experiments. After segmentation, the resulting cell.mat files and the generated labels (masks.png) were imported to MATLAB, organized into structures, and further analyzed using custom scripts. Morphological features, including cell length, cell area, cell width, and circularity, were extracted from the cell labels using MATLAB's function regionprops() and used to filter out incorrect segmentations. For each experiment, histograms were generated for each of the above-mentioned features, and the filtering thresholds were adjusted accordingly.

The mid-cell axis of each cell was calculated using a bivariate fit to the distance matrix of the cell mask and used to identify the location of the cell centroid. For all cells, the distance between the mid-cell axis and the mask outline of each vertical half of the cell was computed, smoothed, inverted, and used to identify peaks (function findpeaks(), with parameters MinPeakProminence = 0.1 and MinPeakDistance = 15). For constricting cells, the division plane (constriction site) was then localized using the location of identified peaks on each cell half. Constriction offset was then

determined by measuring the absolute distance between the construction site and the cell centroid, followed by a normalization by the cell length.

Nucleoids were initially segmented using Otsu, after which nucleoid objects were refined by calculating the distance transform (function bwdist()) of the preliminary nucleoid mask and applying the watershed transformation (function watershed(), using connectivity = 4) on the inverse of the calculated distance transform. Binary opening, closing, and fill operations were applied to the nucleoid mask. For cells with multiple nucleoids, the nucleoid centroid was defined as the mid-point between the pair of the most distal nucleoid objects. Nucleoid asymmetry was then determined by measuring the absolute distance of the nucleoid mid-point to the cell's centroid, which was then normalized by the cell length. The analysis described above was performed using the custom MATLAB script PostSGO_AgarPad_FigureOne_AllODs.m.

To quantify the signal distribution of the different reporters used in this study, we estimated both the signal correlation function (SCF) (Gray et al, 2019) and the normalized signal difference between the two cell poles. To calculate the SCF, the centerline was used to define a rectangular mask with a width of 4 pixels and a length equal to the cell length minus 5 pixels from each cell pole. The SCF mask was then used to extract the signal intensity from the fluorescence images. The correlation coefficient was then calculated between the corresponding signals using the function corrcoef(). Likewise, the normalized pole difference was calculated by defining two masks, each encompassing the region between the cell pole and the cell centroid that corresponds to 30% of the cell length. To avoid boundary effects, the pole masks were eroded by 2 pixels. As described above, the masks were used to extract the signal intensity of the corresponding fluorescence images. The normalized pole signal difference was then calculated by subtracting the average intensity between the poles and dividing it by the average signal intensity for the whole cell. These analyses were performed using a custom MATLAB script PostSGO_PoleRatio_SCF_AllCells_20250615.m. The Spearman correlation coefficient between the normalized signal intensities of free msfGFP and RplA-mCherry was calculated using the corr() function, and the contour plot was obtained using a MATLAB script (ContourPlot_20250311.m).

Time-lapse agarose pad data were analyzed as described above, with added lineage/cell polarity tracking. Tracking was performed by first identifying lineages (microcolonies) through spatial clustering using the MATLAB DBSCAN (Density-Based Spatial Clustering of Applications with Noise) algorithm (epsilon = 45 and minpts = 4) and then using the built-in genealogy information calculated by Supersegger within each lineage. To track the polarity (old vs new pole) of the cells, cells were oriented vertically, and the top-most pole was assigned as pole 1 and the bottom-most pole as pole 2. Cells present at the beginning of the time-lapse sequence were labeled as generation zero, and each of their daughters was labeled as generation one, and so forth. A total of three divisions were analyzed, and polarity was assigned accordingly. Normalized pole signal differences were calculated as described previously. This analysis was performed using the custom MATLAB script PostSGO_AgarPAD_TL_20250613.

The degrees of both constriction position and nucleoid mid-point asymmetry in the WT strain were analyzed for each co-culture (Mix 1 and Mix 2) in the transition phase using a MATLAB code (PostSGO_MixesConstriction_20240312.m). For this analysis, constricting cells were identified as described above using the

centerline to determine the constriction position and the cell centroid. The HupA-mCherry signal in each constricting cell was used to segment the nucleoid and calculate its centroid. The offsets between the constriction position and the cell centroid, and between the nucleoid mid-point and the cell center were calculated and normalized by the length of each constricting cell. The Spearman correlation coefficient and the isocontours were then calculated between the relevant variables as described above.

For the microfluidic experiments, cell segmentation was performed using Omnipose. For this purpose, we retrained Omnipose using a two-step approach. First, a set of synthetic micrographs of bacteria in a mother machine-like system was generated using SyMBac (Hardo et al, 2022). The image simulation parameters were set to match that of the optical system previously described. Individual mother machine images were then tiled, along with their corresponding ground truth masks to produce a training dataset of simulated micrographs ($n = 150$). These synthetic micrographs were then used to retrain Omnipose from scratch. This first model was then used to segment real (non-synthetic) phase-contrast images of cells in our mother machine microfluidics system. The obtained segmentation results were then manually curated (screening_good_bad_trenches.py), augmented by mirroring individual trenches along their y-axis to double the size of the training data, and tiled back together in images containing five trenches (create_training_dataset_trenches.py), to produce an image dataset of 237 images. Both empty and cell-filled trenches were included in this final dataset. Finally, Omnipose was retrained from scratch using both the original synthetic images, as well as the curated and augmented real images, to generate the final Omnipose model (MM_model_omni; $n = 423$ images, with parameters: tyx = 224,224; batch_size = 16; learning_rate = 0.1).

To improve the accuracy of segmentation, fields of view were cropped and aligned using the pre-segmentation modules of Supersegger. After segmentation, the mother-cell lineages were extracted and analyzed using a custom MATLAB script (Post-SGOMM_GC.m). Briefly, lineages were initially identified using the MATLAB DBSCAN function (epsilon = 25 and minpts = 10). From there, mother cells were identified in each lineage and tracked throughout all acquired time points. Only lineages with successfully segmented mother cells and tracking across all (84) frames were considered for the analysis. Instantaneous growth rates were calculated using a linear fit to the log-transformed cell lengths. The normalized nucleoid offset and glycogen area were calculated as described above.

The correlation between polar glycogen accumulations and the nucleoid offset was analyzed from snapshot images of CJW7606 cells (OD 2.83) that were segmented using a combination of SuperSegger-Omnipose and a MATLAB script (PostSGO_GlycogenSensorAllCells_20240312.m), in which the centerline of each cell was calculated as described above. Fluorescent signals from HupA-mCherry and the glycogen sensor were used to segment the nucleoids and the areas of glycogen accumulation, respectively. The difference between the glycogen sensor area for each cell pole was then calculated. Nucleoid offset was determined from the distance between the nucleoid mid-point and the cell center. The Spearman correlation coefficient between the two variables was calculated, and the contour plot was plotted using a MATLAB script (ContourPlot_20240312.m).

To examine the correlation between polar glycogen accumulations and asymmetric division, constricting cells in the transition phase were identified, curated, and analyzed using a custom MATLAB code (PostSGO_GlycogenSensorConstriction_20240312.m). In short, the centerline of each cell was calculated as described above and was used to identify the location of the cell constriction plane. Fluorescence signal was extracted from each cell area flanking the division side (i.e., the future daughter cells) and used to create a mask for the glycogen sensor signal. The differences between each future daughter cell's area and between the glycogen sensor signal areas at each pole were used to calculate the Spearman correlation coefficient. Principal components regression was used to calculate the linear regression shown in Fig. 6B.

### Whole-cell NMR sample preparation and solid-state NMR measurements

Overnight cultures of each bacterial strain were prepared and diluted at least 10,000-fold into 300 mL of the appropriate fresh medium in a 1-L flask. To achieve similar final quantities of biomass, two flasks were prepared identically for exponential-phase cultures, and one flask was prepared for the higher cell density transition-phase samples. Cells were pelleted by centrifugation at $10,000 \times g$ at 4 °C for 10 min and washed twice in cold phosphate-buffered saline (PBS). Each final sample cell pellet was collected in a 50-mL tube, stored at −80 °C, and subsequently lyophilized and packed into a magic-angle spinning NMR rotor.

Solid-state $^{13}$C CPMAS NMR (Schaefer and Stejskal, 1976) experiments were performed using an 89-mm bore 11.7 T magnet (Agilent Technologies) with an Agilent triple resonance BioMAS probe and DD2 console (Agilent Technologies). Samples were spun at 7143 ± 3 Hz at room temperature in thin-walled (36 μL capacity) 3.2-mm outer diameter zirconia rotors. Cross polarization (CP) was performed with a $^{13}$C field strength of 50 kHz and a $^1$H field strength centered at 57 kHz with a 10% ramp for the contact time of 1.5 ms. $^1$H decoupling was performed with two-pulse phase-modulated (TPPM) (Bennett et al, 1995) decoupling at 83 kHz. All CPMAS experiments were obtained using a recycle delay of 2 s. Free induction decays were processed using 80 Hz exponential line broadening prior to Fourier transformation. Spectrometer chemical shift referencing was performed by setting the high-frequency adamantane peak to 38.5 ppm (Morcombe and Zilm, 2003).

### Intracellular salts buffer

To mimic the ionic strength of the *E. coli* cytoplasm, a potassium phosphate buffer (PPB) was prepared and supplemented with the appropriate concentrations of intracellular salts. For this, 200 mL of 1 M stock PPB (pH = 7.5) was prepared by mixing 25.63 g of $K_2HPO_4$ and 7.2 g of $KH_2PO_4$ in MilliQ water. For the intracellular salts buffer (referred to also as IS buffer), a 5× stock was prepared by mixing the 1 M PPB stock to a final 200 mM concentration with 0.425 M NaCl, 1.25 M KCl, 12.5 mM $MgCl_2$, and 0.5 mM $CaCl_2$ (Alatossava et al, 1985; Gangola and Rosen, 1987; Szatmári et al, 2020; Schultz et al, 1962).

### Dynamic light scattering measurements

The hydrodynamic diameter of purified mussel glycogen (Sigma-Aldrich 361507-1 ML) and commercially available PEG/PEO particles with varying molecular weights was measured using a NanoBrook Omni device. Stock solutions of PEG1500 (Sigma-Aldrich 81210-500 G), PEG3000 (Sigma-Aldrich 8190151000), PEG4000 (Hampton

Research HR2-605), PEG6000 (Sigma-Aldrich 81253-250 G), PEG8000 (Sigma-Aldrich 89510-250G-F), PEG10000 (Sigma-Aldrich 81280-1KG), PEG20000 (ThermoFisher A17925.0B), and PEG35000 (Sigma-Adrich 81310-1KG) (30% w/v) were prepared in MilliQ water. Stock solutions 5% and 2.5% w/v were made of PEO100000 (Sigma-Aldrich 181986-250 G) (5% w/v) and PEG1000000 (Sigma-Aldrich 372781-250 G) (2.5% w/v. When needed, solutions were incubated at 42 °C for 1 h to help homogenization.

For measurements, samples were diluted to a final volume of 10 mL to generate a final concentration of 6% (1.5 kDa PEG and 3 kDa PEG), 3% (4 kDa PEG),1.5% (6 kDa PEG, 8 kDa PEG, 10 kDa PEG, 20 kDa PEG and 35 kDa PEG), 0.25% (PEO 100 kDA) or 0.025% PEO 1 MDa in either MilliQ water (pH = 6.5) or in the 1X IS buffer (pH = 7.0). Dilutions of the samples were made to ensure that measurements with each PEG/PEO crowder were done with concentrations below their overlap concentrations to avoid self-crowding artifacts in our measurements (de Gennes and Witten, 1980; Smith et al, 2023; Julius et al, 2019). The hydrodynamic diameter of glycogen was measured in the same IS buffer at a concentration of 2 g/L. To remove dust or other particles that could affect the measurements, all samples were double-filtered, first with a 0.22-μm filter followed by a 0.1-μm filter. Each sample (1 mL) was loaded into a cuvette and sealed with a lid. Samples were measured in triplicate for 300 ms at 25 °C using a 90° angle for the detector.

### Preparation of the microfluidic chips

The microfluidic device used in this study was cast from an epoxy mold kindly shared by the Paulsson lab (Bakshi et al, 2021). In this device, the feeding channel has a width of 350 μm and a height of 25 μm. Cell trenches have a width of 1.3 μm, a length of 25 μm, and a height of 1.25 μm. Polydimethylsiloxane (PDMS), a silicone elastomer composed of dimethylsiloxane monomers, was prepared by mixing the base polymer with its curing agent in a 10:1 weight ratio. The mixture was degassed and poured onto the epoxy mold and then cured at 65 °C for 4 h. The cured PDMS was then carefully cut out and peeled from the mold. Individual mother-machine chips were cut out of the PDMS, and holes for the inlets and outlets were created using a biopsy puncher (0.75 mm diameter). The mother machine-like chips were cleaned with isopropanol, blow-dried with a nitrogen gun, and then cleaned with Scotch tape before bonding. Glass-bottom dishes (40 mm diameter, 14026-20, from Ted Pella Inc., CA, USA) were cleaned with isopropanol and blow-dried with a nitrogen gun. Finally, the PDMS chip and glass dish were plasma-treated (for 20 s at 60 W), immediately bonded, and baked at 65 °C for 4 h. Chips were bonded the day before being used in the experiment.

### Microfluidic setup

A simplified version of a previously described growth curve platform (Bakshi *et al*, 2021) was built using a peristaltic pump (T60-S2&WX10-14-H, Langer Instrument, USA) and an in-house-built bubble trap. Platinum-cured silicon tubing (Masterflex™ L/S™ Platinum-Cured Silicone Precision Tubing, Tubing size 13, Fisher Scientific) was used to create the flow path. Blunt-end needles (1-1/2" Gauge 20, McMaster-Carr) were bent to create stable connections to the microfluidic chip. Before the experiment, the flow path was cleaned sequentially with 20% bleach, 20% ethanol, and Milli-Q water, each for 20 min. After cleaning, fresh M9gluCAAT

medium supplemented with 0.8% Pluronic® F-108 was flowed through the path for 10 min before connecting the loaded chip.

### Microfluidics experiment and analysis

CJW7605 and CJW7668 strains were separately inoculated in 2 mL of M9gluCAAT medium supplemented with 0.08% Pluronic® F-108 and grown overnight at 30 °C with shaking at 220 rpm. An aliquot of the liquid cultures was then diluted 1:10,000 into 50 mL of M9gluCAAT medium in a 250-mL flask and returned to the shaker until reaching an OD of ~0.4. Three milliliters of each culture were spun at 6785 rcf and concentrated into 500 µL. The two cultures were mixed at this point and spun again at 6785 rcf to concentrate them to a final volume of ~30 µL. Cells were loaded into the microfluidic chip using gel-loading tips (Fisherbrand™ Gel-Loading Tips, 1-200 µL, Fisher Scientific). The loaded chip was then spun at $500 \times g$ for 3 min using a modified version of the holder designed and shared by the Paulsson laboratory. The loaded chip was then connected to the flow path and placed inside the microscope incubator for 1 h to allow time for equilibration. After this time, fields of view were selected, and the batch culture of strain CJW2168 was inoculated using a 1:100,000 dilution. Imaging was started immediately following the inoculation of the culture, with a frame rate of 10 min for all the acquired channels.

### In vitro phase separation experiments

To generate the presented phase diagram (Fig. 6E), phase separation was assessed using varying concentrations of both glycogen (2.5–10 g/L) and 3 kDa PEG (8–30 mM). Samples were prepared in the IS buffer to a final volume of 50 µL in 200-µL Eppendorf tubes. For visualization, 0.05 µg/µL of FITC-ConA (Sigma-Aldrich C7642-2MG) was added to the appropriate glycogen volume prior to the addition of other components of the mixture. The appropriate volume of 3 kDa PEG was added last, and a 12-µL aliquot was immediately transferred to a $50 \times 7$ mm glass-bottom dish (PELCO) for imaging at 25 °C.

To determine the minimal concentration of PEG/PEO that promotes glycogen phase separation, samples were prepared as described above with 9 g/L of glycogen. Increasing concentrations of each PEG (1.5–35 kDa) or PEO (100 kDa and 1 MDa) were tested in 50 µL mixtures. Each sample (12 µL) was imaged by phase-contrast microscopy (100x Ph3 objective) to visually assess if glycogen condensates appeared. The smallest concentration at which the solutions transitioned from one phase to another was assigned as the minimal concentration. For PEO 1 MDa, the minimal concentration required to drive phase separation was found to be above the overlap concentration (0.5% w/v) reported for this polymer (Smith et al, 2023), which marks the onset of "self-crowding". Therefore, the hydrodynamic diameter of the polymer above this concentration could not be determined using dynamic light scattering. Unless otherwise indicated, all experiments and imaging were conducted at 25 °C.

For the fluorescent protein exclusion assay, glass-bottom dishes were cleaned using 1 mL of 100% isopropanol, then washed twice with 90% ethanol, and allowed to dry completely. Dishes were then treated with Pluronic® F-108 (Sigma-Aldrich 542342-250 G) to minimize fluorescent protein binding to the glass. Briefly, 200 µL of IS buffer supplemented with 0.1% w/v Pluronic® F-108 was pipetted into the dish and allowed to incubate for 1 h, followed by two washes with 200 µL of IS buffer. All remaining liquid was removed, and the dish was allowed to air dry completely. Mixtures were made to final concentrations of 10 g/L of glycogen, 1× IS buffer, 15 µM GFP (ThermoFisher A42613) or mCherry (Abcam AB199750), and 20 mM 3 kDa PEG as described above. Samples (12 µL) were imaged at 25 °C, 30 °C, and 37 °C. For imaging, the focal plane was set on the surface of the glass to minimize background signal from the column of liquid around the condensates (Fig. 6I), except for the images shown in Movie EV7 for which images were taken as a z-stack (with 0.125 µm steps) above the glass surface. For the images shown in the figures, fluorescent image background subtraction was done using Fiji (Schindelin et al, 2012) and a sliding paraboloid with a rolling ball radius of 50 pixels.

### MinD-GFP experiments

The CJW7872 strain was inoculated in 2 mL of M9gluCAAT medium and grown overnight at 30 °C with shaking at 220 rpm. An aliquot of the culture was then diluted 1:10,000 into 50 mL of medium in a 250-mL flask and returned to the shaker. Samples were taken at OD 1.7, stained with DAPI as described above, and spotted onto agarose pads containing transition-phase spent medium. Snapshots were acquired at time zero in the phase-contrast, DAPI, and FITC channels before the acquisition of a FITC time series (every 5 s for 95 s) to observe the MinD-GFP oscillations. A final snapshot series in all three channels was performed to correct for cell movement during the time series.

### Fluorescence recovery after photobleaching

CJW7605 cells were inoculated in a 2-mL solution of M9gluCAAT and grown overnight at 30 °C with shaking (220 rpm). An aliquot of the culture was then diluted 1:10,000 into 50 mL of medium in a 250-mL flask and returned to the shaker. Samples were taken at OD 1.7 and spotted onto agarose pads containing conditioned medium. Snapshots were taken before bleaching. For each field of view, four cells were, on average, selected for FRAP analysis. A disk (diameter = 5 pixels) was used to specify the region of interest for photobleaching and positioned at one of the cell poles, covering fully or partially the glycogen accumulation area. The regions of interest were bleached with a 405 nm laser at 25% power for 30 ms. Images were then acquired every 50 ms for 30 s following the bleaching.

Photobleaching correction was performed globally for each field of view using a custom Fiji macro (PhotoBleachCorrection.ijm). Briefly, the average fluorescence intensity across all control (non-bleached) cells was calculated for each time point and used to calculate a correction value centered on the fluorescence intensity at $t = 0$ (before photobleaching). The calculated correction factor for each time point was then applied to all pixels in the field of view. For each field of view, the photobleached cells and a randomly selected control cell were further analyzed.

After correction for photobleaching, FRAP analysis was performed using a custom MATLAB script (FRAP_Analysis.m). In brief, intensity profiles were extracted from the photobleached and non-photobleached regions using a circular region of interest of 5 pixels in diameter. Signal profiles were background-subtracted and normalized such that the first frame after photobleaching was zero and the maximum intensity after recovery was one. Each

profile was then fitted by an exponential function to determine the half-time to maximal recovery. A total of 70 photobleached cells were analyzed.

### Atomic force microscopy experiments

*E. coli* cells were immobilized on poly-L-lysine-coated glass-bottom Petri dishes. Fifty microliters of 0.01% poly-L-lysine were dropped on the glass-bottom Petri dish, air-dried for an hour, rinsed with Milli-Q water, and then dried with a nitrogen flow. A diluted cell suspension (500 μL) was dropped on the coated dishes and incubated at room temperature for 30 min before being rinsed with PBS five times to remove loose cells. Then, 2 mL of PBS were added to the dishes before AFM imaging. All AFM experiments were performed with a JPK NanoWizard V instrument (Bruker, USA) mounted on the inverted optical microscope Axio Observer (Zeiss, Germany). Depth-resolved stiffness map and stiffness tomography data were collected using the Quantitative Imaging (QI) mode with a force setpoint of 20 nN. SCANASYST-FLUID cantilevers (Bruker, USA) with a nominal spring constant of 0.35 N/m and a nominal tip radius of 20 nm were used in depth-resolved stiffness map and stiffness tomography experiments.

SCANASYST-FLUID+ cantilevers (Bruker, USA) with a nominal spring constant of 0.35 N/m and a nominal tip radius of 2 nm were used for stiffness measurements of the intracellular content. AFM height images were first collected in QI mode with a minimal force setpoint (0.5 nN). Then, multiple force spectroscopies were collected on the designated area using the Contact mode. The force setpoint, the ramp size, and the tip velocity were set to 10 nN, 2 μm, and 2 μm/s, respectively. For each position, $8 \times 8$ force curves were recorded on a $50 \times 50$-nm$^2$ area. The spring constant and deflection sensitivity were calibrated prior to each experiment using the thermal noise method.

### Atomic force microscopy data analysis

The depth-resolved stiffness map data were analyzed using the JPK NanoWizard V data processing software (Bruker, USA). The slope fit function in JPK NanoWizard V data processing software was applied. The fit range was 100% to 80% of the y channel (vertical deflection), which corresponds to 40-nm indentation depth from the end point of the curve. The stiffness tomography was generated using a customized Python code (AFM_stiffness_tomography_analysis.py), in which every 20-nm indentation segment after the contact point was analyzed for all force-distance curves along the cellular medial axis. The medial axis was extracted using code (Bivariate_medial_axis_estimation.py) from a previous study (Papagiannakis et al, 2025). The intracellular stiffness data were analyzed using a customized Python code (AFM_penetration_experiment_analysis.py), in which membrane rupture events were identified, and the slopes of the force-distance curves (the approach curves) were fitted after the rupture points. Only force-distance curves with membrane rupture events were analyzed.

## Data availability

Image analysis code and the newly trained segmentation models for *E. coli* cells imaged on agarose pads and in the microfluidic device are available on the Jacobs-Wagner lab's Github repository (https://github.com/JacobsWagnerLab/published/tree/master/Thappeta_Canas-Duarte_et_al_2025), along with the code for data curation and training set generation. Images are available on Biostudies S-BIAD2088.

The source data of this paper are collected in the following database record: biostudies:S-SCDT-10_1038-S44318-025-00621-y.

## Peer review information

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

## Acknowledgements

We are grateful to Dr. Johan Paulsson and members of the Paulsson lab (Dr. Lei Sun and Carlos Sanchez) for sharing unpublished strains, the epoxy cast of the microfluidic device, and the microfluidic centrifuge holder design used in this study. We are also thankful to Dr. Somenath Bakshi for supervising Georgeos Hardo in generating the simulated bacterial images used to retrain Omnipose. We thank Drs. Suckjoon Jun and KC Huang for sharing published strains, Dr. Sangjin Kim for generating a P1 transduction that generated strain CJW5685, and Dr. Nadia Makarova for technical help with the AFM. We are thankful to Drs. Onn Brandman, KC Huang, and Jonas Cremer for valuable discussion. Finally, we express our gratitude to the Jacobs-Wagner laboratory for support, discussion, and critical reading of the manuscript. Portions of this work were originally part of the dissertation of one of the co-first authors (YT). Part of this work was performed at the Stanford Nano Shared Facilities (SNSF), supported by the National Science Foundation under award ECCS-2026822. AFM experiments were performed at the Stanford University Cell Sciences Imaging Core Facility (RRID:SCR_017787). This research was supported in part by the Netherlands Organization for Scientific Research (Nederlandse Organisatie voor Wetenschappelijk Onderzoek, NWO), Rubicon 2022-2 Science programme, 019.222EN.001 (to AF), the Biotechnology and Biological Sciences Research Council (BB/M011194/1 to GH), the Strategic Seed Fund from the Graduate School of Technology at the University of Cambridge (to GH), and the National Institutes of Health grant R01GM117278 (to LC). CJ-W is an investigator of the Howard Hughes Medical Institute.

## Author contributions

**Yashna Thappeta**: Conceptualization; Data curation; Formal analysis; Validation; Investigation; Visualization; Methodology; Writing—original draft; Writing—review and editing; Prepared the manuscript, conceptualized the study, designed and performed in vivo experiments, and analyzed data. **Silvia J Cañas-Duarte**: Conceptualization; Data curation; Software; Formal analysis; Validation; Investigation; Visualization; Methodology; Writing—original draft; Writing—review and editing; Prepared the manuscript, conceptualized the study, designed and performed in vivo and in vitro experiments, constructed the glycogen biosensor, developed the analysis pipeline for light microscopy, performed and analyzed microfluidic experiments, optimized the Omnipose segmentation models, and analyzed data. **Haozhen Wang**: Conceptualization; Data curation; Formal analysis; Investigation; Visualization; Methodology; Writing—review and editing; Designed, performed and analyzed the AFM experiments. **Till Kallem**: Investigation; Visualization; Methodology; Writing—original draft; Writing—review and editing; Performed and analyzed whole-cell NMR

experiments. **Alessio Fragasso**: Data curation; Software; Visualization; Methodology; Writing—review and editing; Optimized the Omnipose segmentation models for analysis. **Yingjie Xiang**: Software; Formal analysis; Writing—review and editing; Developed the analysis pipeline for the light microscopy experiments. **William Gray**: Investigation; Writing—review and editing; Collected preliminary observations and data. **Cheyenne Lee**: Investigation; Writing—review and editing; Collected preliminary observations and data. **Georgeos Hardo**: Software; Methodology; Writing—review and editing; Produced the synthetic micrograph training data for the microfluidic experiments. **Lynette Cegelski**: Resources; Formal analysis; Supervision; Funding acquisition; Writing—original draft; Project administration; Writing—review and editing; Designed, performed and analyzed whole-cell NMR experiments, provided supervision, and acquired funding. **Christine Jacobs-Wagner**: Conceptualization; Resources; Supervision; Funding acquisition; Visualization; Writing—original draft; Project administration; Writing—review and editing; Prepared the manuscript, conceptualized the study, designed in vivo and in vitro experiments, provided supervision, acquired funding, and managed the project.

Source data underlying figure panels in this paper may have individual authorship assigned. Where available, figure panel/source data authorship is listed in the following database record: biostudies:S-SCDT-10_1038-S44318-025-00621-y.

## Disclosure and competing interests statement

The authors declare no competing interests.

# Expanded View Figures

**Figure EV1.   Localization of various cytoplasmic probes in cells in transition phase.**

(**A**) Representative fluorescence images of DAPI-stained *E. coli* cells expressing different ribosomal protein fusions in transition phase: RplA-GFP (CJW4677), RplA-msfGFP (CJW7020), and RpsB-msfGFP (CJW7021). Arrowheads indicate cell areas of ribosome signal depletion. Scale bar: 2 μm. (**B**) Representative fluorescence image of DAPI-stained *E. coli* cells (CJW7006) expressing mScarlet-I in transition phase. Arrowheads indicate cell areas of mScarlet-I depletion. Scale bar: 2 μm. (**C**) Representative fluorescence images of DAPI-stained *E. coli* expressing RplA-mCherry as well as free fluorescent GFP variants with the following net surface charges (strain name): -30 (CJW7485), -7 (CJW7486), 0 (CJW7487), +7 (CJW7488), +11a (CJW7489), +11b (CJW7490), +15 (CJW7491), and +25 (CJW7492). For GFP with a net surface charge +11, 'a' and 'b' refer to variations in the distribution of the charge on the protein surface (Schavemaker et al, 2017). Cells were grown in M9gluCAAT and supplemented with 0.4% arabinose to induce GFP expression. Arrowheads indicate cell areas depleted of the GFP variant. Scale bar: 2 μm. (**D**) Fluorescence images of representative DAPI- and RNASelect-labeled *E. coli* cells (CJW7324) in exponential (OD ~ 0.30) and transition (OD ~ 1.72) phase. Also shown are the corresponding fluorescence signal profiles for the indicated (*) cells. Arrowheads indicate areas depleted of RNASelect signal. Scale bar: 2 μm. Source data are available online for this figure.

▶

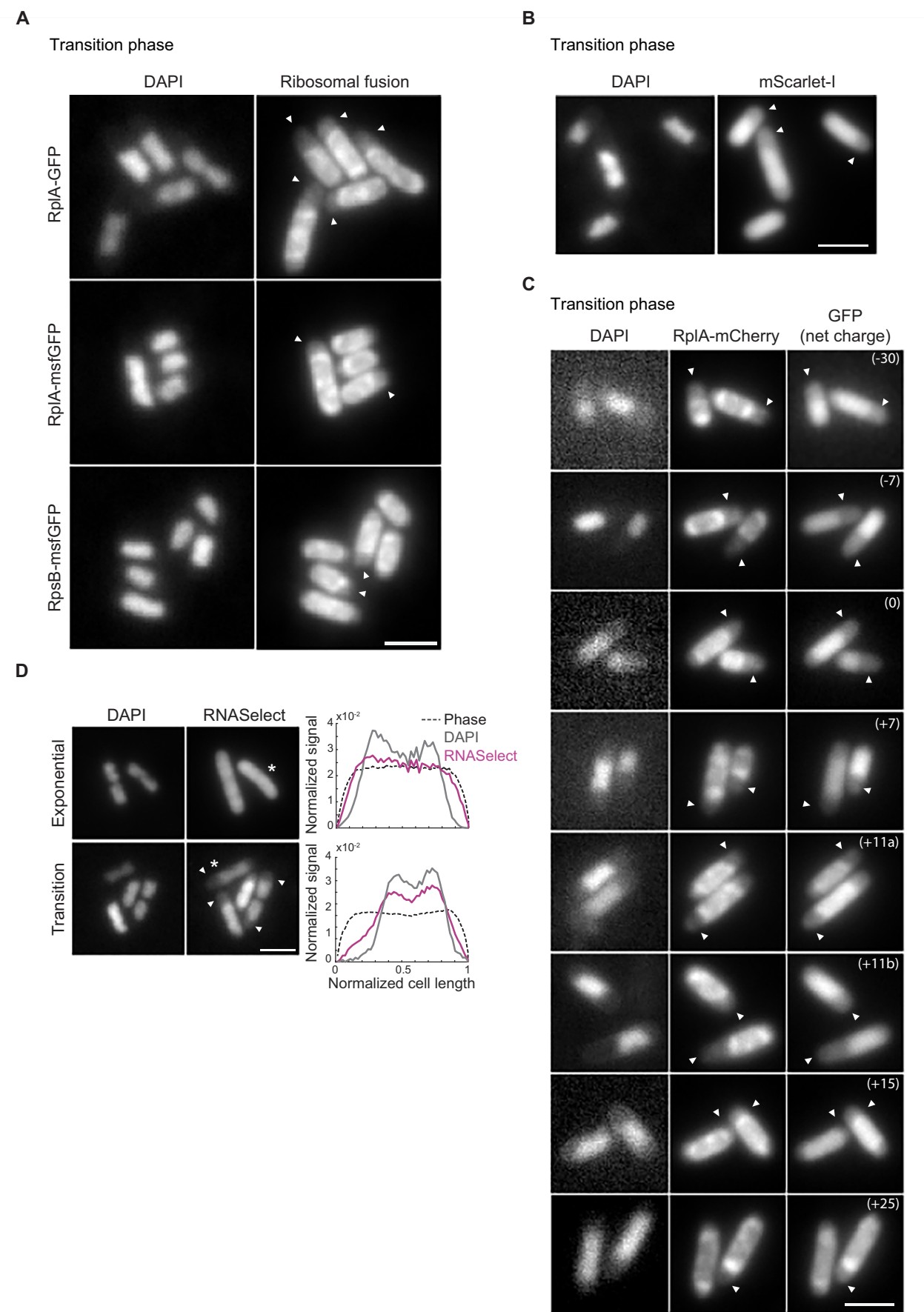

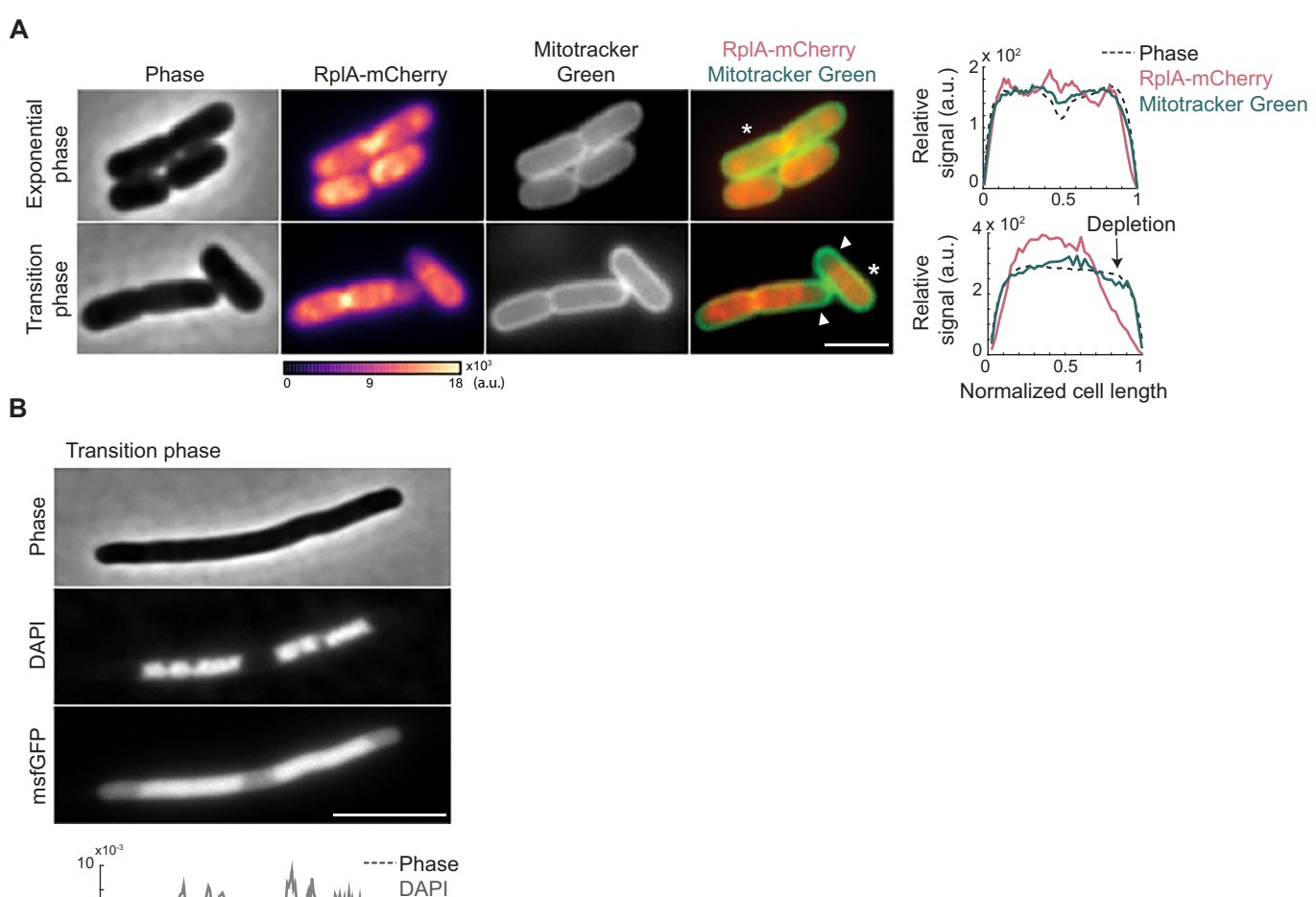

**Figure EV2.   Assessment of the potential effects of membrane retraction and cell division on the localization of cytoplasmic probes in transition-phase cells.**

(**A**) Microscopy images of MitoTracker Green-labeled cells (CJW7324) expressing RplA-mCherry. The samples were obtained from cultures in either exponential or transition phase. Fluorescence intensities are indicated in arbitrary units (a.u.). Signal intensity profiles are provided for the cells indicated by asterisks. White arrowheads show the depletion of RplA-mCherry signal at a cell pole. (**B**) Representative microscopy images of DAPI-stained FtsZ-depleted cells (CJW7588) expressing cytoplasmic msfGFP in transition phase in M9gluCAAT supplemented with 0.4% arabinose to induce the CRISPRi system, thereby blocking the expression of *ftsZ*. The corresponding signal intensity profile is shown below. Scale bar: 5 μm. Source data are available online for this figure.

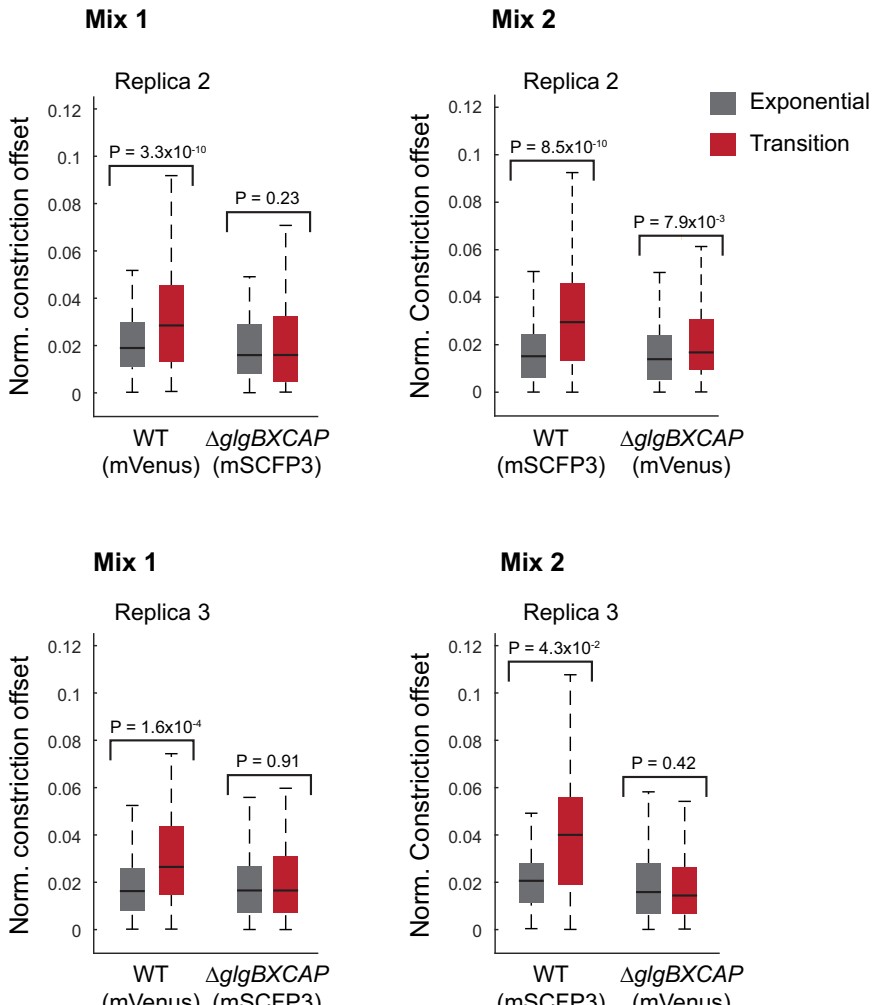

**Figure EV3.  Normalized constriction offset measurements for the WT and Δ*glgBXCAP* cells from co-culture experiments.**

Boxplots of the normalized constriction offset in exponential and transition phases for the biological replicates of the co-culture experiments shown in Fig. 3C. The horizontal lines in the boxes correspond to the medians, with the bottom and top of the boxes showing the 25th and 75th percentiles, respectively. The endpoints of the whiskers mark the minimum and maximum values within a range that excludes the outliers. Outlier values are defined as those more than 1.5 times the interquartile range away from the bottom or top of the boxes. The indicated P values were obtained using a two-sided Wilcoxon rank sum test. Source data are available online for this figure.

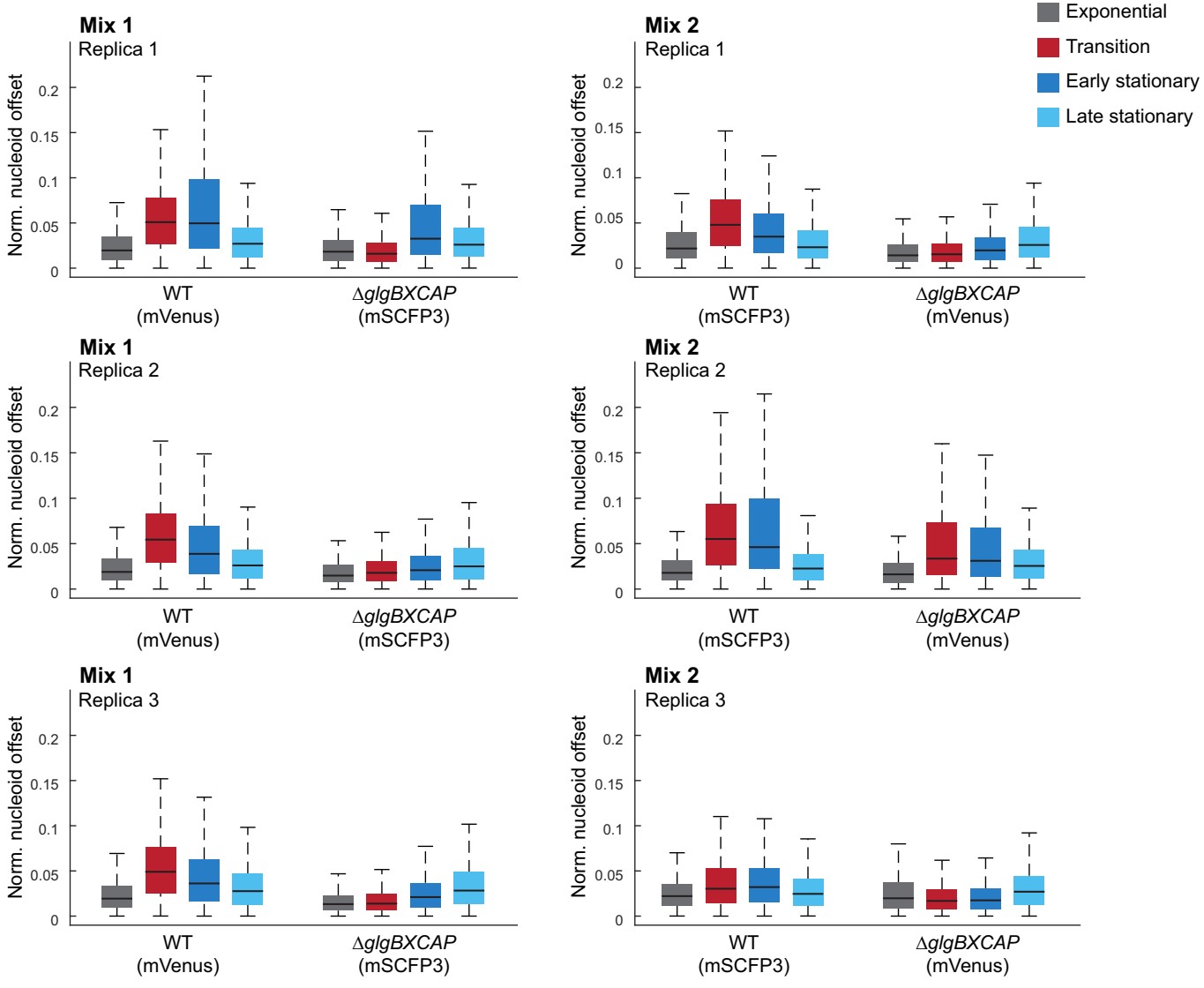

**Figure EV4. Normalized nucleoid offset measurements for the WT and Δ*glgBXCAP* cells from the co-culture experiments across different growth phases.**

Boxplots of the normalized nucleoid offset in exponential, transition, early stationary (24 h), and late stationary phase (72 h) for all the biological replicates of the co-culture experiments. Exponential and transition phase datapoints for Replica 1 of both mixes are shown in Fig. 3D. The horizontal lines in the boxes correspond to the medians, with the bottom and top of the boxes showing the 25th and 75th percentiles, respectively. The endpoints of the whiskers mark the minimum and maximum values within a range that excludes the outliers. Outlier values are defined as those more than 1.5 times the interquartile range away from the bottom or top of the boxes. The indicated P values were obtained using a two-sided Wilcoxon rank-sum test. Source data are available online for this figure.

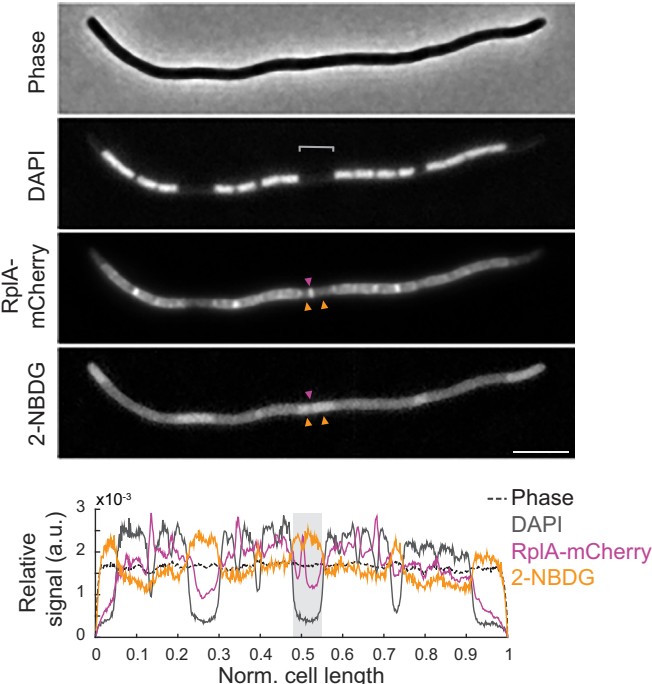

**Transition phase**

**Figure EV5. Representative image of 2-NBDG incorporation in a filamentous polynucleoid cell in transition phase.**

Microscopy images of a cephalexin-treated cell (CJW7324) expressing RplA-mCherry in transition phase (OD ~ 2.0). Yellow arrowheads indicate accumulations of 2-NBDG that sandwich an accumulation of RplA-mCherry signal. Below is the cell signal intensity profile, with the gray shade highlighting the region of interest. Scale bar: 5 μm. Source data are available online for this figure.

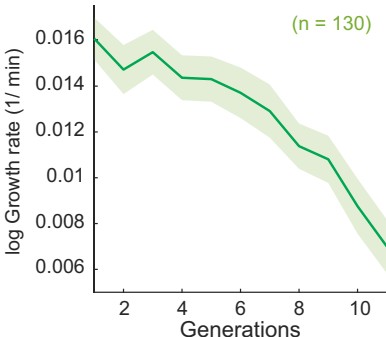

**Figure EV6. Growth rate measurements in the microfluidic device.**

Plot of the calculated log growth rate as a function of cell generations for the strain CJW7605 ($n = 130$ lineages). The solid line and shaded region correspond to the average and the 95% confidence interval, respectively. Source data are available online for this figure.

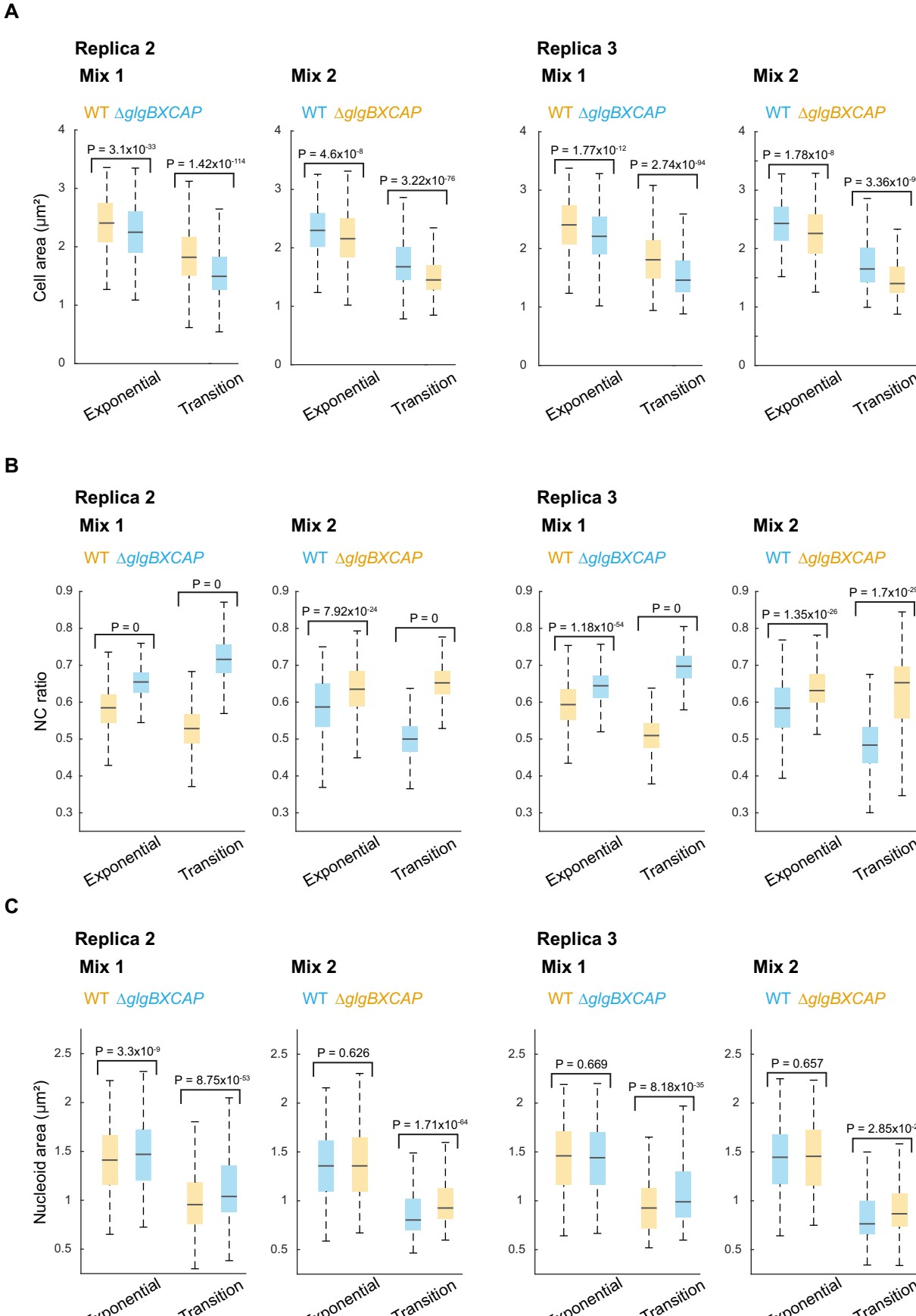

◀  **Figure EV7.  Cell area, NC ratio and nucleoid area measurements for the WT and Δ*glgBXCAP* cells from co-culture experiments.**

(A) Boxplots of the cell area of exponential and transition-phase cells for the biological replicates of the co-culture experiments shown in Fig. 5. The horizontal lines in the boxes correspond to the medians, with the bottom and top of the boxes showing the 25th and 75th percentiles, respectively. The endpoints of the whiskers mark the minimum and maximum values within a range that excludes the outliers. Outlier values are defined as those more than 1.5 times the interquartile range away from the bottom or top of the boxes. The indicated *P* values were obtained using a two-sided Wilcoxon rank-sum test. (B) Same as (A) but for the NC ratio. (C) Same as (A) but for the total nucleoid area. Source data are available online for this figure.

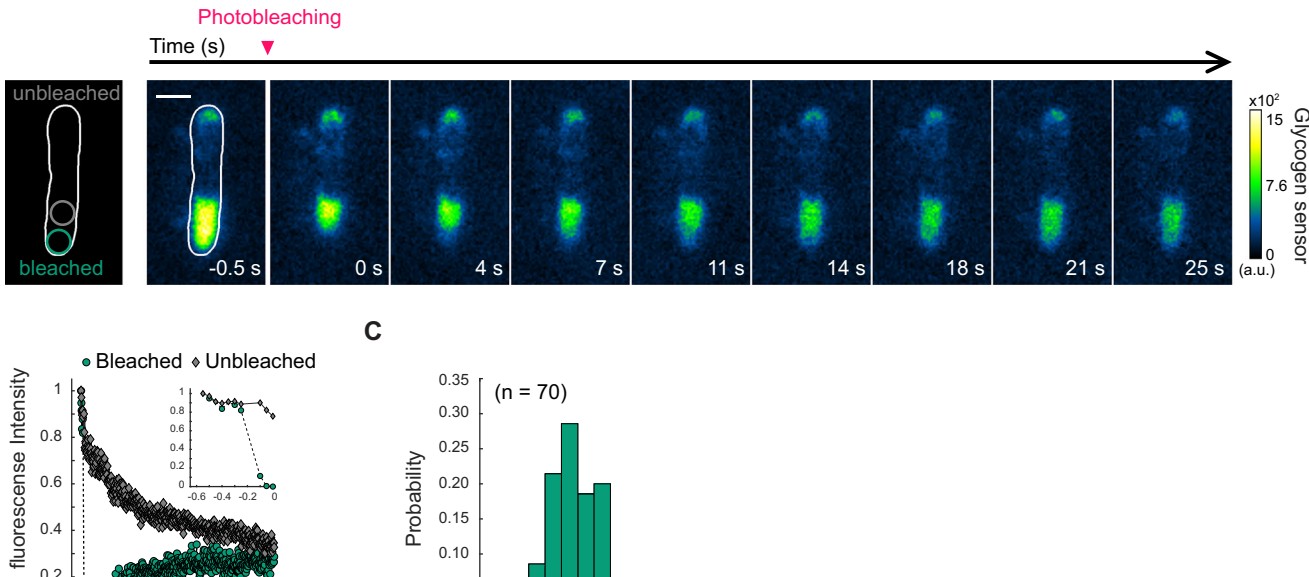

**A**

Photobleaching

Time (s)

unbleached / bleached

-0.5 s  0 s  4 s  7 s  11 s  14 s  18 s  21 s  25 s

x10² 15 / 7.6 / 0 (a.u.) Glycogen sensor

**B**

● Bleached  ◇ Unbleached

Norm. fluorescense Intensity

Time since photobleaching (s)

**C**

(n = 70)

Probability

Half-max recovery time (s)

**Figure EV8. FRAP measurements of the fluorescent glycogen sensor dynamics in glycogen-producing cells in transition phase.**

(A) Time-lapse images of a glycogen-producing cell in transition phase before and after photobleaching a region of the cell pole with the larger glycogen sensor accumulation. The first image on the left shows the time frame before photobleaching, while the subsequent images represent frames captured after photobleaching at the indicated time. The schematic shows the region that was photobleached. Scale bar: 1 μm. (B) Plot showing the evolution of the normalized fluorescence intensity of the glycogen sensor for the unbleached region and the photobleached region of the cell shown in (A), before and after photobleaching. The inset shows the data before and during photobleaching. (C) Histogram of the half-max fluorescence recovery times calculated for 70 cells in which FRAP measurements were obtained. Source data are available online for this figure.

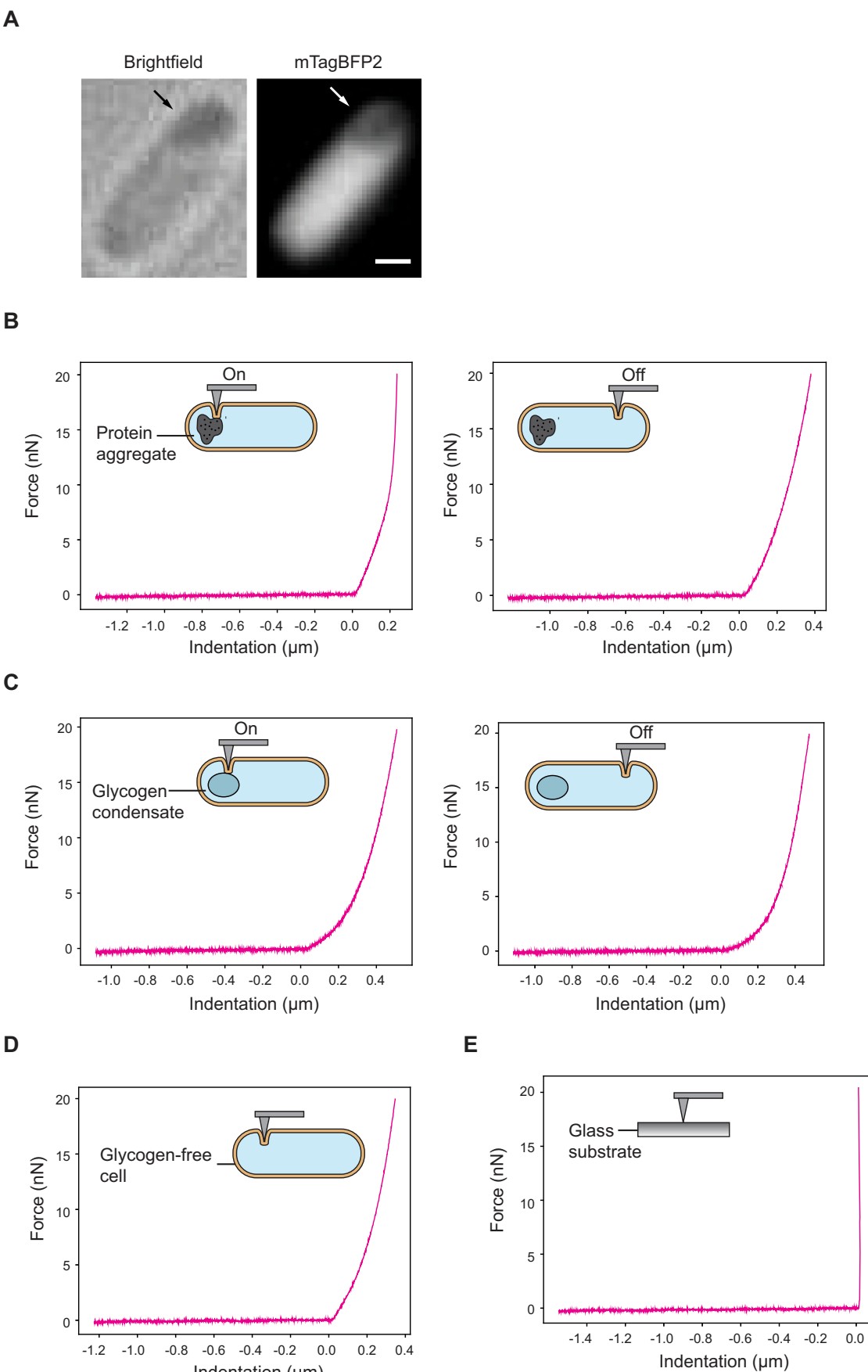

**Figure EV9.  Force-distance curve examples of indentation-based AFM experiments.**

The experimental conditions and strains are the same as in Fig. 7A–C. (**A**) Brightfield image of the protein aggregate-containing cell illustrated in Fig. 7B. The accompanying fluorescence image, which is a duplication of the image in Fig. 7B, is shown for comparison. (**B**) Representative force-distance curves of a cell at regions "On" and "Off" the protein aggregates. (**C**) Same as (**A**) but for a cell with glycogen condensates. (**D**) Same as (**A**) but for a glycogen-free cell. (**E**) Representative force-distance curve on the glass substrate is shown as a control. Source data are available online for this figure.

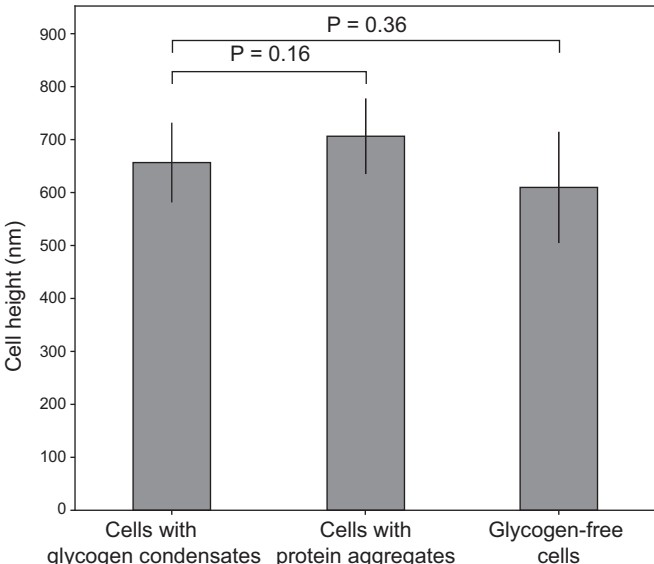

**Figure EV10.  Cell height measurements by AFM microscopy.**

The experimental conditions and strains are the same as in Fig. 7A–C. Plot showing cell height measurements of 21 cells with protein aggregates, 7 cells with glycogen condensates, and 7 cells without glycogen. The cell height information was analyzed from AFM height images collected from more than three biological replicates. Displayed here are the mean values ± the standard deviations. Statistical comparisons were performed using an unpaired two-tailed Student's *t* test. Source data are available online for this figure.

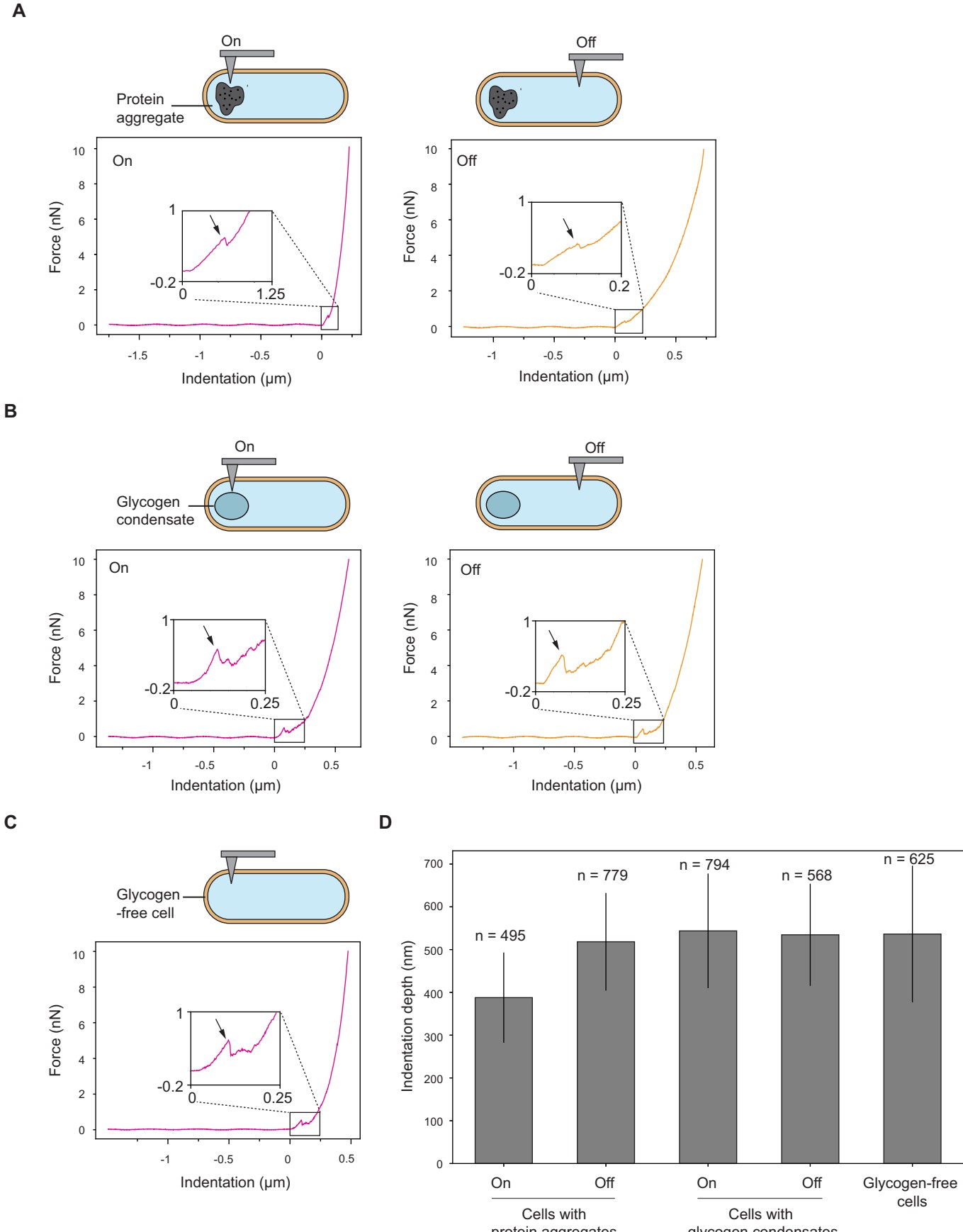

Figure EV11.   Force-distance curve examples of penetration-based experiments.

The experimental conditions and strains are the same as in Fig. 7D,E. Representative force-distance curves showing membrane puncture events (indicated by the arrow) in the figure inset. (A) Representative force-distance curves of a cell at regions "On" and "Off" the protein aggregates. (B) Same as (A) but for a cell with glycogen condensates. (C) Same as (A) but for a glycogen-free cell. (D) Plot showing the indentation depth distributions of 15 cells with protein aggregates, 16 cells with glycogen condensates, and 13 cells without glycogen. Shown are the mean values ± the standard deviations for the total number of indentation events (technical replicates) indicated by the n value. The indentation events were collected from more than three biological replicates. Source data are available online for this figure.

