## [Peer Review File · The EMBO Journal]

Glycogen phase-separation drives macromolecular rearrangement and asymmetric division in *E. coli*

Yashna Thappeta, Silvia Cañas-Duarte, Haozhen Wang, Till Kallem, Alessio Fragasso, Yingjie Xiang, William Gray, Cheyenne Lee, Georgeos Hardo, Lynette Cegelski, and Christine Jacobs-Wagner

Corresponding author: Christine Jacobs-Wagner (jacobs-wagner@stanford.edu)

Review Timeline:

Submission Date:	19th Apr 24
Editorial Decision:	21st Jun 24
Revision Received:	5th Jul 25
Editorial Decision:	29th Aug 25
Revision Received:	3rd Oct 25
Accepted:	17th Oct 25

Editor: Ieva Gailite

Transaction Report:

Dear Dr. Jacobs-Wagner,

Thank you for submitting your manuscript for consideration by the EMBO Journal. I sincerely apologise for the protracted assessment process due to delays in referee report submission. We have now received comments from four reviewers, which are included below for your information.

As you will see from the reports, while reviewer #4 is more critical in their assessment, reviewers #1-3 find the proposed model of asymmetric glycogen condensate accumulation-induced cell division asymmetry interesting. However, all reviewers also indicate several overlapping concerns that would need to be addressed before they can support publication here. In particular, they find that further analysis of the glycogen condensate properties in bacterial cells would be needed, and ask for further analysis on glycogen and nucleoid positioning behaviour in the stationary phase. Furthermore, they indicate that an experiment inhibiting glycogen biosynthesis would be needed to solidify the proposed causal contribution of glycogen condensates to cell asymmetry in transition phase. Finally, they ask for further clarification for the observed asymmetric glycogen cluster distribution and the proposed preferential association with the old pole is needed.

Based on the interest expressed in the reports of reviewers #1-3, I would like to invite you to address these concerns in a revised version of the manuscript. I think it would be helpful to discuss the revision in more detail via email or phone/videoconferencing - please let me know which option you prefer. I should also add that it is The EMBO Journal policy to allow only a single major round of revision and that it is therefore important to resolve the main concerns at this stage.

We generally allow three months as standard revision time, which can be extended to six months in the case of major revisions. Should you foresee a problem in meeting this deadline, please let us know in advance to discuss an extension. As a matter of policy, competing manuscripts published during this period will not negatively impact on our assessment of the conceptual advance presented by your study. However, please contact me as soon as possible upon publication of any related work to discuss the appropriate course of action.

When preparing your letter of response to the referees' comments, please bear in mind that this will form part of the Review Process File and will therefore be available online to the community. For more details on our Transparent Editorial Process, please visit our website: <https://www.embopress.org/page/journal/14602075/authorguide#transparentprocess>. Please also see the attached instructions for further guidelines on preparation of the revised manuscript.

Please feel free to contact me if you have any further questions regarding the revision. Thank you for the opportunity to consider your work for publication. I look forward to discussing your revision.

With best regards,

Ieva

We realize that it is difficult to revise to a specific deadline. In the interest of protecting the conceptual advance provided by the work, we recommend a revision within 3 months (19th Sep 2024). Please discuss the revision progress ahead of this time with the editor if you require more time to complete the revisions.

Referee #1:

In this submitted work, Thappeta et al. reported that the position of nucleoids and cell division sites becomes increasingly asymmetric during the transition phase, and later the authors revealed that the preferential accumulation of the storage polymer glycogen at the old cell pole leads to the observed rearrangements and asymmetric divisions. This is a very interesting study although some technical concerns need to be addressed before the paper can be accepted for publication.

Major concerns:

1) The experiments performed in this work have demonstrated that accumulation of glycogen near the old cell pole leads to the observed asymmetry in cell division. However, the conclusion that the formation of glycogen aggregates is through LLPS is premature. Importantly, in vitro experiments can not replace in vivo experiments since many proteins or cytoplasmic components can form liquid droplets in vitro when their concentrations are high. The authors should try some single-molecule fluorescence tracking or FRAP (fluorescence recovery after photobleaching) on glycogen condensates formed in live *E. coli* cells to figure out whether these condensates are in liquid state, solid state, or gel state. Similar techniques have been widely used in studying LLPS in bacteria.

2) A central question remains unanswered: what happens to those glycogen condensates when *E. coli* cells proceed to the stationary phase? Looking at Figure EV1, it is true that the nucleoid (stained by DAPI) position is more asymmetric in the transition phase than that in the exponential phase. However, such asymmetry is partially restored when the cells enter the stationary phase. And this is common sense that the nucleoid position is not very asymmetric in stationary phase *E. coli* cells. It is difficult to understand this recovery in asymmetry because the stationary phase cells are offsprings of the transition phase cells. If the glycogen condensates are easily formed and attached to the old cell pole during the transition phase, then one should expect to see many stationary phase cells with asymmetric nucleoids, unless the glycogen condensates somehow disappear along with the prolonged culturing time. Can the authors explain?

3) Figure display, Figure 1 is very unimpressive. The authors need to move Figure 2A to Figure 1 to give an intuitive picture of the experimental observation to the readers. Can the authors elaborate why the data error bars are super big for OD=2.5 in Figure 1F and Figure 1G?

Referee #2:

Thappeta and Canas-Durate et al. present evidence of glycogen condensation in *E. coli* cells transitioning from exponential to stationary phase. The concept that glycogen condensation can affect cellular organization during this transition is intriguing; however, the manuscript lacks several critical controls to substantiate the claims.

1. The manuscript would benefit from data showing glycogen localization in exponential, transition, and stationary phases. In Figure EV1, the chromosome appears centralized in the stationary phase, similar to its localization in exponential growth. Does

this imply that glycogen condensates disassemble upon entering stationary phase? Clarification is needed. Additionally, it is sometimes challenging to discern which phase is being discussed, for example, in Figure 4B. Please provide visual cues as part of the figures.

2. The association between larger glycogen clusters and nucleoid offset (lines 253-254) does not necessarily imply a mechanism where glycogen clusters displace the nucleoid. Furthermore, the data from cephalixin-treated cells do not add new insight to support the hypothesis that glycogen drives the transition-phase changes. This claim should either be substantiated with additional experiments or revised. One way to provide more direct evidence could be to examine nucleoid localization in a glycogen-deficient strain (Δ glgBXCAP). Does the chromosome span the entire cell in the Δ glgBXCAP strain?

3. The reason for the formation of one large and one small polar focus of glycogen needs clarification.

4. While the formation of condensates indicates a phase transition, it does not necessarily imply liquid-liquid phase separation. Observing many non-spherical condensates 30 minutes post-PEG addition might suggest a gel-like nature. Please include FRAP experiments to assess the material state and provide statistics on the condensate aspect ratio at various time points. Furthermore, since these condensates adopt different material states in vitro, in-cell FRAP data are necessary to determine the relevant state in vivo.

5. In Figures 2-4, please provide statistics across multiple cells instead of showcasing individual examples.

Referee #3:

The manuscript "Glycogen phase separation drives macromolecular rearrangement and asymmetric division in *E. coli*" Thappeta, Jacobs-Wagner and colleagues describes the effect of glycogen storage polymers on the positioning of cell division in *E. coli*.

The authors show that *E. coli* cells accumulate glycogen storage condensates in an asymmetric fashion close to the old pole (known before). They then observed that the subsequent division in these cells becomes asymmetric leading to daughter cells with unequal distribution of cytoplasmic content. With a series of in vitro experiments the authors show elegantly that glycogen undergoes phase separation in an environment that mimics the cytoplasmic content. Above a threshold the liquid condensates solidify into a rigid storage granule.

Accumulation of glycogen condensates leads to an increase in cell size and displacement of the bacterial nucleoid, eventually leading to the observed asymmetric division.

The paper is very well written and guides the reader through the different experiments that nicely build up. The experimental part is executed with great care and appropriate techniques leaving little to criticize. There are a few minor points that may be useful to address prior publication.

1) The authors discuss the dependence on the Min and nucleoid occlusion systems for precise midcell division. Therefore, it of course raises the questions how these machineries, in particular the Min system, are working in cells with large glycogen condensates. Technically, in cells with glycogen condensates, the Min oscillation should not be able to localize the division plane, but SlmA might play a major role in these cases. I do not think this is absolutely required for this work, but seeing the Min oscillation or the effect of an *slmA* knockout would clarify what dictates division plane in these asymmetric dividing cells. This would corroborate the author's claims made in lines 413-417 of the discussion.

2) What happens to cells when glycogen storage is depleted - I assume they switch back to normal division. Or is the cell with the excess condensate eventually dying (compare aging effect with the Boehm/Ackermann paper).

3) It is astonishing that the glycogen accumulation is asymmetric, despite the fact that condensates are seen on both poles. What makes the condensate enrich at one (the old) pole? How did the authors identify the old pole?

4) Line 45: The sentence reads as if bacteria do not have membrane-bound organelles, which is not true. Consider rephrasing

5) Lines 90-91. Please describe which nutritional element becomes limiting under the chosen growth conditions. *E. coli* produces glycogen when carbon is available in excess, but other elements become limiting.

6) Line 125: Briefly mention the function of RplA when it is mentioned first time.

7) Line 282: The original name for LB is lysogeny broth and not Luria broth.

Referee #4:

The manuscript by Thappeta et al. provides a close look at various cellular characteristics that change during the transition from exponential to stationary phase, defined by the authors as "transition phase", at a single cell, as well as a population, level. An interesting finding was that the position of the cell constriction site, which is symmetric during exponential phase (EP), became asymmetric in the transition phase (TP), and was accompanied by an increase in asymmetric nucleoid positioning. While the

asymmetry in nucleoid positioning during stationary phase (SP) was known, the association with the shift in position of the cell constriction site during the TP is apparently new.

The authors went on to show that during the TP, the cells exhibit asymmetric distribution of ribosomes, RNAs and proteins. In search for the element that drives the observed intracellular reorganization during the TP, the authors focus on a specific increase in carbon intensity in the TP compared to the EP by NMR spectra. The authors define the band that increased in the TP as "characteristic to polysaccharide contributions" and speculate that it is glycogen.

From this point and on, the weak part of the MS starts. The proofs for glycogen being the driving force for the rearrangements during the TP are circumstantial and indirect, to say the least. The main problem of the MS is that the authors chose to present this as fait accompli and as their main finding, and even to phrase the title accordingly. In the text, the authors are more cautious, using the words suggest and likely in the Abstract, and predicting that glycogen accumulation is, at least in part, responsible for the asymmetric positioning. However, I do not think that they proved their prediction, nor can I understand how this can be presented as proven and as the main message. Hence, I am afraid that the MS cannot be published in its present form.

Major points:

1. As written above, the main claim of the MS, or what the authors chose as their main finding, is not proven. The firm conclusions are unjustified and the manuscript does not live up to its title.
2. Line 246 and in other places: Where is the proof that glycogen accumulates at the old pole? This might seem reasonable to assume, but has not been proven, for example by a microfluidic "mother machine".
3. Lines 244-254: How does this correlation prove the prediction that glycogen accumulation drives asymmetric positioning of the nucleoid?
4. Line 255: In accord with the previous point, not only is this not "Another line of support for glycogen driving the transition-phase phenotype", as there is no previous proof, but also the results reported in this paragraph do not prove the causality.
5. The proof that glycogen phase separates in vitro is kind of expected. The authors went through some hardships to show that "glycogen forms liquid condensates under in vitro conditions that mimic the E. coli cytoplasm". However, the size of the droplets that they managed to form exceeds the size of an E. coli cell!! Of note, the scale bar for most images in Fig. 5 is 10 μ m (the average size of an E. coli cell is 1-1.5 μ m). Indeed, the authors go on to show that glycogen condensates are associated with increased cell size, but this increase is very modest relative to the size of the droplets.
6. Lines 339-340: I do not see why Fig. 5I is the proof for the exclusion of fluorescent proteins by glycogen condensates
7. Proofs for glycogen phase separation in vivo are totally missing. Why didn't the authors use some conventional methods, such as addition of hexanediol (not a proof by itself, but still...) FRAP, to show reversible formation of glycogen condensates in vivo?
8. What is also missing is the causality. Even if glycogen phase separates in vivo, what is the proof that this causes the rearrangements and asymmetry during the TP?
9. To go one step further, what is the proof that glycogen is "the" driver of all that, as the Title implies. It could still be one of many factors.

Less major points:

1. The proofs for the asymmetric distribution of RNAs and proteins could be supported by additional experimentation. For example, by imaging the distribution of specific E. coli proteins and RNAs that were shown to be distributed through the cytoplasm.
2. The exact material state of the glycogen droplets has not been determined even in vitro. The authors acknowledge this fact with regard to the in vivo, where the glycogen granules were not even shown to be phase-separated condensates. Hence, they should refrain from using the term LLPS. Rather, they should call it PS.

Minor points:

1. Line 20: Erase "in nature and the laboratory"
2. Line 30: suggest and likely are redundant. Omit one of them
3. Fig. 1A: Why are the OD units on the Y axis not given in the same scale used in the text for OD600?
4. Lines 128-130: Did the author mean to say that "The latter is consistent" rather than "was consistent" (which implies that they have shown it somehow)?

We thank the editor and reviewers for their time and helpful feedback. We have revised the manuscript to address all comments. Please see below for a point-by-point response (in blue).

Editor:

As you will see from the reports, while reviewer #4 is more critical in their assessment, reviewers #1-3 find the proposed model of asymmetric glycogen condensate accumulation-induced cell division asymmetry interesting. However, all reviewers also indicate several overlapping concerns that would need to be addressed before they can support publication here.

In particular, they find that further analysis of the glycogen condensate properties in bacterial cells would be needed, and ask for further analysis on glycogen and nucleoid positioning behavior in the stationary phase.

Response 1: We tried to explain in the original submission that demonstrating the phase of the glycogen condensates in living bacterial cells is technically challenging. Phase separation in bacteria has been described for proteins for which it is straightforward to generate covalently labeled versions by creating fluorescent protein fusions. Such fusions allow for the implementation of single-molecule or FRAP experiments, such as those requested by the reviewers. This is not the case for polysaccharides like glycogen. Using a protein that non-covalently binds to glycogen, such as our glycogen sensor, is problematic for the implementation of single-molecule or FRAP experiments as its dynamics may reflect not only the diffusion of glycogen molecules (its binding partner), but also its association/dissociation kinetics to/from glycogen. Nevertheless, we performed the requested FRAP experiment with the glycogen sensor and showed that the recovery time after photobleaching is fast, as expected for a liquid state. Furthermore, our results are consistent with the time scale previously reported for liquid glycogen condensates in eukaryotic cells (Liu et al, 2021). These new FRAP data are presented in the new Fig EV7. However, we warn the readers about the difficulty of interpreting these results.

Given the caveat with the FRAP measurements, we explored other ways to probe the mechanical properties of glycogen condensates inside *E. coli*. Ultimately, inspired by mechanical measurements on intracellular organelles in eukaryotic cells, we devised a method based on atomic force microscopy (AFM) to measure—for the first time to our knowledge—the stiffness of intracellular content in bacterial cells. We found that regions with glycogen condensates are as soft as other (liquid) cytoplasmic regions, consistent with a liquid state. In contrast, intracellular regions with protein aggregates are much stiffer. These AFM results are described in new Figures 7 and EV8-10.

Response 2: Regarding the nucleoid position in stationary-phase cells, we have added an analysis in Fig 1I and Fig. EV5.

Furthermore, they indicate that an experiment inhibiting glycogen biosynthesis would be needed to solidify the proposed causal contribution of glycogen condensates to cell asymmetry in transition phase.

Response 3: We have carried out multiple experiments and analyses (Fig 3B-F, Fig 4E, Fig 5C-E, and Fig EV4) to demonstrate quantitatively that the cellular asymmetries observed in glycogen-producing cells virtually disappear in mutant cells deficient in glycogen biosynthesis.

Finally, they ask for further clarification for the observed asymmetric glycogen cluster distribution and the proposed preferential association with the old pole is needed.

Response 4: We performed timelapse experiments in a microfluidic device connected to a liquid culture to demonstrate that the gradual accumulation of glycogen at the old pole occurs through inheritance over generations. These experiments also allowed us to visualize the behavior of glycogen condensate and nucleoid positioning simultaneously. Glycogen-producing (WT) and glycogen-devoid (Δ glgBXCAP) cells were directly compared, as they were loaded in the same microfluidic device and experienced the same environment at the same time. Quantitative analysis of this new set of data shows how the nucleoid position offset increases over time and generations with glycogen accumulation. These data are presented in a new figure (Figure 4).

Response 5: The text and figures have been revised to include these additional results and clarify potential confusing points.

Referee #1:

In this submitted work, Thappeta et al. reported that the position of nucleoids and cell division sites becomes increasingly asymmetric during the transition phase, and later the authors revealed that the preferential accumulation of the storage polymer glycogen at the old cell pole leads to the observed rearrangements and asymmetric divisions. This is a very interesting study although some technical concerns need to be addressed before the paper can be accepted for publication.

Response 6: Thank you for your appreciation of our findings and for your helpful comments.

Major concerns:

1) The experiments performed in this work have demonstrated that accumulation of glycogen near the old cell pole leads to the observed asymmetry in cell division. However, the conclusion that the formation of glycogen aggregates is through LLPS is premature. Importantly, in vitro experiments can not replace in vivo experiments since many proteins or cytoplasmic components can form liquid droplets in vitro when their concentrations are high. The authors should try some single-molecule fluorescence tracking or FRAP (fluorescence recovery after photobleaching) on glycogen condensates formed in live *E. coli* cells to figure out whether these condensates are in liquid state, solid state, or gel state. Similar techniques have been widely used in studying LLPS in bacteria.

Response 7: We agree that we had not formally demonstrated the liquid-like state of glycogen condensates inside cells, which we acknowledged in the original submitted version of our manuscript. Examining the diffusivity of glycogen molecules within condensates by single-molecule fluorescence or FRAP microscopy is challenging, as polysaccharides like glycogen cannot be covalently labeled with fluorescent tags inside cells as readily as proteins (through genetic engineering). To address the reviewer's request, we nevertheless performed FRAP experiments on glycogen condensates fluorescently labeled with our glycogen sensor. These experiments, which are presented in new Figure EV7, show that fluorescence recovery after photobleaching is fast. However, because the glycogen sensor binds glycogen non-covalently, reversible association/dissociation to/from glycogen may contribute to the observed fluorescence recovery kinetics. To address this limitation, we sought alternative approaches to assess the mechanical properties of glycogen condensates within *E. coli*. Drawing inspiration from studies on

intracellular organelles in eukaryotic cells, we developed an atomic force microscopy (AFM)-based method to measure—for the first time to our knowledge—the stiffness of intracellular regions in bacterial cells. Our findings reveal that areas containing glycogen condensates are as soft as other (liquid) cytoplasmic regions, consistent with a liquid-like state. In contrast, regions with protein aggregates (thought to be more solid-like) display higher stiffness. These AFM results are presented in the new Figures 7 and EV8-10.

2) A central question remains unanswered: what happens to those glycogen condensates when *E. coli* cells proceed to the stationary phase? Looking at Figure EV1, it is true that the nucleoid (stained by DAPI) position is more asymmetric in the transition phase than that in the exponential phase. However, such asymmetry is partially restored when the cells enter the stationary phase. And this is common sense that the nucleoid position is not very asymmetric in stationary phase *E. coli* cells. It is difficult to understand this recovery in asymmetry because the stationary phase cells are offsprings of the transition phase cells. If the glycogen condensates are easily formed and attached to the old cell pole during the transition phase, then one should expect to see many stationary phase cells with asymmetric nucleoids, unless the glycogen condensates somehow disappear along with the prolonged culturing time. Can the authors explain?

Response 4: Yes, it is well documented that glycogen is consumed by the cells during stationary phase, leading to a reduction in nucleoid position asymmetry. We now show in a new figure (Fig EV5) that the nucleoid position offset increases in transition phase relative to exponential phase and then decreases during stationary phase. In contrast, it remains low in glycogen-deficient (Δ glgBXCAP) cells across culture phases.

In addition, we have revised Figure 1 to better showcase—both qualitatively (with images in Fig 1B) and quantitatively (with probability density plots in Fig 1I)—the distribution of nucleoid position asymmetry in snapshot images of cells in exponential, transition, or stationary phase.

3) Figure display, Figure 1 is very unimpressive. The authors need to move Figure 2A to Figure 1 to give an intuitive picture of the experimental observation to the readers. Can the authors elaborate why the data error bars are super big for OD=2.5 in Figure 1F and Figure 1G?

Response 5: We agree that showing images earlier would help the reader. We are now providing representative images as Figure 1B. Thank you for the suggestion!

When investigating the source of the error bar, we noted that a small fraction of our measurements were obtained from cultures grown in the same medium, except for the inclusion of trace elements, which were absent in the other cultures. While the effect of these trace elements is small, they contributed to a small systematic shift of the parameter values consistent with the trace elements slightly increasing the OD saturation point. We have now removed these data from the dataset and redone the analyses with cells grown under the same conditions. The conclusions are unchanged. The error bar for the OD range between 2.5 and 3 remains substantial, which we attribute to day-to-day variability.

Referee #2:

Thappeta and Canas-Durate et al. present evidence of glycogen condensation in *E. coli* cells transitioning from exponential to stationary phase. The concept that glycogen condensation can affect cellular organization during this transition is intriguing; however, the manuscript lacks several critical controls to substantiate the claims.

Response 6: Thank you for your interest and valuable feedback.

1. The manuscript would benefit from data showing glycogen localization in exponential, transition, and stationary phases. In Figure EV1, the chromosome appears centralized in the stationary phase, similar to its localization in exponential growth. Does this imply that glycogen condensates disassemble upon entering stationary phase? Clarification is needed. Additionally, it is sometimes challenging to discern which phase is being discussed, for example, in Figure 4B. Please provide visual cues as part of the figures.

Response 7: We apologize for the confusion. We have added labels about the relevant growth phase to all the relevant figures for added clarity.

It is well documented that glycogen is consumed during stationary phase, which we now explicitly comment on in the revised text with citations. Consistent with this, we observed that nucleoid positioning becomes less asymmetric during stationary phase (new Fig EV5).

2. The association between larger glycogen clusters and nucleoid offset (lines 253-254) does not necessarily imply a mechanism where glycogen clusters displace the nucleoid. Furthermore, the data from cephalixin-treated cells do not add new insight to support the hypothesis that glycogen drives the transition-phase changes. This claim should either be substantiated with additional experiments or revised. One way to provide more direct evidence could be to examine nucleoid localization in a glycogen-deficient strain (Δ glgBXCAP). Does the chromosome span the entire cell in the Δ glgBXCAP strain?

Response 8: We have carried out multiple experiments and analyses (Fig 3B-F, Fig 4E, Fig 5C-E, and Fig EV4) to demonstrate quantitatively that the cellular asymmetries observed in glycogen-producing cells virtually disappear in the mutant cells deficient in glycogen biosynthesis.

In addition, we performed microfluidic experiments in which both glycogen-producing and glycogen-deficient cells experienced nutrient deprivation at the same time. These experiments show how nucleoid localization becomes gradually more asymmetric with glycogen accumulation over generations, whereas the nucleoid positioning remains unchanged in Δ glgBXCAP cells. This new evidence is now presented in Figure 4.

In addition, we now provide illustrative images and quantifications of nucleoid areas and nucleocytoplasmic (NC) ratios for both glycogen-producing (WT) cells and glycogen-deficient (Δ glgBXCAP) mutants. These data (new Figure 5C-D) show that the nucleoid does indeed span the entire cell in the Δ glgBXCAP strain, primarily because glycogen-deficient cells are smaller than WT cells, which have the 'bonus' space associated with glycogen condensate formation.

3. The reason for the formation of one large and one small polar focus of glycogen needs clarification.

Response 9: Glycogen tends to accumulate at the old cell pole compared to the new pole because of the inheritance of glycogen condensates from the previous division cycle. To substantiate this claim, we have added microfluidic (mother cell machine) experiments in which we tracked glycogen accumulation (using the glycogen sensor) in the mother cells over generations. These experiments show that the glycogen signal increases at the old pole with each generation. This is now shown in illustrative images and through quantification (new Figure 4).

4. While the formation of condensates indicates a phase transition, it does not necessarily imply liquid-liquid phase separation. Observing many non-spherical condensates 30 minutes post-PEG addition might suggest a gel-like nature. Please include FRAP experiments to assess the material state and provide statistics on the condensate aspect ratio at various time points. Furthermore, since these condensates adopt different material states in vitro, in-cell FRAP data are necessary to determine the relevant state in vivo.

Response 10: The images were taken on the surface of the slide to facilitate visualization of droplets within the same focal plane. The drawback is that the interaction of droplets with the glass surface (surface wetting) affects the fusion process, resulting in oval-shaped condensates. We have added a new video (Video EV4) showing that droplets are spherical in the liquid column before and after fusion. The shape becomes less round only when the droplets settle and fuse on the glass surface. We were unable to segment droplets within the liquid column for quantitative image analysis of the aspect ratio, as droplets come in and out of focus due to motion. However, it is clear by visual inspection that the vast majority of the droplets are spherical and not oval, consistent with a liquid state. The text was updated accordingly.

As noted in response 1, we would need glycogen to be covalently labeled with a fluorophore for a FRAP experiment to be conclusive. While covalently tagging proteins with a fluorophore (such as GFP) is straightforward, to our knowledge, no method currently exists for doing so with polysaccharides like glycogen inside cells. Nevertheless, to fulfill the reviewer's request, we performed FRAP experiments on glycogen condensates fluorescently labeled with our glycogen sensor. These experiments, which are presented in the new Figure EV7, show that fluorescence recovery after photobleaching is fast. However, because the glycogen sensor binds glycogen non-covalently, reversible association/dissociation to/from glycogen may contribute to the observed fluorescence recovery kinetics.

Given these caveats, we sought alternative approaches to assess the mechanical properties of glycogen condensates within *E. coli*. Drawing inspiration from studies on intracellular organelles in eukaryotic cells, we developed an atomic force microscopy (AFM)-based method to measure—for the first time to our knowledge—the stiffness of intracellular regions in bacterial cells. Our findings reveal that areas containing glycogen condensates are as soft as other (liquid) cytoplasmic regions, consistent with a liquid-like state. In contrast, regions with protein aggregates (thought to be more solid-like) display higher stiffness. These AFM results are presented in the new Figures 7 and EV8-10.

5. In Figures 2-4, please provide statistics across multiple cells instead of showcasing individual examples.

Response 11: We have added population quantifications and n values.

Referee #3:

The manuscript "Glycogen phase separation drives macromolecular rearrangement and asymmetric division in *E. coli*" Thappeta, Jacobs-Wagner and colleagues describes the effect of glycogen storage polymers on the positioning of cell division in *E. coli*.

The authors show that *E. coli* cells accumulate glycogen storage condensates in an asymmetric fashion close to the old pole (known before). They then observed that the subsequent division in these cells becomes asymmetric leading to daughter cells with unequal distribution of cytoplasmic content. With a series of in vitro experiments the authors show elegantly that glycogen undergoes phase separation in an environment that mimics the cytoplasmic content. Above a threshold the liquid condensates solidify into a rigid storage granule.

Accumulation of glycogen condensates leads to an increase in cell size and displacement of the bacterial nucleoid, eventually leading to the observed asymmetric division.

The paper is very well written and guides the reader through the different experiments that nicely build up. The experimental part is executed with great care and appropriate techniques leaving little to criticize. There are a few minor points that may be useful to address prior publication.

Response 12: Thank you so much for your appreciation of our work.

1) The authors discuss the dependence on the Min and nucleoid occlusion systems for precise midcell division. Therefore, it of course raises the questions how these machineries, in particular the Min system, are working in cells with large glycogen condensates. Technically, in cells with glycogen condensates, the Min oscillation should not be able to localize the division plane, but SlmA might play a major role in these cases. I do not think this is absolutely required for this work, but seeing the Min oscillation or the effect of an *slmA* knockout would clarify what dictates division plane in these asymmetric dividing cells. This would corroborate the author's claims made in lines 413-417 of the discussion.

Response 13: We have added data showing that MinD-GFP continues to oscillate from pole to pole during transition phase (new Video EV2). This suggests that there is cytoplasmic space between glycogen condensates and the cytoplasmic membrane to allow MinD-GFP to diffuse around the glycogen condensates and interact with the membrane at the cell poles.

We did not test a *slmA* gene deletion strain because, to our knowledge, this mutant doesn't have any cell division defect on its own. Also, recent work from our lab indicates that FtsZ ring formation is unlikely to be regulated by nucleoid occlusion (Govers et al, 2024 PMID 38157847). Nucleoid occlusion more likely regulates cell constriction (i.e., constriction of the FtsZ ring rather than its assembly).

2) What happens to cells when glycogen storage is depleted - I assume they switch back to normal division. Or is the cell with the excess condensate eventually dying (compare aging effect with the Boehm/Ackermann paper).

Response 14: The reviewer is correct. It is well documented that glycogen is consumed by cells in stationary phase. Consistent with this depletion, the asymmetry in nucleoid positioning decreases during stationary phase (see revised Fig 1I and new Fig EV5).

3) It is astonishing that the glycogen accumulation is asymmetric, despite the fact that condensates are seen on both poles. What makes the condensate enrich at one (the old) pole? How did the authors identify the old pole?

Response 15: It is correct that glycogen can, in principle, accumulate at either pole, but the gradual accumulation at the old cell pole occurs through inheritance from the previous generations. To substantiate this claim, we performed microfluidic experiments to track “mother” cells (cells at the closed end of the trenches), which, at each division, inherits the old cell pole. These experiments demonstrate that the accumulation of glycogen at the old cell pole keeps increasing with each generation. These experiments also shows that the gradual asymmetry of glycogen accumulation between cell poles correlates with the asymmetry in nucleoid positioning. These data are presented in a new figure (Fig 4).

4) Line 45: The sentence reads as of bacteria do not have membrane-bound organelles, which is not true. Consider rephrasing

Response 16: Good point. It now reads: “While bacteria typically lack membrane-bound organelles in their cytoplasm, ...”.

5) Lines 90-91. Please describe which nutritional element becomes limiting under the chosen growth conditions. E. coli produces glycogen when carbon is available in excess, but other elements become limiting.

Response 17: Identifying the limiting nutrient is not trivial. Importantly, we have added data to show that all the phenotypes observed in transition phase disappear in a glycogen-deficient mutant (Fig 3B-F, Fig 4E, Fig 5C-E, and Fig EV4), demonstrating that they are linked to glycogen. Thus, the exact limiting element shouldn't be important as long as it causes an accumulation of glycogen. However, we agree that it is important to clarify that glycogen accumulates when the carbon source is in excess and a nutritional element such as nitrogen, sulfur, or phosphate becomes limiting. We have clarified this point in the revised paper and included relevant references.

6) Line 125: Briefly mention the function of RplA when it is mentioned first time.

Response 18: Good point. Done.

7) Line 282: The original name for LB is lysogeny broth and not Luria broth.

Response 19: Corrected. Thank you!

Referee #4:

The manuscript by Thappeta et al. provides a close look at various cellular characteristics that change during the transition from exponential to stationary phase, defined by the authors as "transition phase", at a single cell, as well as a population, level. An interesting finding was that the position of the cell constriction site, which is symmetric during exponential phase (EP), became asymmetric in the transition phase (TP), and was accompanied by an increase in asymmetric nucleoid positioning. While the asymmetry in nucleoid positioning during stationary phase (SP) was known, the association with the shift in position of the cell constriction site during the TP is apparently new.

The authors went on to show that during the TP, the cells exhibit asymmetric distribution of ribosomes, RNAs and proteins. In search for the element that drives the observed intracellular reorganization during the TP, the authors focus on a specific increase in carbon intensity in the TP compared to the EP by NMR spectra. The authors define the band that increased in the in the TP as "characteristic to polysaccharide contributions" and speculate that it is glycogen.

From this point and on, the weak part of the MS starts. The proofs for glycogen being the driving force for the rearrangements during the TP are circumstantial and indirect, to say the least. The main problem of the MS is that the authors chose to present this as *fait accompli* and as their main finding, and even to phrase the title accordingly. In the text, the authors are more cautious, using the words suggest and likely in the Abstract, and predicting that glycogen accumulation is, at least in part, responsible for the asymmetric positioning. However, I do not think that they proved their prediction, nor can I understand how this can be presented as proven and as the main message. Hence, I am afraid that the MS cannot be published in its present form.

Response 20: Thank you for the valuable feedback. We clearly did not do a good job of presenting the evidence. We have clarified the text and added experimental data to support our claim.

Major points:

1. As written above, the main claim of the MS, or what the authors chose as their main finding, is not proven. The firm conclusions are unjustified and the manuscript does not live up to its title.

Response 21: We agree that we did not do a good job of presenting the evidence in the original submission. We have added experimental data (Fig 3B-F, Fig 4E, Fig 5C-E, and Fig EV4) and clarified our arguments to further support our claim that the condensation of glycogen is the major driver of the cellular asymmetries that we report. These cellular asymmetries correlate with glycogen accumulation quantitatively and they disappear when cells cannot make glycogen due to the complete deletion of the glycogen biosynthetic operon.

2. Line 246 and in other places: Where is the proof that glycogen accumulates at the old pole? This might seem reasonable to assume, but has not been proven, for example by a microfluidic "mother machine".

Response 22: Great suggestion! We have added mother machine experiments to show that glycogen preferentially accumulate at the old cell pole because the old pole gets a "head start"

relative to the new pole by inheriting glycogen from the previous generation. These data are shown in a new figure (Fig 4).

3. Lines 244-254: How does this correlation prove the prediction that glycogen accumulation drives asymmetric positioning of the nucleoid?

Response 23: In fairness, we wrote that the correlation was consistent with the prediction and not that it proved that glycogen accumulation drives the asymmetric positioning of the nucleoid. But we agree that a correlation is not sufficient. The original manuscript showed that the asymmetry in nucleoid positioning disappears in a glycogen-deficient mutant grown in coculture with the wild-type (glycogen-producing) cells (former Fig 6E). However, we acknowledge that the data were not presented in a clear way. In the revised manuscript, we not only clarified the text, but also added quantitative analyses (Fig 3D and Ev5) and experimental evidence (new Fig 4) to further support our claim.

4. Line 255: In accord with the previous point, not only is this not "Another line of support for glycogen driving the transition-phase phenotype", as there is no previous proof, but also the results reported in this paragraph do not prove the causality.

Response 24: Our goal is to present multiple lines of support, one at a time, which collectively support our claim. At this particular point in the paper, we speak of correlations. But we agree that we needed further experimental data to support a causal link. We have revised the paper to clarify our arguments and added experimental data and quantitative analyses (Fig 3B-F, Fig 4E, Fig 5C-E, and Fig EV4) to substantiate our claim.

5. The proof that glycogen phase separates in vitro is kind of expected. The authors went through some hardships to show that "glycogen forms liquid condensates under in vitro conditions that mimic the E. coli cytoplasm". However, the size of the droplets that they managed to form exceeds the size of an E. coli cell!! Of note, the scale bar for most images in Fig. 5 is 10 μm (the average size of an E. coli cell is 1-1.5 μm^2). Indeed, the authors go on to show that glycogen condensates are associated with increased cell size, but this increase is very modest relative to the size of the droplets.

Response 25: It is common for condensates to become large under in vitro conditions for several reasons. First, they do not experience cellular confinement. Second, inside cells, large glycogen accumulations are excluded from the nucleoid meshwork, further reducing the free available space for expansion. Third, we use physiological concentration, but not amount, of glycogen in vitro because the volume difference between a cell (in vivo) and 50 μl in an Eppendorf tube (in vitro) is over 10 orders of magnitude. Therefore, there is a lot more glycogen to drive the size expansion of condensates in vitro. Finally, inside cells and in contrast to the in vitro conditions, the system is out-of-thermodynamic equilibrium due to glycogen synthesis and degradation. We have revised the text to explain these points.

6. Lines 339-340: I do not see why Fig. 5I is the proof for the exclusion of fluorescent proteins by glycogen condensates

Response 26: Fig. 5I shows that there is less fluorescent signal from fluorescent proteins in the glycogen condensates relative to the background. This is consistent with glycogen condensates excluding fluorescent proteins. If there were no exclusion effect, we would expect the fluorescent signal to be equally distributed between condensates and the surrounding milieu, which is not what we observe. We have revised the text to clarify this point.

7. Proofs for glycogen phase separation in vivo are totally missing. Why didn't the authors use some conventional methods, such as addition of hexanediol (not a proof by itself, but still...) FRAP, to show reversible formation of glycogen condensates in vivo?

Response 27: Unfortunately, the conventional methods are suitable for proteins (which dominates the literature) but unfortunately not for polysaccharides such as glycogen. Protein condensates usually form through weak hydrophobic interactions, often involving intrinsically disordered regions. The amphipathic molecule, 1,6-hexanediol disrupts these hydrophobic interactions. Glycogen is a hydrophilic molecule and therefore glycogen condensates are not expected to be affected by hexanediol. Indeed, we found that 1,6-hexanediol has no effect on glycogen condensates formed in vitro. So, it is not expected to affect glycogen accumulation in vivo, which we also verified. We did not include these negative results in the revised manuscript, as they are not informative. But if deemed important, we would be happy to include them.

As for FRAP experiments, they rely on the molecule of interest being covalently labeled with a fluorophore. While covalently tagging proteins with a fluorophore (such as GFP) is straightforward, to our knowledge, no method currently exists for doing so with polysaccharides like glycogen inside cells. We mentioned this technical limitation in the original submission.

Nevertheless, to fulfill the reviewer's request, we did perform FRAP experiments on the glycogen condensates labeled with the glycogen sensor and found the kinetics to be fast, as expected for a fluid state. Our FRAP results are consistent in time scales with those previously reported for liquid glycogen condensates in liver cells (Liu et al, 2021). Our FRAP data are presented in a new supplementary figure (Fig EV7). However, we warn the readers that these results are not definitive because the reversible interactions between glycogen and its fluorescent sensor (for which the dissociation constant is unknown) may contribute to the observed fast diffusion kinetics.

Given the caveats associated with conventional methods, we sought alternative approaches to assess the mechanical properties of glycogen condensates within *E. coli*. Drawing inspiration from studies on intracellular organelles in eukaryotic cells, we developed an atomic force microscopy (AFM)-based method to measure—for the first time to our knowledge—the stiffness of intracellular regions in bacterial cells. Our findings reveal that areas containing glycogen condensates are as soft as other (liquid) cytoplasmic regions, consistent with a liquid-like state. In contrast, regions with protein aggregates (thought to be more solid-like) display higher stiffness. These AFM results are presented in the new figures (Fig 7 and Fig EV8-10).

8. What is also missing is the causality. Even if glycogen phase separates in vivo, what is the proof that this causes the rearrangements and asymmetry during the TP?

Response 28: We agree that our evidence was insufficient. As noted above, we have performed additional experiments and analyses to show that the cellular asymmetries are not present in the glycogen-deficient mutant grown in coculture with the WT cells. In addition, we performed microfluidic experiments to substantiate our claim, as mentioned above. The new data and quantitative analyses are shown in new figures (Fig 3B-F, Fig 4, Fig 5C-E, Fig EV4, and Fig EV5).

9. To go one step further, what is the proof that glycogen is "the" driver of all that, as the Title implies. It could still be one of many factors.

Response 29: These cellular rearrangements and asymmetries are not observed in mutant cells lacking the glycogen biosynthetic pathway, as mentioned above. This, together with the in vivo correlations and the in vitro evidence strongly suggest that glycogen is the main driver. We hope that the added evidence and the textual revisions will be more convincing to the reviewer.

Less major points:

1. The proofs for the asymmetric distribution of RNAs and proteins could be supported by additional experimentation. For example, by imaging the distribution of specific *E. coli* proteins and RNAs that were shown to be distributed through the cytoplasm.

Response 30; We showed the asymmetric distribution of proteins with two different fluorescent proteins (mScarlet-I and GFP), which have different sequences and origins. This is presented in Fig 1A-B, Fig. 3G, and Fig EV1B). We also show this phenotype for 8 different GFP derivatives with varying net charge (Fig EV1C). We feel that this makes our observations more generalizable to proteins in general and excludes the possibility of the localization phenotype being caused by specific interactions. The latter couldn't be excluded by using specific *E. coli* proteins. The same is true for specific *E. coli* RNAs. Furthermore, most *E. coli* mRNAs are in insufficient copy number (0.05 to 5) per cell (PMID: 20671182) to be used for this purpose. Please note that we have already used two different ways to visualize the localization phenotype of bulk RNAs and they both show similar polar exclusion (Fig. 2C-D and Fig EV1D).

2. The exact material state of the glycogen droplets has not been determined even in vitro. The authors acknowledge this fact with regard to the in vivo, where the glycogen granules were not even shown to be phase-separated condensates. Hence, they should refrain from using the term LLPS. Rather, they should call it PS.

Response 31: We have added a video (Video EV4) that show that glycogen condensates in the column are spherical and fuse with each other. Their shape only became less spherical after they settled on the surface and interacted with the glass. The observation of droplet fusion events and surface wetting is often used in the literature as a demonstration of a liquid state.

Minor points:

1. Line 20: Erase "in nature and the laboratory"

Response 32: Done

2. Line 30: suggest and likely are redundant. Omit one of them

Response 33: Done

3. Fig. 1A: Why are the OD units on the Y axis not given in the same scale used in the text for OD600?

Response 34: We want to show when the culture deviates from exponential growth, which is best shown in a logarithmic plot. We have revised this figure to include an inset that shows the linear scale to help the readers see the difference (Fig 1A).

4. Lines 128-130: Did the author mean to say that "The latter is consistent" rather than "was consistent" (which implies that they have shown it somehow)?

Response 35: Corrected. Thank you.

Dear Christine,

Thank you for submitting a revised version of your manuscript. We have now received input from all original reviewers, who are satisfied with the revisions and now recommend acceptance of the manuscript after final minor revisions. Additionally, there remain a few editorial points that need to be addressed before I can extend official acceptance of the manuscript:

1. CRediT has replaced the traditional author contributions section because it offers a systematic, machine-readable author contributions format that allows for more effective research assessment. Please remove the Authors Contributions from the manuscript and use the free text boxes beneath each contributing author's name in our online submission system to add specific details on the author's contribution. More information is available in our guide to authors.
2. Please remove Tables EV1-6 from the manuscript text and upload as separate files, one per table. Please add the corresponding legend to each table file.
3. Please correct the callouts for Table S1-5 that remain in the text.
4. Please rename movie files into Movie EV1 - EV6. The legends should be removed from the manuscript text file and zipped with each movie file. Further information is available here:
<https://www.embopress.org/page/journal/14602075/authorguide#expandedview>
5. There seems to be a mislabeling with the movie file or upload for Movies EV5 and EV6, please check.
6. Please rename "Materials and Methods" section into "Methods" and remove the reagents and tools table from the manuscript text. Reagents and Tools Table should be uploaded as a separate file choosing the file type "Reagent Table" and using our template, which you can find in our author guidelines:
<https://www.embopress.org/page/journal/14602075/authorguide#structuredmethods>
7. Please check the order of the figure callouts, as they should be mentioned in the text in a sequential manner. Currently, Fig EV3 and EV4 are called out before Fig EV5, Fig EV10 is called out before Fig EV9B,C.
8. In the Data Availability section, please add a resolvable link for S-BIAD2088 dataset. More information about the format of this section can be found here: <https://www.embopress.org/page/journal/14602075/authorguide#dataavailability>.
9. Source data for figure panel 3H appear to be mislabelled as 3G, please check.
10. In our standard image integrity check, we noted a potential image reuse between figure panels 6B (10 nm) and 6G (10mM). Please check.
11. Our data editors have flagged the following issues in figure legends that need correcting:
 - Please indicate the statistical test used for data analysis in the legends of figures 5C-E.
 - Please define the box plots in terms of minima, maxima, centre, bounds of box and whiskers, and percentile in the legends of figures 5C-E; EV5.
 - Please define the box plots in terms of minima, maxima, bounds of box and whiskers, and percentile in the legend of figure 7E.
 - Please provide information on the number and nature of replicates in the legends of figures 5C-E; 6A.
 - Please define the error bars in the legend of figure 6A.
12. Papers published in The EMBO Journal are accompanied online by a 'Synopsis' to enhance discoverability of the manuscript. It consists of A) a short (1-2 sentences) summary of the findings and their significance, B) 3-4 bullet points highlighting key results and C) a synopsis image that is 550x300-600 pixels large (width x height, jpeg or png format). You can either show a model or key data in the synopsis image. Please note that the image size is rather small and that text needs to be readable at the final size.

With best wishes,

Ieva

We realize that it is difficult to revise to a specific deadline. In the interest of protecting the conceptual advance provided by the work, we recommend a revision within 3 months (27th Nov 2025). Please discuss the revision progress ahead of this time with the editor if you require more time to complete the revisions.

Referee #1:

The authors have addressed all my concerns and the manuscript is ready for publication.

Referee #2:

The authors have adequately addressed my major points raised in the previous review. In particular, the addition of AFM experiments to assess the material properties of glycogen condensates is an excellent approach. Congratulations on this intriguing data. Overall, the manuscript has significantly improved and is now clearer and stronger.

However, I advise the authors to tone down some of their mechanistic claims regarding glycogen phase separation. While phase separation of glycogen is demonstrated *in vitro*, it remains unclear whether a phase transition specifically underlies glycogen accumulation at the cell poles *in vivo*, and consequently, whether glycogen accumulation drives the observed intracellular rearrangement and asymmetric division. Although the correlation between glycogen accumulation and rearrangement is compelling, causality is not definitively demonstrated.

The current data set does not establish that an LLPS mechanism specifically drives the observed phenotypes. This point is crucial, especially given the broader community's ongoing discussions around condensate interpretations and their functional roles. Specifically, the lack of mutations or other perturbations that specifically alter the mechanism of condensation without completely eliminating glycogen synthesis makes it challenging to confidently ascribe a functional role to LLPS. The manuscript would benefit from explicitly acknowledging this limitation and emphasizing that further work is needed to clarify the mechanistic relationship.

Additionally, there is some missing information in Figure 6 that would improve clarity. Specifically, the concentrations used in panels C and D should be explicitly stated, and the incubation time on coverslips should be indicated clearly. If the incubation time or the specific settings on the coverslip significantly affect the observed material states, this should be explicitly described. Similarly, panel E would benefit from clarification regarding the total number of measurement points, their spatial selection criteria, and clear reporting of summary statistics such as mean and standard deviation.

In summary, while the manuscript is much improved and the AFM data are particularly noteworthy, careful attention to accurately qualifying claims and providing additional methodological details in the highlighted sections would further enhance the clarity and impact of the work.

Referee #3:

The revised manuscript "Glycogen phase transition drives macromolecular rearrangement and asymmetric division in *Escherichia coli*" by Thappeta and colleagues provides now a wealth of new data that are, in summary, supportive of their clearly formulated hypothesis. The authors replied to all points raised by myself and also the points brought up by the other referees in a clear and comprehensive way. The revised work contains several important new experimental additions. In my opinion the most impressive (and supportive) evidence for the LLPS formation of glycogen is the AFM study shown in the new Fig. 7 and its related figures EV8-10. These experiments are complemented by new FRAP data shown in new Fig. EV7. There is also additional data on nucleoid positioning (Fig. 1, EV5) and a more comprehensive analysis of the cellular asymmetry in glycogen-producing cells and mutants deficient in glycogen biosynthesis (new data in Fig. 3-5, respectively). The authors use now a mother machine microfluidic chamber to follow the gradual accumulation of glycogen at the old pole and conclude that the old pole accumulation is inherited over generations (new Fig. 4).

In summary, the authors have done a great job in revising this manuscript and I have only very minor points that should be addressed before publication.

The microfluidic experiments shown in Fig. 4 are an important addition. However, I wonder why Fig. 4B only shows one or two cells? There should be a line of cells? Do the offspring cells not establish a glycogen condensate? How was transition phase behavior mimicked in the mother machine setup?

Fig. 7B: Protein aggregates can be seen best in DIC or phase contrast. Absence of fluorescence could, in theory have several reasons.

At few places there are spaces missing between words (Line 368: Space missing: events(Woldringh et al, 1990). - check ms in a final round of revision.

Referee #4:

The revised manuscript presents a greatly improved version of the study of Thappeta et al. All in all, the authors made real efforts to address the different points raised by me (and from what I can superficially tell, also by the other reviewers). They undertook technical challenges, such as doing experiments in a mother machine, and found out-of-the-box solutions, such as the development of an AFM-based method to measure condensates stiffness.

My main concern - the lack of a direct proof for glycogen being the driving force for the rearrangements - has been addressed and answered in a satisfying way.

The experimental proofs provided by the authors in response to the other points I raised, plus the explanations and clarifications in the revised text are satisfactory. Although I do not fully agree with the deterministic manner that the authors draw conclusions in some cases, e.g., that the glycogen condensates are certainly in liquid and not a gel state, I understand that some issues cannot be indisputably solved at the present time.

Hence, I find the revised MS suitable for publication in EMBO journal.

Referee #1:

The authors have addressed all my concerns and the manuscript is ready for publication.

Response: Thank you!

Referee #2:

The authors have adequately addressed my major points raised in the previous review. In particular, the addition of AFM experiments to assess the material properties of glycogen condensates is an excellent approach. Congratulations on this intriguing data. Overall, the manuscript has significantly improved and is now clearer and stronger.

Response: Thank you!

However, I advise the authors to tone down some of their mechanistic claims regarding glycogen phase separation. While phase separation of glycogen is demonstrated in vitro, it remains unclear whether a phase transition specifically underlies glycogen accumulation at the cell poles in vivo, and consequently, whether glycogen accumulation drives the

observed intracellular rearrangement and asymmetric division. Although the correlation between glycogen accumulation and rearrangement is compelling, causality is not definitively demonstrated.

Response: In our discussion section, we offer mechanistic proposals and not claims. We purposely state: “We propose...” or use the word “suggest(s)”. We don’t speak of demonstration or definitive proof. To be even more careful, we have revised the discussion to add “our data suggests that” when speaking of cell division resulting in the inheritance of glycogen condensates. Overall, we feel that our tone is appropriate for the presented evidence. We also think that an alternative interpretation is not easy to imagine.

This said, we agree that a specific aspect of our proposed mechanism (i.e., the physiological role of phase separation) is compelling but not definitive (see also response below). We added a note to acknowledge this limitation in the Discussion.

The current data set does not establish that an LLPS mechanism specifically drives the observed phenotypes. This point is crucial, especially given the broader community's ongoing discussions around condensate interpretations and their functional roles. Specifically, the lack of mutations or other perturbations that specifically alter the mechanism of condensation without completely eliminating glycogen synthesis makes it challenging to confidently ascribe a functional role to LLPS. The manuscript would benefit from explicitly acknowledging this limitation and emphasizing that further work is needed to clarify the mechanistic relationship.

Response: We agree and have added an explicit acknowledgement of this limitation in the Discussion (lines 625-630).

Additionally, there is some missing information in Figure 6 that would improve clarity. Specifically, the concentrations used in panels C and D should be explicitly stated, and the incubation time on coverslips should be indicated clearly. If the incubation time or the specific settings on the coverslip significantly affect the observed material states, this should be explicitly described. Similarly, panel E would benefit from clarification regarding the total number of measurement points, their spatial selection criteria, and clear reporting of summary statistics such as mean and standard deviation.

Response: We have added information about replicates, concentrations, incubation times, etc. to the legends of Fig 6C and D. No differences in material properties were observed, though we did not widely vary the imaging conditions (e.g., incubation time on the glass-bottom dishes).

Regarding Fig 6E, we have added information about the number and type of replicates. We did not, however, add mean and standard deviations as the results were determined

based on whether (or not) the mixture resulted in droplet or aggregate formation through visual inspection. This is now clarified in the legend.

In summary, while the manuscript is much improved and the AFM data are particularly noteworthy, careful attention to accurately qualifying claims and providing additional methodological details in the highlighted sections would further enhance the clarity and impact of the work.

Response: We agree and have complied. We appreciate the valuable feedback.

Referee #3:

The revised manuscript "Glycogen phase transition drives macromolecular rearrangement and asymmetric division in *Escherichia coli*" by Thappeta and colleagues provides now a wealth of new data that are, in summary, supportive of their clearly formulated hypothesis. The authors replied to all points raised by myself and also the points brought up by the other referees in a clear and comprehensive way. The revised work contains several important new experimental additions. In my opinion the most impressive (and supportive) evidence for the LLPS formation of glycogen is the AFM study shown in the new Fig. 7 and its related figures EV8-10. These experiments are complemented by new FRAP data shown in new Fig. EV7. There is also additional data on nucleoid positioning (Fig. 1, EV5) and a more comprehensive analysis of the cellular asymmetry in glycogen-producing cells and mutants deficient in glycogen biosynthesis (new data in Fig. 3-5, respectively). The authors use now a mother machine microfluidic chamber to follow the gradual accumulation of glycogen at the old pole and conclude that the old pole accumulation is inherited over generations (new Fig. 4).

In summary, the authors have done a great job in revising this manuscript and I have only very minor points that should be addressed before publication.

Response: Thank you!

The microfluidic experiments shown in Fig. 4 are an important addition. However, I wonder why Fig. 4B only shows one or two cells? There should be a line of cells? Do the offspring cells not establish a glycogen condensate? How was transition phase behavior mimicked in the mother machine setup?

Response: We apologize for the confusion.

Regarding the three first questions: Indeed, Fig 4B only shows the evolution of the mother cell (cell at the end of the microfluidic trench) because we computationally extracted this cell (i.e., remove the progenies) to help the reader visualize the relevant information. This is meant to be illustrative as we provide the quantitative analysis of 130 lineages in the

following figure panels (Fig. 4C-E). In the revised manuscript, we now clearly state what is shown (and why) in the text related to Fig. 4B and also provide a video (Movie S2) that shows the corresponding data for all the cells in a lineage (mother cell and progenies) over time.

Regarding the fourth question: The design of the microfluidic system is such that the cells in the microfluidic trenches experience the same culture conditions than those in a culture flask (see schematic in Fig 4A). As a result, the cells in the trenches undergo the same transition from exponential to stationary phases as cells growing in a batch culture. This design was originally described by the Paulsson lab (PMID: 34017106). In our study, we demonstrate that the mother cells that we tracked did indeed undergo transition phase over time (generations) by quantitatively showing the decrease in growth rate (Fig EV6). This is explained in the text related to Fig 4.

Fig. 7B: Protein aggregates can be seen best in DIC or phase contrast. Absence of fluorescence could, in theory have several reasons.

Response: We have added the brightfield image in Fig EV8 (panel A) next to the corresponding mTagBFP2 fluorescence image (duplicated from Fig 7B) to demonstrate that the absence of mTagBFP2 fluorescence corresponds to the increased contrast in the brightfield image, consistent with the presence of protein aggregates. The image duplication, which was done to facilitate direct comparison, is mentioned in the legend of Fig EV8.

At few places there are spaces missing between words (Line 368: Space missing: events(Woldringh et al, 1990). - check ms in a final round of revision.

Response: Apologies for these typos, They have been fixed.

Referee #4:

The revised manuscript presents a greatly improved version of the study of Thappeta et al. All in all, the authors made real efforts to address the different points raised by me (and from what I can superficially tell, also by the other reviewers). They undertook technical challenges, such as doing experiments in a mother machine, and found out-of-the-box solutions, such as the development of an AFM-based method to measure condensates stiffness.

Response: Thank you for your appreciation of our efforts.

My main concern - the lack of a direct proof for glycogen being the driving force for the rearrangements - has been addressed and answered in a satisfying way.

Response: Thank you.

The experimental proofs provided by the authors in response to the other points I raised, plus the explanations and clarifications in the revised text are satisfactory. Although I do not fully agree with the deterministic manner that the authors draw conclusions in some cases, e.g., that the glycogen condensates are certainly in liquid and not a gel state, I understand that some issues cannot be indisputably solved at the present time.

Response: We appreciate the reviewer's understanding. Nevertheless, we agree with the reviewer's lingering concern. Therefore, we have revised the discussion to acknowledge that we cannot rule out a gel-like state in vivo (lines 589-591).

Hence, I find the revised MS suitable for publication in EMBO journal.

Response: Thank you!

Dear Christine,

Thank you for incorporating the final formatting requests in the manuscript. I am now pleased to inform you that your manuscript has been accepted for publication. Congratulations with a great study!

Before we forward your manuscript to our publishers, we would like to propose some edits in the manuscript abstract and the summary paragraph of the synopsis - please see below and in the attached file. I have also written a short blurb that will accompany the title of your manuscript in our online table of contents. Please take a look and let me know if any corrections are needed.

Blurb:
Stationary-phase *E. coli* cells exhibit asymmetric division and macromolecule localization due to uneven polar accumulation of glycogen condensates.

Synopsis:
Subcellular organization of bacterial cells upon their exit from exponential growth phase has not been well investigated. This study shows that stationary-phase *E. coli* cells exhibit asymmetric division and macromolecule localization due to uneven accumulation of glycogen between the cell poles over divisions.

If you have any questions, please do not hesitate to contact the Editorial Office or me directly. Thank you for your contribution to The EMBO Journal!

Best wishes,

Ieva
